# A scalable reinforcement learning approach for screening large peptide libraries for bioactive peptide discovery

Mohit Pandey [1,2], Jane Foo[1,2,8], Shabnam Massah[1,2,8], Morgan A. Alford[3,8], Hazem Mslati[4,8], Gopeshh Subbaraj[5], Mira Saba[1], Francesco Gentile [4,6], Nada Lallous [1,2], Evan F. Haney [3], Robert E. W. Hancock [3], Martin Ester[7] & Artem Cherkasov [1,2] ✉

Bioactive peptides such as anticancer peptides (ACPs) offer a promising therapeutic alternative to small molecules due to their efficiency and selectivity against tumors and minimal toxicity towards healthy human cells. However, their rational discovery requires navigating a vast chemical space using computationally demanding in silico tools. Herein, we present a computational method enabling cost-efficient exploration of large peptide libraries using reinforcement learning and posterior sampling. Practical application of the developed approach results in identification of membranolytic peptides with therapeutic potential. The developed computational method reduces the search space by over 90% compared to exhaustive library screening and enables effective balancing between dataset's exploration and exploitation. We demonstrate the scalability of this method by screening a focused library of 36 million structurally resolved helical peptides curated from the Protein Data Bank. When screened in in vitro assays, 15 of the top 100 selected candidates exhibit cytotoxic activity against breast cancer cells including drug resistant triple-negative breast cancer, with the three lead compounds further characterizing as non-toxic towards healthy human cells. This study highlights the potential of using deep reinforcement learning to expedite bioactive peptide discovery, offering a promising path for developing new peptide-based cancer therapies.

Cancer continues to be one of the most formidable human health threats[1], driving a persistent search for novel and innovative therapies[2]. Amidst the variety of emerging approaches, peptide-based therapeutics have risen to prominence due to their ability to selectively target cancer cells[3]. A specific class of anti-cancer peptides (ACPs), e.g., Mastoparan, are typically composed of 10–60 amino acids, are

positively charged and form specific amphiphilic configurations that enable effective, selective binding and disruption of cancer cell membranes. Often, such ACPs possess a unique direct-acting mechanism of anti-cancer activity, and bring several important advantages, such as low cost of synthesis, high specificity and reduced toxicity[3–8]. Evidence of peptides' therapeutic potential are growing and

[1]Vancouver Prostate Centre, University of British Columbia, Vancouver, BC, Canada. [2]Faculty of Medicine, University of British Columbia, Vancouver, BC, Canada. [3]Centre for Microbial Diseases and Immunity Research, Department of Microbiology and Immunology, University of British Columbia, Vancouver, BC, Canada. [4]Department of Chemistry and Biomolecular Sciences, University of Ottawa, Ottawa, ON, Canada. [5]Mila, Université de Montréal, Montreal, QC, Canada. [6]Ottawa Institute of Systems Biology, Ottawa, ON, Canada. [7]School of Computing Science, Simon Fraser University, Burnaby, BC, Canada. [8]These authors contributed equally: Jane Foo, Shabnam Massah, Morgan A. Alford, Hazem Mslati. ✉e-mail: acherkasov@prostatecentre.com

databases like DrugBank list 29 ACP drugs[9], with over 350 clinical trials completed and more than 160 currently underway[5,10].

Despite these promising developments, a major challenge in advancing ACP therapy development lies in searching through the vast combinatorial space of possible sequence variations. Virtual screening (VS) methods offer a solution by enabling efficient screening of large peptide databases to identify therapeutic candidates[11–14]. Since the early 2000s, our group and others have demonstrated the effectiveness of quantitative structure activity relationship (QSAR) descriptors and models for discovering therapeutic peptides[15–18]. More recently, machine learning (ML) and deep learning (DL) techniques relying on QSAR models have gained prominence due to their improved predictive power. In these works, a QSAR model is trained on molecular descriptors to predict peptide potency[17,19–24]. MLACP[25], AntiCP 2.0[26], and ACPred-Fuse[27], rely on descriptors like amino acid composition (AAC) and dipeptide composition (DPC) combined with ML classifiers, including support vector machines (SVM), random forests (RF), and LightGBM. Deep learning models, such as ACP-DL[28], DeepACP[29], and ACP-check[30], employ architectures like Bidirectional- Long Short-Term Memory networks, Convolutional Neural Network, and Transformer networks to capture sequence-based patterns and physicochemical properties more effectively. More recent efforts, such as ACP-ESM, leverage large-scale protein language models like ESM[31] to enhance predictive accuracy. Several recent reviews have focused on broad use of artificial intelligence approaches for ACP predictions (though without experimental validation)[32–35]. In contrast, recent work by Yue et al.[36] represents a notable step towards integrating DL-methods with experimental validation by predicting ACP activity of 3.8 million sequences from UniProt and 100,000 sequences produced by deep generative models. However, scaling such approaches to ultra-large peptide libraries still remains a hurdle due to computational expense for QSAR descriptors as well as DL models training and inference over large libraries.

The scalability limitation is evident from many recent DL-based studies of bioactive peptides that have only managed to screen libraries containing a few million sequences[11,36,37]. The challenge, therefore, is scaling QSAR- and DL-based VS approaches to handle much larger peptide datasets. Thus, we hypothesize that by focusing on activity cliffs−regions where peptides exhibit high cell inhibition towards a particular breast cancer cell, within peptide libraries−can effectively prioritize high-potential candidates. This approach can significantly reduce the number of calls to computationally expensive deep QSAR model, allowing for a more efficient exploration of ultra-large libraries by concentrating resources on promising candidates and avoiding the computationally expensive processing of inactives.

As a solution to such scalability problem, we propose Target Adaptive Reinforcement Learning for Sampling Activity Landscapes (TARSA). TARSA leverages reinforcement learning (RL) and Markov Chain Monte Carlo (MCMC) to explore peptide libraries by identifying regions rich in potential high-scorers and then explore the local neighborhoods of such regions. This approach parallels sparse sampling techniques in high-dimensional NMR spectroscopy[38–40], which accelerate data acquisition by leveraging the inherent sparsity of NMR spectra. Instead of acquiring a full Nyquist-sampled dataset, sparse sampling selectively captures a subset of data points in the indirect dimensions (such as frequency), significantly reducing experiment time while preserving spectral resolution. Similarly, TARSA strategically samples a subset of peptides from a library where most peptides are inactive to maximize the discovery of active candidates while minimizing computational costs. Unlike NMR spectra, which represent continuous signals, TARSA operates on a discrete peptide space, adapting sparse sampling principles to a biological search framework.

Here, we show that by leveraging scalability and computational efficiency, the developed TARSA algorithm enables fast screening of a comprehensive library comprising all experimentally resolved α-helical segments derived from known protein structures within the Protein Data Bank (PDB). We find that 15 of the top 105 candidates identified through TARSA exhibit cytotoxic activity against Triple-Negative Breast Cancer (TNBC) cells in in vitro assays. The top three candidates demonstrate broad-spectrum efficacy against other sub-types of breast cancer while exhibiting selective non-toxicity towards healthy human peripheral blood mononuclear cells (PBMCs). To our knowledge, this study represents the largest HTVS effort for therapeutic peptide discovery, accompanied by cost-effective in vitro validation of over 100 candidates. These findings underscore the potential of TARSA, which integrates RL-driven exploration with efficient screening, as a fast, scalable, and cost-effective platform for identifying synthesizable peptides with validated membranolytic activity.

## Results

Our peptide screening workflow, PepSce, was designed to systematically refine ultra-large peptide libraries for therapeutic activity (Fig. 1A). As a demonstrative example, we sought ACP Mastoparan-like helical peptides with membranolytic activity. Since, helical peptides are the most extensively studied structural class of ACPs[41–45], and shorter peptides are preferred due to cost-effective synthesis. Therefore, the presented approach prioritized the discovery of short helical candidates by screening peptides with putative helical structure as provided by DSSP, consistent with earlier works[46]. To construct a suitable search space, we extracted all unique 5–15-mer peptide sequences ($n = 1.6 \times 10^9$) from the PDB and applied secondary structure filtering to retain only helical peptides. This resulted in PDB$_{large}$, a dataset containing 36 million experimentally resolved helical peptide sequences and fragments Table 1 summarizes the different datasets used in this work; see Supplementary Method 1 for library curation details.

Computational screening of such vast peptide space was driven by reinforcement learning (RL), leveraging policy transfer[47–49], a technique in which decision-making knowledge from a pretrained policy in a source task is adapted to a similar target task to accelerate learning and improve generalization (Fig. 1B). RL has demonstrated strong performance in exploring large chemical spaces, where RL-based generative models efficiently sample the continuous chemical landscape[50–52]. Inspired by these results, we developed TARSA, an RL-based algorithm tailored for navigating large but finite peptide libraries, with the goal of prioritizing the identification of membranolytic candidates (Fig. 2A).

TARSA trained an RL agent[53] to explore a two-dimensional principal component analysis (2D-PCA) projection of the peptide library, aiming to identify activity cliffs (Fig. 2B). The RL agent received reward signals from a deep QSAR model (Fig. 2C), trained to predict the percentage inhibition of TNBC cells (MDA-MB-231), allowing it to focus exploration on peptides with higher membranolytic activity. To optimize exploration, activity cliff identification was framed as a multi-armed bandit problem[54], where each activity cliff was treated as an independent "bandit", and posterior sampling guided the agent toward the most promising regions. By learning reward-maximizing navigational trajectories−paths through the 2D-PCA space associated with highly potent peptides−the RL agent could efficiently prioritize high-potential candidates while minimizing the computational burden of exhaustive QSAR evaluations on all peptides in this 2D-PCA space. To scale TARSA for ultra-large peptide libraries, the dataset was partitioned into manageable subsets, and screening was parallelized across multiple computational nodes, enabling efficient exploration of millions of peptides (Fig. 1B).

The proposed PepSce workflow included four key phases: motif discovery, candidate search, cytotoxicity filtering, and in vitro validation (Fig. 2D). In the motif discovery phase, TARSA was applied to PDB$_{large}$ to identify potent anticancer motifs, which were subsequently

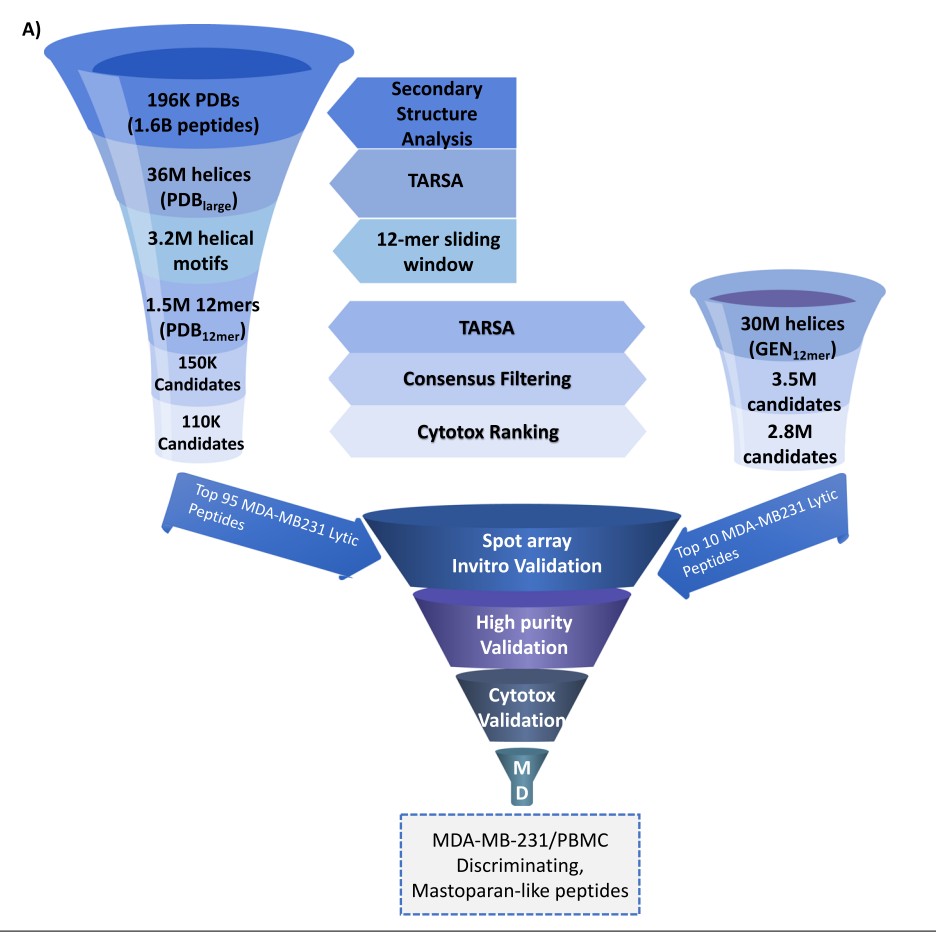

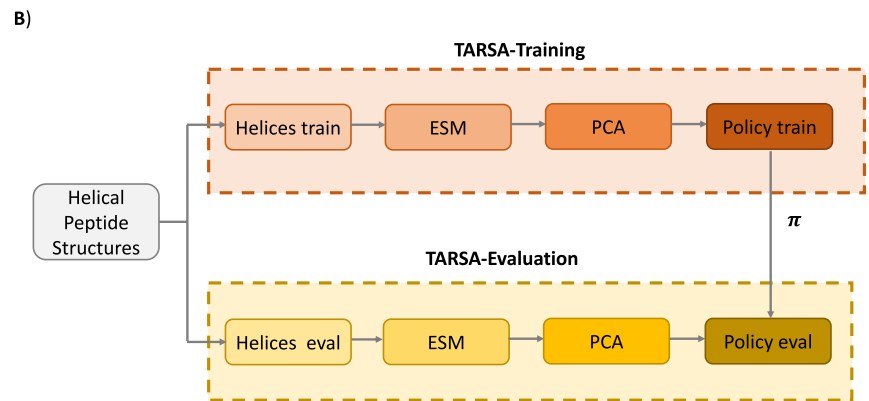

**Fig. 1 | Pepsce Workflow and Target Adaptive Reinforcement Learning for Sampling Activity Landscapes (TARSA) Training for Helical Peptide Search Space Reduction. A** Phase-wise reduction of the search space of helical peptides through various phases of PepSce workflow. TARSA is the Reinforcement Learning-component of PepSce workflow responsible for screening libraries. **B** Training TARSA policy on a subset of screening dataset ($D_{train}$) and then using the trained policy to screen remainder of the dataset in evaluation mode.

used to construct a library of fixed-length 12-mer peptides (PDB$_{12mer}$). In the candidate search phase, TARSA was again utilized to screen PDB$_{12mer}$ and discover promising candidates. In cytotoxicity filtering phase, pre-trained toxicity models were then employed to exclude peptides predicted to be toxic to PBMCs. Ultimately, for the in vitro validation, 105 ACP candidates were selected and synthesized using a cost-effective cellulose peptide array[55]. Of these, 15 top-ranked hits were synthesized at high purity and tested for their half-maximal inhibitory concentration (IC50) against breast cancer cells (MDA-MB-231), PBMCs, and red blood cells (RBCs).

## Descriptor relevance in QSAR modeling for peptide potency
Accurate peptide modeling and potency prediction are essential for large-scale screening efforts in bioactive peptide discovery. As such, in this study, the effectiveness of various peptide representations for modeling membranolytic Mastoparan-like helical peptides, and ML algorithms to predict the cell inhibition of MDA-MB231 breast cancer cells at 12.5 μM were investigated. Five distinct peptide representations were evaluated: three QSAR descriptors- Modlamp[15], iFeatures[17], and inductive descriptors[56], and two data-driven representations- auto-encoders, and Evolutionary Scale Modeling[31] (ESM) from pre-trained

**Table 1 | Summary of datasets used in this work**

| Dataset | Labels | Application | Size | Description |
|---|---|---|---|---|
| Mastoparan MDA-MB231 | Continuous | ACP Oracle Proxy Training | 210 | Cell inhibition at 12 µM of Mastoparan single amino acid substitution derivatives on TNBC cells. |
| Mastoparan PBMC | Continuous | Normal Cells Filtering | 210 | Cell inhibition at 12 µM of Mastoparan derivatives on healthy human blood cells |
| CancerPPD | Binary | Hits Filtering | 590 | Dataset of known unique cancer peptides in public domain |
| PDB-12mer | Unlabeled | TARSA Training & Evaluation | 1.36 M | Unique 12-mers from helices in entire PDB |
| PDB-large | Unlabeled | TARSA Training & Evaluation | 36 M | 5-15-mers from helices in entire PDB |
| Gen-12mer | Unlabeled | TARSA Training & Evaluation | 30 M | Helical 12-mer ACPs sampled from a probabilistic generative model. |

*ACP* anti cancer peptide, *TARSA* target adaptive reinforcement learning for sampling activity landscapes

protein language models (Supplementary Note 3). These representations have been previously shown to work well in bioactivity prediction across multiple peptide-based studies[15–17,57]. Given the high computational cost associated with 3D descriptors, such as inductive descriptors, this study focused exclusively on 2D descriptors, which prioritized computational efficiency without sacrificing predictive power.

For each representation, random forest[58], XGBoost[59], and Multi-Layer Perceptron (MLP) regression models[60] were trained to assess predictive performance. The selection of the ML approach was guided by the need to balance prediction speed and accuracy for efficient virtual screening of ultra-large libraries. Simpler models were prioritized over computationally intensive architectures, such as Transformer-based or large-language models, which were deemed impractical for this task. Notably, MLP models consistently outperformed tree-based regressors (random forest and XGBoost) due to their higher capacity and ability to handle the complexity of these representations. Despite a relatively small dataset size $D_{Mastoparan}$ ($n = 210$)—a challenge frequently encountered in early-stage drug discovery efforts, the experiments revealed that QSAR descriptors—particularly Modlamp and iFeatures—were consistently more predictive than data-derived representations (Fig. 3A). Models trained on these descriptors yielded higher Pearson correlation values (highest $\rho = 0.65 \pm 0.1$ for ifeature with MLP), illustrating their ability to capture the peptide characteristics influencing bioactivity.

To further enhance predictive power, a fusion model integrating MLP and convolutional neural networks (CNNs) was developed using Modlamp and iFeature descriptors. This approach significantly improved performance, achieving a 5-fold cross-validated Pearson correlation of $0.81 \pm 0.04$ (Fig. 3B) for MDA-MB231 cells and $0.83 \pm 0.10$ for PBMC cells (Fig. 3C). Additionally, this model identified key physicochemical properties—such as sequence length, charge density, and hydrophobic moment (Fig. 3D)—which are well-established determinants of peptide bioactivity, particularly in helical peptides[61,62]. Training details for this model can be found in Supplementary Note 4, Supplementary Method 1. Hereafter, this optimized membranolytic bioactivity predictor is referred to as oracle proxy $f_\theta^o$.

A major bottleneck in large-scale inference with $f_\theta^o$ over massive peptide libraries was the computational expense of its constituent Modlamp and ifeatures QSAR descriptors that needed to be computed for entire ultra-large peptide screening library. Though faster than 3D descriptors, these descriptors too can be slow to compute (Supplementary Table 1). Estimates indicated that screening a 36-million-peptide library could require up to two years on a single CPU. TARSA's RL component overcame this expense by selectively identifying potent inhibitors, limiting the need for costly descriptor calculations to only these potent peptides.

## Large-scale virtual screening of peptides with TARSA discovered potent motifs

The main goal of TARSA was to process an ultra-large peptide library by ensuring that only a manageable number of samples needed to be screened using computationally expensive oracle proxy $f_\theta^o$. In subsequent experiments, the screening capabilities of TARSA were investigated, and its scalability was demonstrated. To this end, a large peptide library, PDB_large, was processed by training TARSA's RL component on 2D-PCA projections of 2 million random peptides drawn from the library.

Through this training, TARSA learned a policy that enabled it to selectively avoid screening the vast majority of peptides in the ultra-large library that are predicted to be inactive or less relevant. Instead, the agent mapped reward-maximizing paths toward regions enriched with peptides predicted to have high MDA-MB231 cell inhibition, reducing the need for brute-force screening across the entire peptide library (Supplementary Fig. 22). Once the policy was trained, a batched rollout strategy was used for screening the remaining peptides in the library. Rather than processing all remaining peptides in a single pass, TARSA divided the remaining library into multiple smaller, randomly drawn batches for screening. Each batch contained peptides sampled from the same distribution as the policy training data, ensuring the policy remained valid throughout screening (Supplementary Note 6). This batched rollout strategy effectively mitigated two issues associated with single-pass online inference over large libraries. First, since PCA reduces high-dimensional peptide features into two dimensions, some peptides with similar properties may cluster together. Screening in batches instead of a single pass reduced the risk of overlooking relevant peptides due to projection overlap. Second, processing peptides in batches allowed screening to be distributed across multiple computational nodes, significantly accelerating the overall inference process. When the trained policy was applied to batches from the library, the agent no longer queried the computationally expensive $f_\theta^o$ model for every peptide rather, only for those deemed to be bioactive by the learned policy. As a result, TARSA processed the entire PDB_large library in just 14 days, yielding 3.2 million unique motifs with predicted cell viability inhibition exceeding 40%, and identifying a total of 4.5 million peptides with an average predicted cell inhibition of 45.2% towards MDA-MB231 cells.

TARSA's performance was further evaluated on fixed-length peptide sequence libraries. This evaluation was necessary to facilitate the synthesis and experimental validation of the discovered membranolytic candidates. Two 12-mer datasets were curated for this purpose: PDB_12mer, created by applying a sliding window subsequencing to 3.2 million top peptides discovered from PDB_large (PDB_12mer curation details are in Supplementary Note 9), and GEN_12mer, composed of 30 million helical peptides sampled from a probabilistic generative model[63]. Screening on both datasets followed the same procedure as for PDB_large. On PDB_12mer, TARSA demonstrated a significant difference in predicted inhibition ($p < 0.001$) between discovered peptides (mean% cell inhibition towards MDA-MB-231 = 44.9%) and random peptides from the same dataset (mean% cell inhibition towards MDA-MB-231 = 29.8%) (Supplementary Fig 17), confirming the effectiveness of the screening protocol. Similar to PDB_large, TARSA identified hotspots (putative activity cliffs) in the peptide library PDB_12mer and mapped reward-maximizing paths to

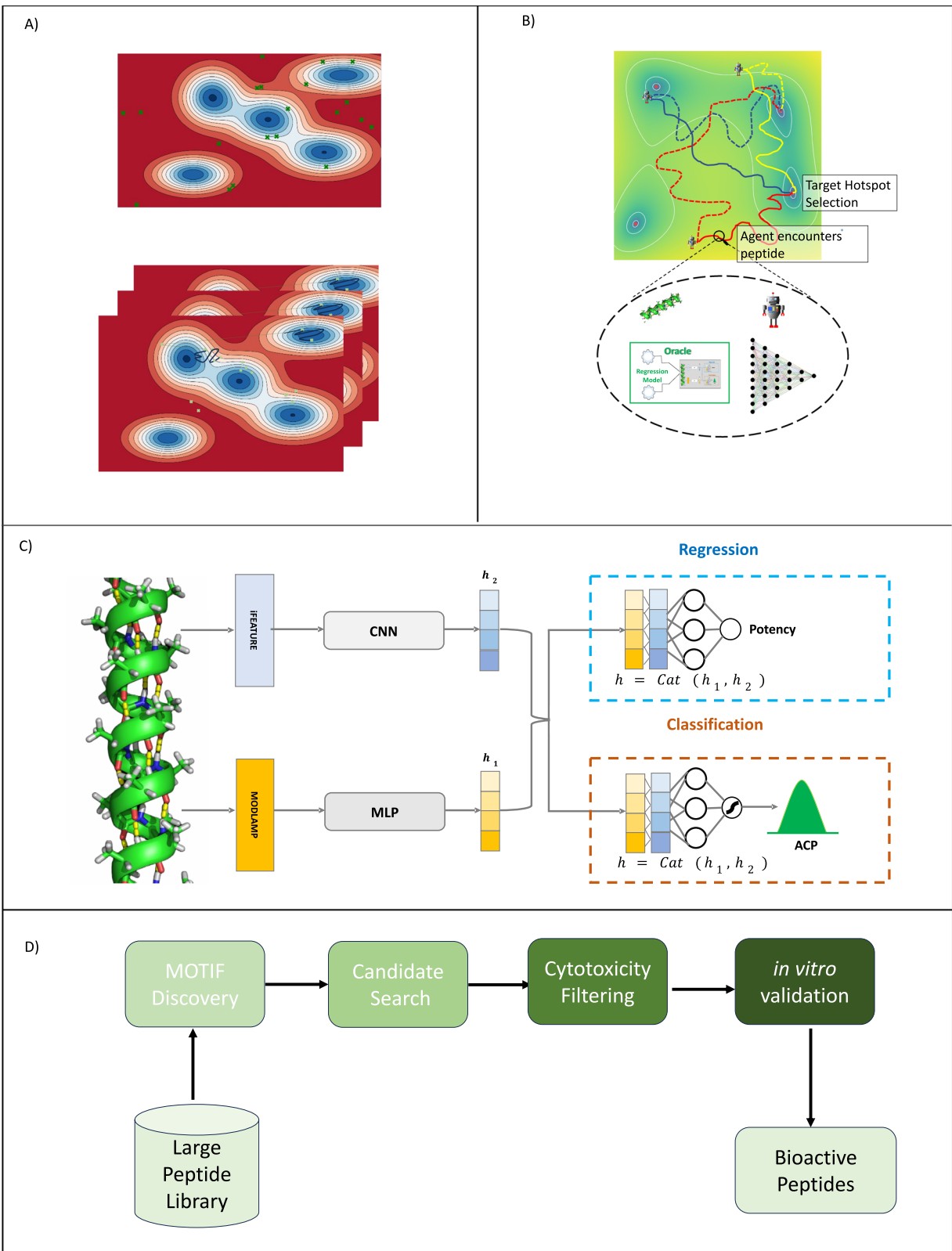

**Fig. 2 | Target adaptive reinforcement learning for sampling activity landscapes (TARSA) algorithm and model architecture for cell inhibition prediction. A** Random search in peptide chemical space is less focused, while TARSA-driven search targets activity cliffs and selectively explores chemical space. **B** Schematic diagram of TARSA algorithm training on 2D Principal Component projections of $D_{\text{train}}$. **C** Model architecture: The regression head predicts the percentage cell-inhibition of peripheral blood mononuclear cells (PBMC) and MDA-MB231 cells at 12.5 μM by peptides. The classification head predicts active vs inactive ACPs during the consensus filtering phase. **D** The four phases of PepSce workflow.

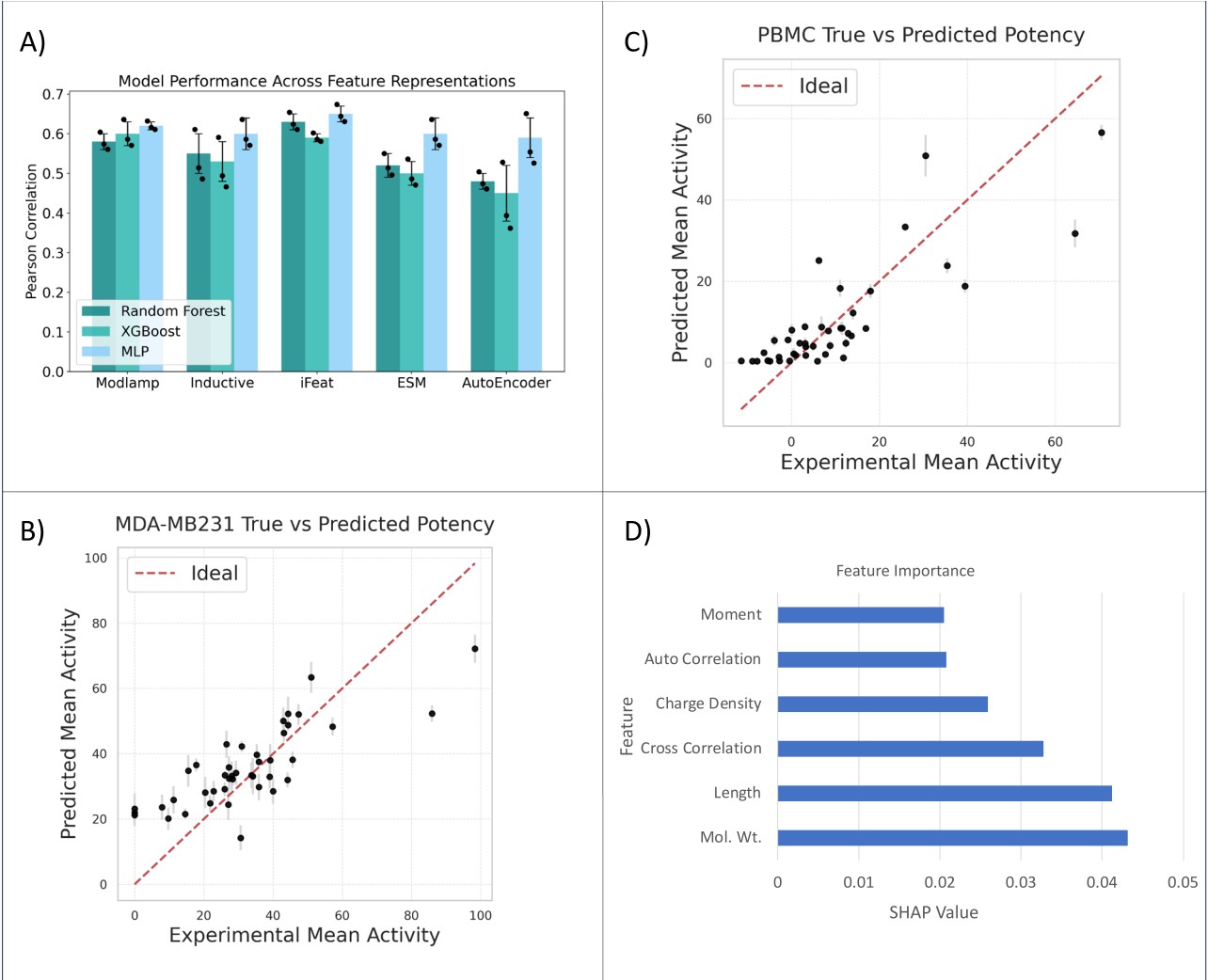

**Fig. 3 | Performance comparison and feature analysis. A** Comparison of domain-informed descriptors (Modlamp, Inductive, iFeatures) and deep representations (Evolutionary Scale Modeling, AutoEncoder), with domain-informed descriptors outperforming on small datasets $D_{Mastoparan}$. **B, C** Oracle proxy and cytotoxicity model performance on Mastoparan datasets, with cross-validated Pearson correlations of 0.83 and 0.81 corresponding to peripheral blood mononuclear cells (PBMC) and MDA-MB231 respectively. **D** Top features identified by the Oracle proxy trained on CancerPPD, aligning with known anticancer properties. Reported metrics are averaged over three independent runs with different random seeds, and error bars represent the standard deviation across these runs.

these regions, enabling the discovery of high-predicted MDA-MB231 cell inhibition peptides while ignoring less relevant ones (Fig. 4).

On $GEN_{12mer}$, TARSA also achieved significant search space reduction. Specifically, it discovered 3.5 million peptides with an average predicted MDA-MB-231 cell inhibition of 44.8%, reducing the search space by 92.5%. This result again highlighted TARSA's capacity to efficiently narrow down vast search spaces. In fact, when compared to other machine learning based screening approaches, TARSA not only achieved a faster screening time but also identified peptides with a higher average predicted MDA-MB-231 cell inhibition (Supplementary Table 2 and Supplementary Note 2).

Additionally, analysis of a randomly selected subset showed that the first-k discovered peptides had similar predicted mean MDA-MB231 cell inhibition as the top-k peptides (for $k = 100$, 1000, and 50,000), demonstrating TARSA's ability to prioritize the discovery of highly membranolytic candidates early in the screening process (Fig. 5A–C, Supplementary Fig. 16). These outcomes not only demonstrated TARSA's efficiency in screening ultra-large libraries but also supported the objective of reducing the number of calls to the QSAR model by focusing on high-potential peptides, thereby lowering the overall computational burden.

## In silico assessment of discovered anticancer peptide motifs

Before proceeding with costly peptide synthesis and experimental validation, it was essential to establish confidence in the identified candidates by evaluating their statistical properties and their ability to interact with cancer cell membranes. Statistical analysis included evaluating amino acid composition, while, membrane interaction studies assessed whether the identified peptides exhibit structural and electrostatic features conducive to disrupting cancer cell membranes, a hallmark of many known Mastoparan-like membranolytic ACPs. These in silico validations served as an intermediatory sanity check to determine whether the discovered peptides exhibit characteristics consistent with known membranolytic Mastoparan-like peptides, rather than as a filtering step to refine the candidate selection.

To this end, the amino acid distribution of the topmost 480 motifs out of the 3.2 million motifs discovered peptides from $PDB_{large}$ was evaluated. Interestingly, this amino acid distribution aligned closely with known membranolytic ACP traits. These motifs predominantly featured leucine, lysine, and isoleucine amino acids known for their roles in disrupting cancer cell membranes (Supplementary Fig. 13). Lysine's positive charge facilitated interaction with the negatively charged membranes of cancer cells[64], while hydrophobic residues like leucine and isoleucine

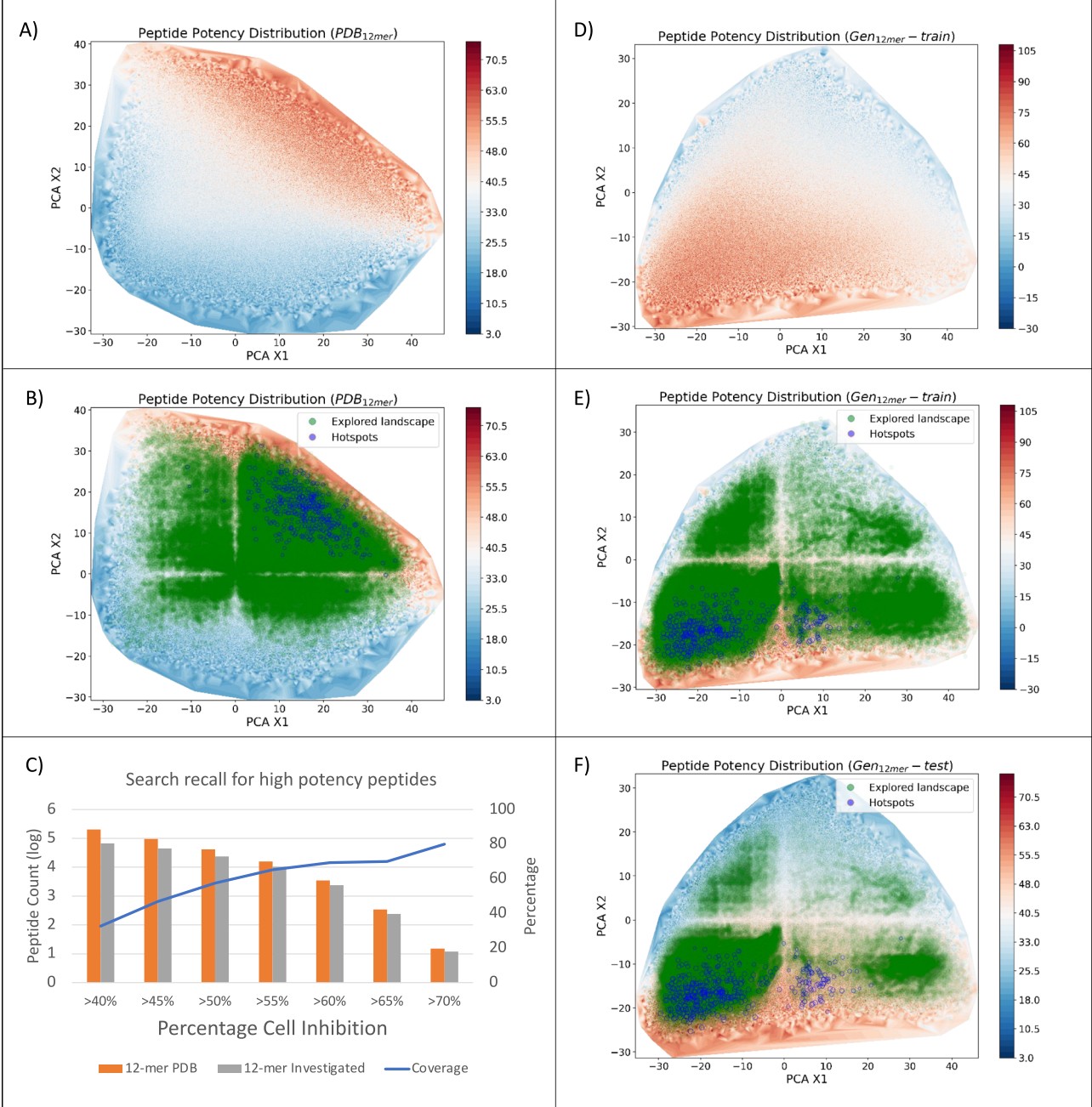

**Fig. 4 | TARSA screening results for curated peptide libraries.** The peptides discovered by running a single instance of trained policy (represented by the green dots) revealed the exploration of high-inhibition (red) regions and the avoidance of low-inhibition (blue) regions. **A, B** Screening results for PDB$_{12mer}$ library, visualized across the first two Eigen vectors of the peptide chemical space. **C** Retrieval of MDA-MB-231 inhibitory peptides from the PDB$_{12mer}$ dataset using the Target Adaptive Reinforcement Learning for Sampling Activity Landscapes (TARSA) policy. **D–F** Exploration and discovery of peptides with high predicted MDA-MB-231 inhibition in the GEN$_{12mer}$ dataset, with TARSA avoiding low-potency regions. Potency refers to predicted percentage cell inhibition of MDA-MB-231 cells.

contributed to membrane destabilization[65], crucial for peptide-induced cell death. In contrast, negatively charged residues such as glutamic and aspartic acid were scarcely observed, reflecting their reduced efficacy in membrane interaction[6,66]. The distribution of top ACP-influencing features for these motifs did not exactly match the distribution of ACPs in CancerPPD because the oracle proxy model was trained on a Mastoparan-derived dataset, which inherently constrained the physico-chemical properties of the discovered peptides to those of amphiphatic α-helical ACPs. However, the predicted peptides still retained biologically relevant ACP properties and lay intermediate between the D$_{Mastoparan}$ and CancerPPD datasets, highlighting the model's ability to generalize beyond training data (Fig. 5D–F).

Further analysis revealed that several of the discovered motifs bore striking similarity to well-characterized ACPs. For instance, the motif YLLKALFKAL, found in the PDE3A-SLFN12 complex (PDB: 7eg0), was associated with apoptosis induction[67] while WIVIIAKYLAQWY, identified in the NEDD8-activating enzyme complex (PDB: 3gzn), was involved in cancer cell growth inhibition[68]. These motifs, along with others, including LLKLLKLL (PDB: 7nef) and LWKALALKL (PDB: 2l9a), originated from proteins reported to have antimicrobial bioactivity. This was expected, as many antimicrobial peptides share similar sequence characteristics, and numerous peptides are known to exhibit multiple biological activities, including antibacterial, antiviral, and anticancer properties[69].

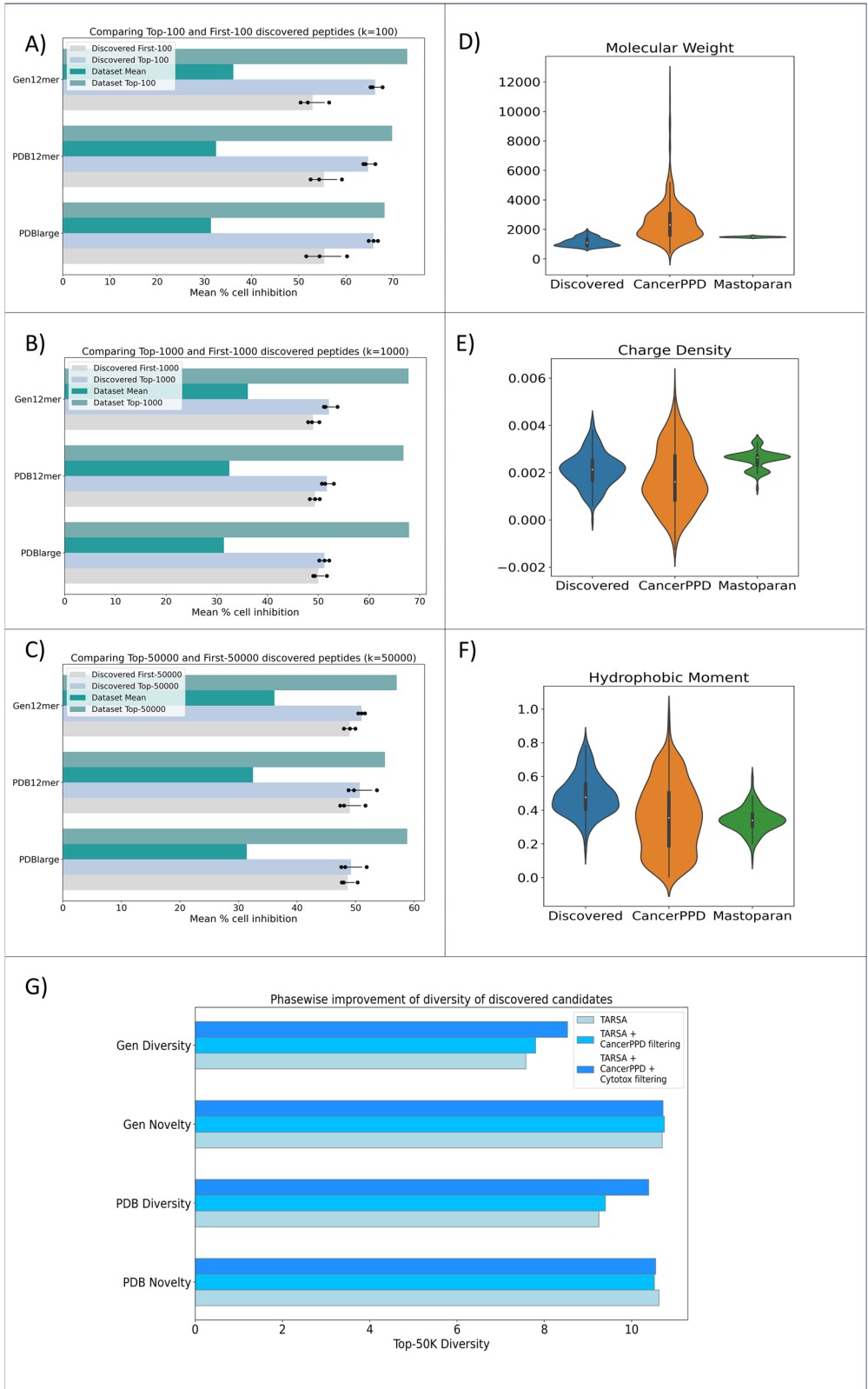

**Fig. 5 | Predicted cell inhibition, feature distribution, and diversity analysis of target adaptive reinforcement learning for sampling activity landscapes (TARSA)-discovered peptides. A–C** Mean percentage MDA-MB231 cell inhibition of TARSA-discovered peptides across three datasets compared to randomly sampled sets. The error bars represent mean ± SD ($n = 3$). **D–F** Distribution of top-ranking membranolytic features of helical peptides from PDB$_{large}$ ($n = 482$),

Mastoparan datasets, and anticancer peptides from CancerPPD. **G** Improved diversity in peptide sets after filtering with CancerPPD while maintaining novelty with respect to training data. Violin plots represent the probability density of the data. The embedded boxplots indicate the median (center line), 25th and 75th percentiles (box bounds), and whiskers extend to data points within 1.5× the interquartile range.

These analyses suggested that TARSA not only rapidly identified efficacious functional motifs from an ultra-large library but also demonstrated strong generalization by discovering motifs with known anti-cancer traits and similarities to well-characterized bioactive peptides. Additionally, in silico validation and coarse-grained molecular dynamics simulations confirmed the potential of these motifs for further experimental testing. A detailed discussion on these in silico assessments is presented in Supplementary Note 6.

### Consensus filtering and visual analysis of candidate peptides

While the in silico validation confirmed that the identified peptides shared key characteristics of amphipathic α-helical ACPs, similar to Mastoparan, an additional selection process was required to refine the candidate pool for experimental testing. All TARSA-discovered 12-mer peptides with predicted cell inhibition of ≥40% underwent consensus filtering and cytotoxicity ranking as part of the proposed screening workflow (PepSce). During the consensus filtering stage, a binary ACP classification model, trained on a diverse set of publicly available ACPs from CancerPPD and non-ACPs curated by Vijayakumar et al.[70] was employed. Since peptides in $D_{Mastoparan}$, used for the oracle proxy ($f_\theta^o$; TARSA reward) training, were compositionally homogeneous, this ensemble approach mitigated overfitting to Mastoparan-like peptides and ensured diversity among the identified hits.

Following this, cytotoxicity toward healthy white blood cells (PBMCs) at 12.5 μM was predicted for the filtered candidates ($\rho = 0.75 \pm 0.11$) (Fig. 3C). This step enriched the discovered hits for those sequences that were selectively toxic toward MDA-MB231 cells over healthy human blood (PBMC and RBC) cells. Only peptides with predicted cell inhibition of <15% toward PBMCs were retained, narrowing the library to 200,000 peptides. Each of these filtering phases showed improvement in novelty, although a slight decrease in diversity among the discovered peptides was noted, possibly due to the absence of simultaneous selectivity and heterogeneity in ACP datasets (Fig. 5G).

Subsequent to these steps, a set of 105 peptide candidates was selected for in vitro evaluation and qualitatively inspected (Fig. 6). As this study prioritized validation of PDB-derived peptide libraries through virtual screening, with generative models included as a secondary exploratory module, 95 peptides were chosen from the $PDB_{12mer}$ library and 10 peptides from the $Gen_{12mer}$ library. These peptides were dominated by hydrophobic amino acids (valine, leucine, and isoleucine) and basic amino acids (lysine and arginine), both characteristic features of known membranolytic bioactive peptides[4,71]. The predicted isoelectric points for these peptides were above neutrality (pH > 7), indicating a positive charge at physiological pH, which would enhance their electrostatic interactions with negatively charged cancer cell membranes. Additionally, the combination of a large hydrophobic moment and high isoelectric point would facilitate peptide integration into the hydrophobic region of the lipid bilayer. It was noteworthy that the physicochemical properties of these peptides deviated from those of the Mastoparan training dataset, which consisted of 210 derivatives limited to single amino acid substitutions of the Mastoparan scaffold. In contrast, the discovered peptides originated from $PDB_{large}$—a diverse dataset of naturally occurring helices with broader distributions of charge, molecular weight, and hydrophobicity, explaining the divergence of the physicochemical properties of the discovered peptides.

The peptides were also enriched in helix-promoting residues (alanine, leucine), supporting the formation of helical structures, which could increase their structural stability in the membrane environment. The high aliphatic index indicated a predominance of aliphatic residues, contributing to the thermostability of the helical structures in hydrophobic environments, such as cell membranes. While aromatic residues were less abundant, they played a crucial role in stabilizing peptide interactions through π-π and cation-π interactions, particularly near membrane interfaces.

Through this rigorous filtering and analysis process, the size of the peptide library was reduced while ensuring that the selected candidates contained diverse peptides with favorable physicochemical properties for membrane interactions.

### Discovered peptides exhibited toxicity toward MDA-MB-231 breast cancer cells

The 105 peptides were SPOT-synthesized on cellulose membranes and were obtained as cleaved free peptides associated with the cellulose membrane. Peptide stock solutions were prepared by dissolving each peptide in water, and their anti-cancer effects were assessed by determining the cell viability of MDA-MB-231 breast cancer cells in the presence of the peptide at 25 μM. Among the 105 peptides tested, fifteen reduced cancer cell viability by >60% at this peptide concentration and thus were selected for further investigation (Supplementary Fig 21).

### Discovered peptides selectively discriminated MDA-MB-231 cells from healthy human blood cells

The 15 candidate peptides were synthesized to 95% purity, and their activity was then assessed in vitro to identify those selectively inhibiting breast cancer cell viability while sparing normal cells. These peptides were tested across a range of concentrations for their ability to inhibit the viability of MDA-MB-231 cells, RBCs, and PBMCs by half ($IC_{50}$). Normal human RBCs and PBMCs were used in viability assays to model non-cancerous cells, consistent with prior studies[72–74]. Peptides were classified as active if their $IC_{50}$ values against MDA-MB-231 cells were at least 5-fold lower than their $IC_{50}$ values against PBMCs or RBCs.

Seven of the 15 peptides exhibited $IC_{50}$ values of less than 30 μM in viability assays against MDA-MB-231 cells. In comparison, these seven peptides exhibited higher $IC_{50}$ values in viability assays against RBCs and PBMCs, indicating a higher level of toxicity and specificity towards cancer cells (Fig. 7). Notably, peptides 11 (LLQWLLKRLKAK), 29 (IILKKLLDFILK), and 57 (TLLTAIVKLFLK) displayed $IC_{50}$ values of $31.9 \pm 2.0$ μM, $14.3 \pm 1.5$ μM, $15.9 \pm 2.0$ μM respectively, for MDA-MB-231 cells, while their $IC_{50}$ values for non-cancerous cells occurred at concentrations at least 5.5-fold higher. These results suggested that peptides 11, 29, and 57 selectively inhibited MDA-MB-231 breast cancer cells, validating their identification as MDA-MB-231/PBMC-discriminating helical peptides.

Additionally, the toxicity of peptides 11, 29, and 57, as well as the positive control Mastoparan, toward additional breast cell lines was evaluated to further characterize the specificity of these peptides toward other cancer cells. Additional cell lines tested included MCF10A, a normal breast epithelial cell line; MCF7, an estrogen-responsive breast cancer cell line; and TamR3, a MCF7-derived tamoxifen-resistant cancer cell line. Interestingly, all four candidate peptides and Mastoparan reduced the viability of all four breast cell lines comparably, though in some instances the cytotoxic activity of peptide 11 was more potent than that observed for Mastoparan (Fig. 8). These findings highlighted a fundamental challenge in helical, membranolytic ACP discovery: achieving selective toxicity toward cancer cells while minimizing off-target effects on normal tissues.

Notably, the observed lack of selectivity among the discovered peptides, as well as Mastoparan, which exhibited cytotoxicity against multiple breast cancer cell lines (MCF7, MDA-MB-231, and TamR3), as well as against the normal breast epithelial cell line MCF10A, suggested that membrane-targeting mechanisms may not sufficiently discriminate between malignant and non-malignant breast cells. However, these peptides remained selective toward normal blood cells (PBMCs and RBCs). This pattern aligned with findings from a recent study by Yue et al.[36], which underscored the difficulty in identifying ACPs that exhibit broad cancer selectivity without explicitly incorporating diverse cancer and normal cell types into model training.

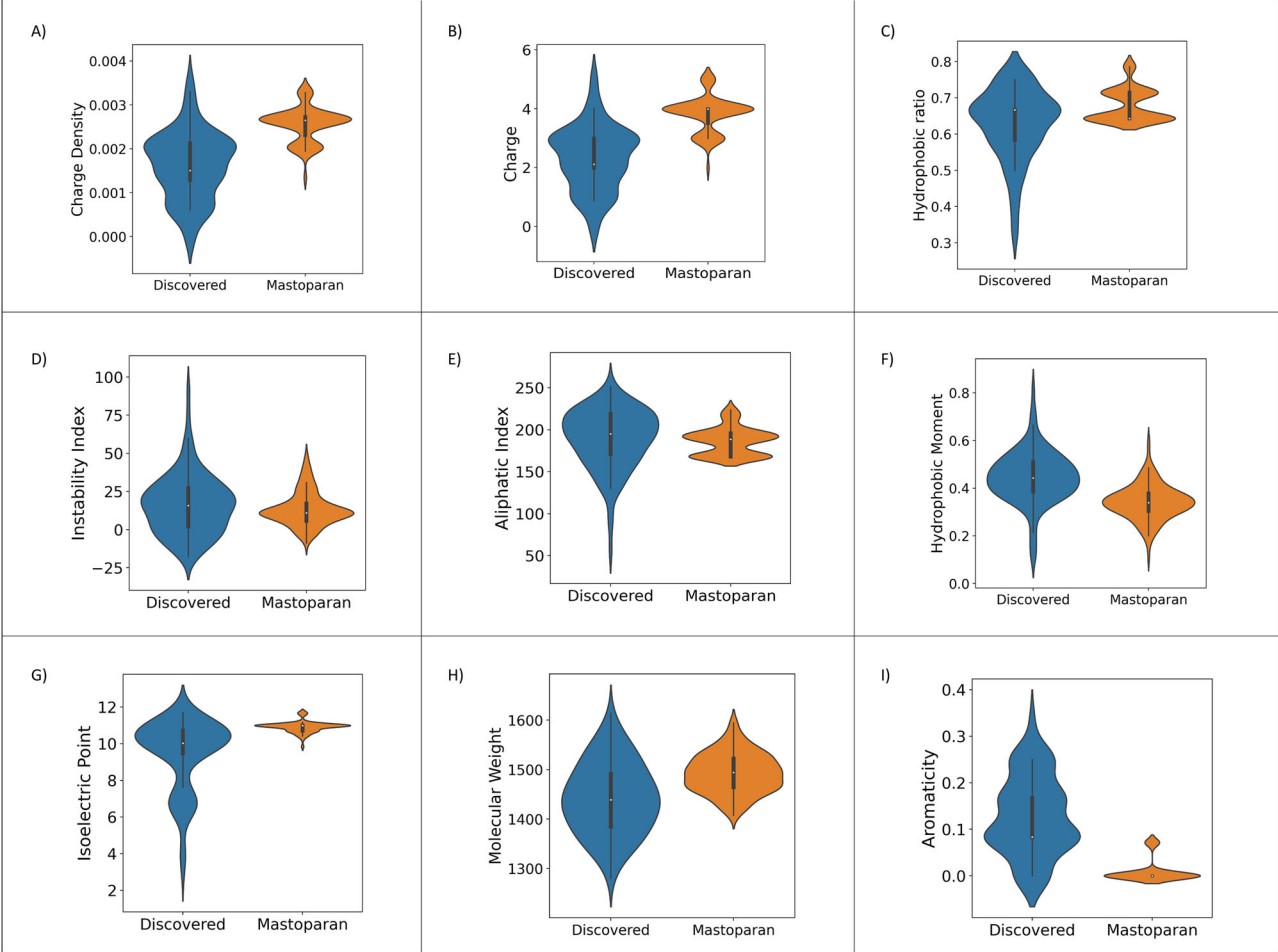

**Fig. 6 | Physicochemical properties of top experimentally validated candidates.** Comparative analysis of physicochemical properties between 105 discovered peptides and training sequences ($D_{Mastoparan}$), highlighting their dissimilarity in key anti-cancer activity factors. The observed differences in mean values reflect the fundamental differences in dataset composition: the Mastoparan dataset is constrained to a single scaffold with limited physicochemical variability, while the discovered peptides originate from PDB$_{large}$, a diverse dataset of helical peptides. Furthermore, Target Adaptive Reinforcement Learning for Sampling Activity Landscapes (TARSA)'s reinforcement learning-driven selection process optimizes functional activity rather than enforcing structural similarity to Mastoparan, leading to broader physicochemical distributions. The embedded boxplots within the violin plot indicate the median (center line), 25th and 75th percentiles (box bounds), and whiskers extend to data points within 1.5× the interquartile range. **A** Charge Density, **B** Charge, **C** Hydrophobic Ratio, **D** Instability Index, **E** Aliphatic Index, **F** Hydrophobic Moment, **G** Isoelectric Point, **H** Molecular Weight, **I** Aromaticity.

The oracle proxy model, trained exclusively on MDA-MB-231 and PBMC data, was able to generalize its predictions to other cancer cell lines (MCF7 and TamR3) and normal blood cells (RBCs) but failed to accurately generalize toxicity against MCF10A. This limitation highlighted the need to incorporate negative control (non-cancerous) cell-line-specific data in training to improve model robustness. Moving forward, the TARSA-screened and experimentally validated data from this study could be leveraged to retrain the oracle proxy model for better cell-type specificity, including for MCF10A. Alternatively, rational peptide engineering approaches may be explored to enhance the broad-spectrum selectivity of discovered ACPs.

The off-target cytotoxicity of peptides against healthy breast cell lines (e.g., MCF10A) could likely be attributed to the similar membrane composition of normal and cancerous breast cells. For detailed lipid composition analysis and peptide specificity mechanisms, see Supplementary Note 8.

**All-atom molecular dynamics**
The CG-MD findings were further extended with all-atom molecular dynamics (AA-MD) simulations performed on three peptide candidates identified by the PepSce workflow: IILKKLLLDFILK, TLLTAIVKLFLK, and LLQWLLKRLKAK (Fig. 9A–C). Initially positioned near the membrane surface (7 Å), these peptides began dynamically interacting with the membrane as the simulation progressed, eventually stabilizing the system. Around 300 ns into the simulation, all three peptides exhibited measurable increases in peptide-membrane contacts, signaling deeper insertion or tighter association. Peptides 29 (IILKKLLLDFILK) and 57 (TLLTAIVKLFLK) both showed lower contact counts (~400–800) at the outset (50 ns), which then rose steadily, surpassing 2000 contacts by 750 ns, indicative of significant insertion events within the bilayer. Peptide 11 LLQWLLKRLKAK, in contrast, displayed a relatively high contact count (over 2300) at the outset (50 ns) and continued to fluctuate between 2000 and 2800 contacts up to 450 ns, suggesting an earlier, more extensive engagement with the membrane compared to peptides 29 (IILKKLLLDFILK) and 57 (TLLTAIVKLFLK). Across all three replicas, the observed variations in contact counts imply transient peptide reorientation, partial withdrawal, or dynamic lateral diffusion within the bilayer. However, the overall contact profile points to robust membrane engagement that may lead to local disturbances or bilayer destabilization, consistent with a lytic mechanism. Taken together, these contact analyses reinforce the ability of these peptide candidates to insert and perturb the

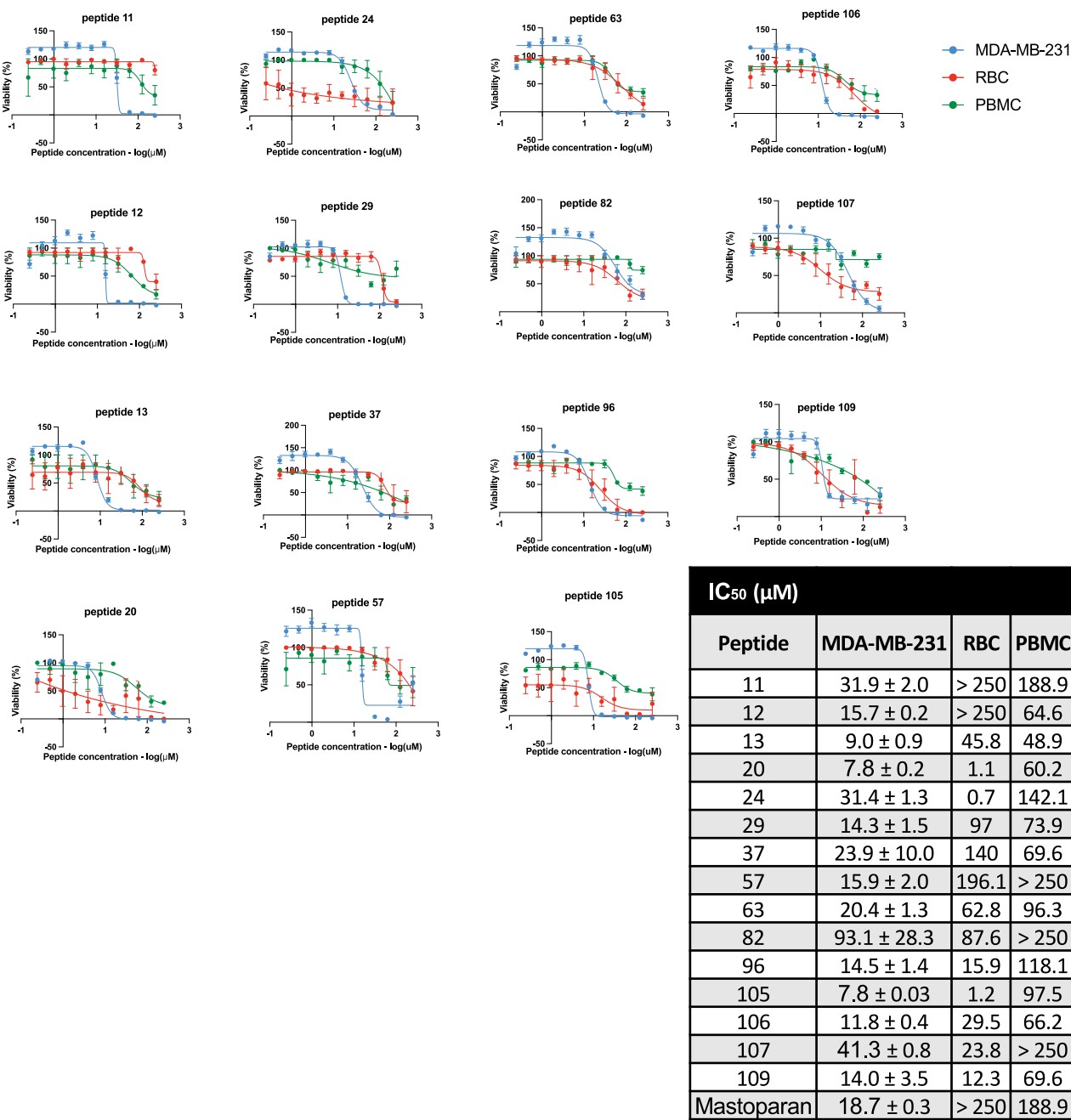

**Fig. 7 | Dose response inhibition of viability in MDA-MB-231, RBC and PBMCs.** IC$_{50}$ of lead peptides in MDA-MB-231, RBC and PBMCs was measured by PrestoBlue assay for viability. The dose–response curve for Mastoparan is shown for comparison. IC$_{50}$ values were calculated by curve fitting the dose–response curves in Graphpad Prism software (v 10.2.1). The table shows the comparative IC50 values for these peptides in the three cell lines. The data is from two biological replicates, each with four technical replicates. The error bars represent standard error of the mean. Eleven out of 95 considered Peptides (ID < 100) originated from PDB$_{12mer}$, while remaining four out of 10 considered peptides originated from GEN$_{12mer}$.

| IC$_{50}$ (μM) | | | |
| --- | --- | --- | --- |
| Peptide | MDA-MB-231 | RBC | PBMC |
| 11 | 31.9 ± 2.0 | > 250 | 188.9 |
| 12 | 15.7 ± 0.2 | > 250 | 64.6 |
| 13 | 9.0 ± 0.9 | 45.8 | 48.9 |
| 20 | 7.8 ± 0.2 | 1.1 | 60.2 |
| 24 | 31.4 ± 1.3 | 0.7 | 142.1 |
| 29 | 14.3 ± 1.5 | 97 | 73.9 |
| 37 | 23.9 ± 10.0 | 140 | 69.6 |
| 57 | 15.9 ± 2.0 | 196.1 | > 250 |
| 63 | 20.4 ± 1.3 | 62.8 | 96.3 |
| 82 | 93.1 ± 28.3 | 87.6 | > 250 |
| 96 | 14.5 ± 1.4 | 15.9 | 118.1 |
| 105 | 7.8 ± 0.03 | 1.2 | 97.5 |
| 106 | 11.8 ± 0.4 | 29.5 | 66.2 |
| 107 | 41.3 ± 0.8 | 23.8 | > 250 |
| 109 | 14.0 ± 3.5 | 12.3 | 69.6 |
| Mastoparan | 18.7 ± 0.3 | > 250 | 188.9 |

membrane environment, a possible route to anticancer activity (Fig. 9D). AA-MD thus provides insights into a potential mechanism of action for discovered MDA-MB-231 inhibitors.

## Discussion

The goal of DL-based peptide screening is to streamline the computational selection of promising candidates from vast pools. Thus, the need for devising accurate and efficient strategies for performing global searches in large peptide search space is urgent. To address this problem, an RL algorithm TARSA was introduced herein for significant search space reduction. The scalability of TARSA was demonstrated

through a fast and cost-effective virtual screening of ultra-large peptide libraries for helical, membranolytic Mastoparan-like peptides with demonstrated activity towards MDA-MB231.

Despite advancements in DL, there remains a scarcity of DL-driven studies for bioactive peptide discovery that are complete with experimental validation[75,76]. This gap may be attributed to data insufficiency leading to poor generalization, computational intractability of the search space, and the high costs associated with peptide synthesis and experimental validation (both for positive and negative samples).

The proposed method aimed to address all three problems. By injection of priors through the choice of physiochemical descriptors,

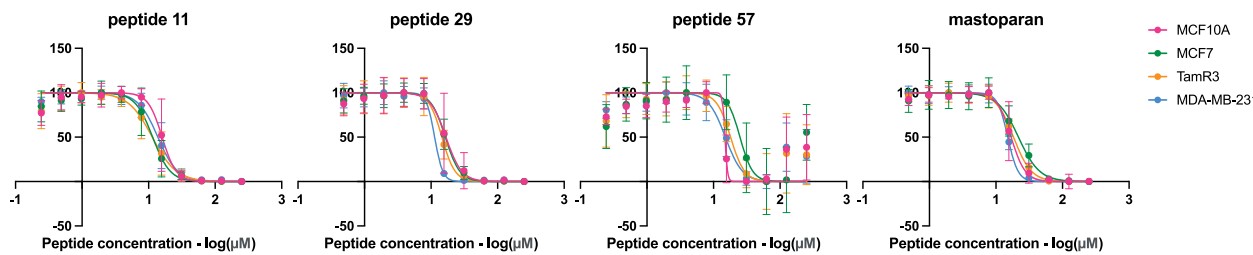

| IC$_{50}$ (μM) | | | | |
|---|---|---|---|---|
| Peptide | MCF10A | MCF7 | TamR3 | MDA-MB-231 |
| 11 | 15.99 | 11.51 | 11.37 | 13.80 |
| 29 | 16.76 | 16.39 | 14.57 | 11.33 |
| 57 | 14.92 | 25.01 | 18.05 | 15.10 |
| Mastoparan | 16.91 | 21.79 | 18.39 | 15.06 |

**Fig. 8 | IC50 values of the top 3 peptides and Mastoparan were determined in MCF10A, MCF7, TamR3, and MDA-MB-231 cells.** The cells were treated with selected peptides at serial dilutions ranging from 250 μM to 0.244 μM in RPMI media supplemented with 2.5% FBS. The data is from three biological replicates, each with three technical replicates. The error bars represent the standard error of the mean.

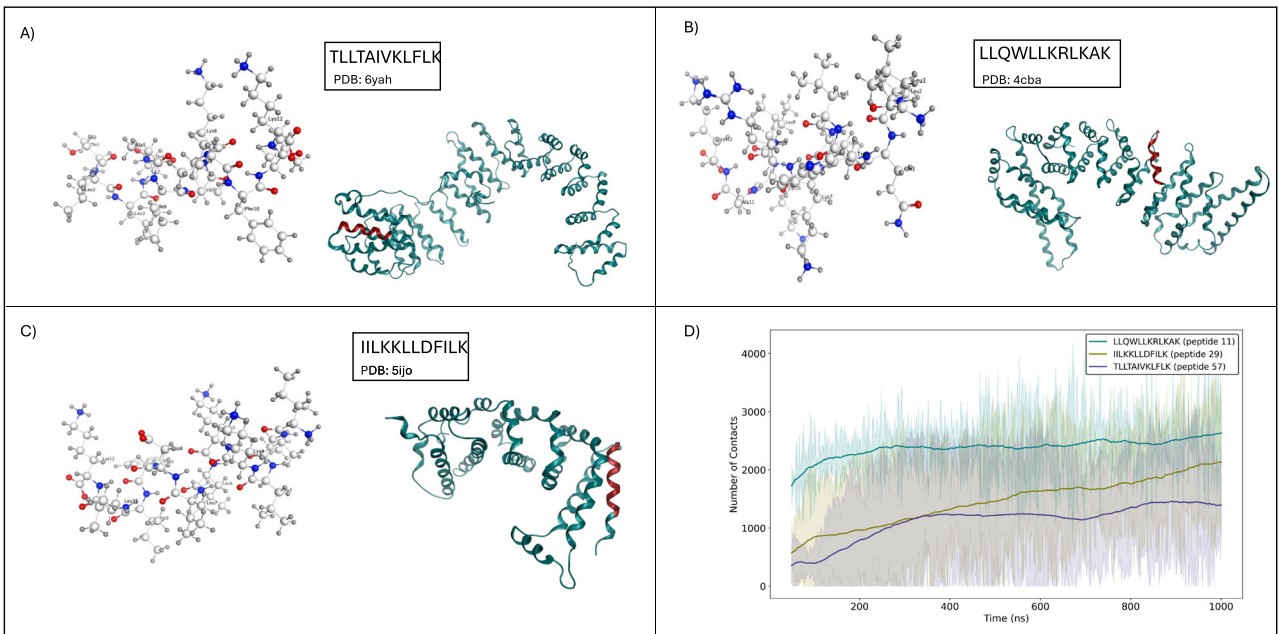

**Fig. 9 | Molecular dynamics (MD) and structural analysis of selective anti cancer peptides. A–C** Ball-stick renderings and corresponding PDB structures of the three most MDA-MB-231/peripheral blood mononuclear cells (PBMC) discriminating peptides—11 (LLQWLLKRLKAK), 29 (IILKKLLDFILK), and 57 (TLLTAIVKLFLK). **D** Time evolution of the average number of contacts between the three peptides and a model membrane during a 1 μs All-Atom MD simulation; curves show the mean across three independent simulation replicates per peptide (n = 3) with shaded error bands indicating the standard deviation across replicates.

model generalizability was improved despite using a small-sized dataset. The training set used was biased towards a set of Mastoparan derivatives and we acknowledge the inherent potential bias. However, this dataset was uniquely valuable due to its continuously-valued % cell inhibition measurements obtained under standardized conditions with particular cancer cells, in contrast to literature bioactive ACP datasets that generally use non-standardized and/or binary classification results with varied cell lines. To further mitigate overfitting and improve

generalization, we used an ensemble approach to incorporate external ACP datasets with greater sequence diversity.

Additionally, TARSA balanced the exploration-exploitation tradeoff by selectively sampling peptides from bioactivity cliffs, significantly reducing the search space and computation time. This was exemplified by the virtual screening of all helices and 12-mers in the PDB within three weeks, leading to the discovery of three lead peptides with selective toxicity toward TNBC cells. Finally, high-throughput

spot-array peptide synthesis enabled the creation and preliminary validation of over 100 peptides, nearly five times more than in prior studies[16]. SPOT-array synthesis also made it cost-effective to validate less effective peptides, which are crucial as negative examples for training subsequent DL models for iterative optimization with experimental feedback. Thus, future work could involve training surrogate models for to enhance breast cancer specificity, which could expand the known membranolytic ACP space and improve subsequent DGMs. It is notable that DGMs have emerged as powerful tools for exploring large sample spaces by training neural networks to approximate peptide distributions. This capability allow DGMs to explore extensive chemical spaces and generate compositionally novel peptide candidates. However, the lack of large, reliable, and diverse training datasets can make membranolytic peptide DGMs susceptible to overfitting, restricting them to sampling peptides from a narrow distribution[77], limiting translational progress in DGM-driven peptide discovery. Additionally, while DGMs excel in broad exploration, they often do not leverage decades of accumulated knowledge on natural and synthetic peptide sequences[78,79], a gap that virtual screening methods like TARSA can address. Furthermore, TARSA could also be adapted for the discovery of other therapeutic modalities (e.g., small molecule therapies) in the future.

While practical and scalable, TARSA presented certain limitations. Although 2D-PCA mitigated the sparsity problem associated with higher dimensions, it was prone to overlapping points (i.e., peptides can overlap or cluster closely together), making them difficult to distinguish. Reducing the ESM embedding to 2D resulted in a significant loss of information and variance, potentially obscuring critical features and relationships present in the original high-dimensional space. The quality of initial peptide representation (i.e., ESM) also heavily influenced TARSA's outcome; if the embedding failed to adequately capture the essential biological features, the PCA results were correspondingly limited (Supplementary Note 3). These factors may partly explain the lower selectivity of the discovered candidates on MCF10-A cells compared to other breast cancer cell lines. Furthermore, the identified peptides with low micromolar IC50 represent the first round of candidates and despite the limitations of using a primary training set with restricted variance it is clear that our ensemble approach generated predicted peptides that varied substantially from the training peptides. Further iterative studies would increase diversity and selectivity. Indeed the use of larger more diverse datasets to predict improved ACP has suffered from the lack of commonality of methods and cancer cell line used for the training set peptides.

It is also noteworthy that the most advanced ACPs in clinical development are predominantly non-helical and non-membranolytic, often acting through mechanisms distinct from direct membrane disruption. In contrast, the TARSA-identified peptides presented here represent Mastoparan-like, membranolytic helical scaffolds that achieve selective cytotoxicity against MDA-MB-231 cells relative to PBMCs. While this provides a clear proof-of-concept for TARSA in recovering biologically active peptides from large-scale screening, subsequent studies should also leverage TARSA to explore libraries enriched in non-helical and non-membranolytic peptides, thereby aligning more closely with the structural and mechanistic hallmarks of clinically advanced ACPs.

The screening approach described here leveraged a priori knowledge from known peptides to assist with in vitro validation and has the potential to advance and accelerate automated AI-driven drug discovery. Application of this approach will allow others to identify functional peptides from rich candidate libraries and offer an elegant solution to complex bioactive peptide discovery problems.

In this work, we developed a scalable deep RL-based approach for screening ultra-large peptide libraries. Through the integration of RL and MCMC sampling, the developed TARSA approach enabled efficient exploration of 36-million-peptide library and identified regions of the chemical space enriched in Mastoparan-like membranolytic peptides. This study represents one of the largest virtual screening efforts for bioactive peptides reported to date. The use of TARSA resulted in the identification and cost-effective in vitro validation of 105 peptide candidates, of which 15 demonstrated cytotoxic activity against breast cancer cells. These results characterize TARSA as a promising method for scalable and efficient discovery of biologically active alpha-helical peptides.

## Methods

### Ethics statement

To obtain cells used in the peptide toxicity assays, venous blood was collected in sodium heparin tubes (BD Biosciences) from three healthy volunteers in accordance with protocols approved by the University of British Columbia's Research Ethics Board (protocol number H21-01910). All participants gave their written informed consent prior to participation, and they received no financial compensation for participating in the study. Sex, gender and age were not considered as variables in this study and this data was not collected from study participants.

### TARSA: reinforcement Learning for goal-directed peptide discovery

TARSA operates on a navigational board constructed using 2D-PCA projections of peptides. The algorithm uses RL to guide an agent toward regions (hotspots) with higher predicted cell inhibition (Fig. 2A, B). This section describes the methods used to design the environment, define the agent's states and actions, and train the RL policy for peptide discovery.

**Low-dimensional peptide representation.** Peptides were represented in a low-dimensional space to facilitate efficient RL policy learning. Using representations derived from ESM of protein sequences, peptide embeddings were first computed (Supplementary Note 3). These embeddings, originally high-dimensional (768 dimensions), were reduced to a 2D space using PCA. This reduction aimed to improve computational efficiency and reduce the sparsity associated with searching a high-dimensional peptide landscape. The 2D-PCA projections served as the state space on the navigational board, where the agent explored and screened peptides.

**Reinforcement learning setup.** The peptide discovery process was modeled as a Markov Decision Process (MDP) consisting of states (S), actions (A), transition probabilities (P), rewards (R), and a discount factor ($\gamma$). The state space (S) corresponded to the 2D-PCA coordinates of peptides on the navigational board, while the action space (A) consisted of movement parameters, including the step size and direction. The transition dynamics were deterministic, meaning that the agent's movement in the environment strictly followed the action taken.

**Environment design.** The environment was represented as a 2D navigation board, where peptides were positioned according to their PCA projections. The agent was tasked with navigating this space, moving from an initial random position to a target hotspot ($h_{target}$), a dynamically updated location that represented a region with highly potent peptides. The board could present one of four possibilities at any location: (i) blank, where no peptide was present; (ii) peptide, where a peptide was located; (iii) goal, when the agent reached the target hotspot; and (iv) revisit, when the agent returned to a previously explored location. Rewards were assigned based on these outcomes, encouraging the agent to explore new areas and discover potent peptides. An episode began with the agent at a random location on the board, and it could take actions to move based on its policy. The episode could end under three conditions: (i) the agent moved out of bounds of navigation board, (ii) a more potent peptide was discovered ($h_{target}$ updated), or (iii) the agent reached 1000 steps.

**Agent's states and actions.** At each timestep ($t$), the agent's state was represented by its location on the board, the Euclidean distance to the target hotspot ($h_{\text{target}}$), and the coordinates of the hotspot. The agent's action space included two components: step size ($\lambda_t$), constrained between 0.95 and 1.05, and direction ($\theta_t$), constrained between $-\pi/3$ and $\pi/3$ radians. The state transitions were deterministic, with the agent's new location calculated based on the rotation matrix for the given action. This ensured precise control over the agent's movements.

**Reward function.** To drive the agent's exploration and ensure efficient peptide screening, a reward function ($r_t$) was designed that incorporated three components: distance-based reward ($r_{\text{dist}}$), oracle-based reward ($r_{\text{oracle}}$), and penalty ($r_{\text{penalty}}$). The distance-based reward promoted convergence towards the target hotspot by rewarding the agent as it moved closer to the target. This was calculated as a function of the Euclidean distance between the agent's current position and the target hotspot. The oracle-based reward encouraged exploration of regions with potent peptides, using predictions from an external oracle model that estimated peptide cell inhibition towards MDA-MB231 cells. The penalty term was included to discourage long and inefficient paths by penalizing the agent for taking unnecessary steps. These reward components were weighted by hyperparameters ($\beta_i$) to balance exploration and exploitation.

**Policy training and hotspot sampling.** To ensure the agent explored all promising regions (activity cliffs), the goal ($h_{\text{target}}$) was updated dynamically using posterior sampling. The agent's rewards when it reached a candidate hotspot ($h_i$) were modeled using a Gaussian distribution, and posterior distributions for the mean ($\mu_i$) and variance ($\sigma_i$) of the rewards were estimated using MCMC sampling. The target hotspot was selected by sampling from this posterior distribution, allowing the agent to focus on regions with higher predicted cell inhibition. Policy optimization was performed using the Proximal Policy Optimization[53] (PPO) algorithm, which updated the agent's policy based on an advantage function that incorporated the sampled $h_{\text{target}}$. This advantage function is calculated as the difference between the action-value function ($Q_\psi$) and the value function ($V_\psi$), both parameterized by architecturally similar MLPs. Incorporation of $h_{\text{target}}$ to policy optimization objective $J(\phi)$, like a goal-conditioned RL setting, was used. Here, the agent's goals ($h_{\text{target}}$) were provided explicitly. This allowed the agent to focus on efficiently exploring the peptide landscape without the need to discover the goals themselves. The advantage function was further modified to include the composite reward structure, ensuring that the agent optimized for both exploration of the chemical space and discovery of potent peptides. Further exploration was encouraged using intrinsic motivation. Intrinsic motivation, akin to curiosity, played a crucial role in guiding the agent through vast peptide landscapes. The $r_{\text{oracle}}$ reward component acted as an intrinsic reward, encouraging the agent to explore novel regions and avoid local optima. Together, these components enabled TARSA to efficiently screen large peptide libraries, identify MDA-MB231 inhibitory peptides, and adapt dynamically to changing goals during exploration. The overall objective for policy optimization was given as

$$J(\phi) = \mathbb{E}_{s, a \sim \pi_\phi}\left[\mathcal{L}^{CLIP}(\phi) - c.\left[H(\pi_\phi(.|s))\right]\right] \quad (1)$$

where, $\mathcal{L}^{CLIP}$ is PPO's surrogate objective and $H(.)$ is policy entropy term encouraging exploration to novel states. A more formal treatment of our method with implementation details of TARSA is available in Supplementary Method 1 and Supplementary Note 5.

**Inference phase using policy transfer.** Once the agent has learned a policy $\pi$ on a subset of the peptide library, $\boldsymbol{S}$, rollouts from $\pi$ are

obtained to screen the remainder of the peptide library $D - \{\boldsymbol{S}\}$ (Fig. 1B). The agent is initialized at random points on the PCA-projected space of $D - \{\boldsymbol{S}\}$. For each trajectory, the target hotspot $h_{target}$ is selected as $h_{target} = \text{argmax}_i(X_1, X_2, \ldots, X_n)$, where $X_i \sim N(\mu_i, \sigma_i) \forall i \in \boldsymbol{H}_c$, $\boldsymbol{H}_c$ is a set of candidate target hotspots, and $\mu_i, \sigma_i$ are parameters learned during training. Posterior sampling was not performed during inference. During inference, agent collects peptides based on their predicted cell inhibition, stopping when average potency of collected set drops below 40% inhibition or other termination criteria are met (Supplementary Note 6). For GEN$_{12mer}$, the trained policy was rolled out on 14 disjoint batches of 2 million samples each, discovering the top 7.5% peptides (-150,000), with a mean cell inhibition of 45.2%. The screening process, parallelized across 14 Tesla V100 GPUs, completed in five days.

**Metrics.** The quality of the sampler is evaluated using mean predicted cell inhibition, novelty, diversity, and hit rate:

$$Mean(\boldsymbol{S}) = \frac{\sum_{i=1}^{|\boldsymbol{S}|} f^\theta(s_i)}{|\boldsymbol{S}|} \quad (2)$$

$$Novelty = \frac{\sum_{i=1}^{|\boldsymbol{S}|} Lev(s_{mastoparn}, s_i)}{|\boldsymbol{S}|} \quad (3)$$

$$Diversity = \frac{\sum_{i=1}^{|\boldsymbol{S}|} \sum_{j=1, j\neq i}^{|\boldsymbol{S}|} Lev\left(s_i, s_j\right)}{|\boldsymbol{S}|(|\boldsymbol{S}| - 1)} \quad (4)$$

$$Hit\ Rate = \frac{hits_A}{|\boldsymbol{N}|} \times 100 \quad (5)$$

where, $s_i, s_j \in \boldsymbol{S}$ and $\boldsymbol{S}$ is the sampled set of candidate peptides, $f^\theta(.)$ is the oracle proxy. $Lev(.)$ is the Levenshtein distance. $hits_A$ are the number of discovered/generated peptides with predicted % cell inhibition >50% and $|N| = 50,000$.

## Wet-laboratory characterization

**Spot array.** Peptide arrays on cellulose membranes[55] were made by Kinexus Bioinformatics Corporation (Vancouver, BC, Canada) and were obtained as cleaved free peptides associated with the punched-out spot of the cellulose membrane. Peptide stock solutions were prepared by soaking the peptide-loaded cellulose membrane in 200 µl of dH$_2$O for -2 h and then the resulting peptide solution was transferred to a fresh microfuge tube for use in subsequent in vitro toxicity tests. Peptide concentrations were measured by using nanodrop to calculate the concentration of aromatic containing peptides and then dilute them to 250 µM. Peptides for which the concentration couldn't be measured, the average value from the array was used. The concentrations were adjusted to a starting concentration of 250 µM.

**ACP preparation.** In vitro toxicity tests were performed to validate the selective cytotoxic activity of candidate peptides. Since the oracle proxy model was trained on Mastoparan-NH$_2$ (INLKALAALAKKIL-NH$_2$), which has a free N-terminus and an amidated C-terminus, all synthesized peptides for experimental validation retained this amidation to ensure consistency between computational predictions and biological testing. Highly pure synthetic peptides were purchased from Peptide 2.0 Inc. (Virginia, USA). The peptides were synthesized by standard solid-phase methods, and the purities ($\geq$95%) and molecular weights were determined by high-pressure liquid chromatography (HPLC) and mass spectrometry, respectively. Peptide stock solutions were prepared by dissolving lyophilized peptide powder in endotoxin-free water according to their molecular weight and diluted to 10× the final concentration in endotoxin-free water.

**Cell culture.** MCF10A, MCF7 and MDA-MB-231 cell lines were obtained from the American Type Culture Collection (ATCC). TamR3 cells were gifted from Euphemia Leung (University of Auckland, New Zealand). MCF10A, MCF7 and MDA-MB-231 were cultured in RPMI 1640 media containing 10% fetal bovine serum (FBS; Invitrogen Life Technologies) and TamR3 in RPMI 1640 media containing 10% charcoal-stripped serum (CSS; Invitrogen Life Technologies) and 1 μM Tamoxifen (Sigma-Aldrich). The cells were cultured at 37 °C in a humidified incubator with 5% $CO_2$.

**Viability assay in breast cells.** The cells were seeded at 5000 cells per well in 96-well Black clear bottom plates (Fisher Scientific 12-566-70) in RPMI 1640 media supplemented with 10% FBS. 24 h later, the media was removed and exchanged with 90 μl RPMI 1640 media supplemented with 2.5% FBS. The negative and positive control cells were then treated with either 10 μl $dH_2O$ or 1% NP-40. The rest of cells were then treated with 10 μl of each peptide at final concentration of 25 μM for the initial screening or at serial dilutions from 250 μM to 0.244 μM $IC_{50}$ measurements. The cells were then incubated at 37 °C for 24 h. Then 10 μl of PrestoBlue Cell Viability Reagent (Thermo Fisher A13262) was added to each well and incubated at 37 °C for 1 h. Fluorescence intensity was measured using the TECAN M200Pro plate reader with emission and excitation wavelengths of 535 nm and 612 nm, respectively. Results were normalized with $dH_2O$ set as 100% viability and 1% NP40 as 0% viability. In our study, biological replicates refer to independent experiments performed on different days with freshly prepared cell cultures, while technical replicates refer to repeated measurements within the same experiment (i.e., parallel wells in the same plate).

**Normal cell isolation from whole blood.** Whole venous blood from three healthy, consenting volunteers was collected in BD vacutainer tubes with sodium heparin as an anticoagulant (BD Biosciences, Mississauga, ON, Canada), using protocols approved by the University of British Columbia Research Ethics Board. Whole blood was centrifuged ($280 \times g$, 5 min) and washed three times with phosphate-buffered saline (PBS) to isolate red blood cells (RBCs). The cell suspension was diluted with PBS to a hematocrit of 5% RBCs, then dispensed into 96-well flat-bottom tissue culture plates. Peripheral blood mononuclear cells (PBMCs) were isolated by diluting whole blood in PBS (1:1), then carefully layering the suspension onto Lymphoprep density gradient medium (STEMCELL Technologies, Vancouver, BC, Canada). The cells were centrifuged ($1200 \times g$, 12 min) and the buffy coat containing the PBMCs was collected. The PBMCs were then washed twice with PBS and resuspended in RPMI-1640 media supplemented with 5% fetal bovine serum (FBS). The cells were seeded in 96-well flat-bottom tissue culture plates and were incubated (37 °C, 5% $CO_2$) 1 h prior to cytotoxicity assays.

**Cytotoxicity and hemolysis assays for PBMCs and RBCs.** Peptide treatments (10× final concentrations) were added to RBCs (final hematocrit 2.5%), then plates were incubated (37 °C, 5% CO2) for 4 h, as previously described[80]. Endotoxin-free water and PBS were used as negative controls, and 1% Triton X-100 was used as a positive control. After incubation, the plates were centrifuged (1400 g, 5 min) and the supernatant was transferred to a new 96-well flat-bottom plate. The absorbance at 490 nm was recorded for each sample using an Epoch microplate spectrophotometer (Bio-Tek Instruments Inc. Winooski, VE, USA) to measure the amount of hemoglobin present in the supernatant. The half maximal inhibitory concentration (IC50) of peptide was calculated in GraphPad Prism (v 10.2.1) using nonlinear least squares regression analysis of the normalized dose response curves for each peptide. After resting PBMCs for 1 h, peptide treatments (10x final concentrations) were added to the plate (final PBMC density of $1 \times 10^5$ cells/ml). Treated PBMCs were incubated overnight (37 °C, 5% $CO_2$) and the supernatants were collected in fresh 96-well

plates following centrifugation (1150 rpm, 5 min). Sample supernatants were used immediately in the lactate dehydrogenase assay (Roche Diagnostics, Basel, Switzerland) following the manufacturer's instructions. Endotoxin-free water and PBS were used as negative controls, and 1% Triton X-100 was used as a positive control. The IC50 was calculated as described above. All hemolysis and PBMC cytotoxicity assays were carried out using cells isolated from three separate donors and each biological replicate was performed in technical duplicate.

**Reporting summary**
Further information on research design is available in the Nature Portfolio Reporting Summary linked to this article.

## Data availability
The inhouse datasets–Mastorparan MDA-MB231 and Mastoparan PBMC, and the experimental validation results are available as supplementary information under Supplementary Notes 1–7. The screening dataset from the PDB is available from https://www.rcsb.org/. Source data are provided with this paper as a Source Data file. Source data are provided with this paper.

## Code availability
The code used to develop the model/perform the analyses and generate results in this study is publicly available and has been deposited at https://github.com/diamondspark/PepSce.git, under MIT license. The specific version of the code associated with this publication is archived in Zenodo and is accessible via https://doi.org/10.5281/zenodo.17231194.

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

## Acknowledgements

This research was enabled by funds provided by a Canadian Institutes of Health Research (CIHR) doctoral award (FRN: FBD-187593) to M.P. and FDN-154287 to R.E.W.H. A.C. and M.E. would like to acknowledge the support from their respective NSERC Discovery grants. The authors would like to thank Dr. Ashley Hilchie for her contributions towards generating Mastoparan datasets.

## Author contributions

M.P., R.E.W.H. and A.C. contributed to the original idea. M.P. devised, coded and ran experiments for TARSA, PepSce and the benchmarks. S.M., M.S., J.F., M.S. and N.L. conducted experimental assays for peptide validation on breast cancer cells. M.A.A. and E.F.H. performed assays on PBMCs and RBCs. H.M., F.G. performed molecular dynamics experiments. G.S. contributed to the writing methods section for TARSA. M.P. wrote the majority of the first manuscript draft with inputs from all coauthors on their respective sections. N.L., E.F.H., R.E.W.H., M.E. and A.C. provided supervision, and all authors contributed to proofreading and editing the manuscript.

## Competing interests

The authors declare no competing interests.
