## [Transparent Peer Review file · Nature Communications]

A Scalable Reinforcement Learning Approach for Screening Large Peptide Libraries for Bioactive Peptide Discovery

Corresponding Author: Professor Artem Cherkasov

Version 0:

Reviewer comments:

Reviewer #1

(Remarks to the Author)

The manuscript describes the development of an ML-based virtual screening pipeline for anti-cancer helical peptides, aiming to achieve high performance for giga-scale libraries. While the topic is undoubtedly valuable, the current form of the manuscript makes it challenging to evaluate the true performance of the application.

Major Concerns:

Training Set Bias: I am particularly concerned about the potential bias due to the small initial training set size (210 Mastoparan derivatives). This limitation means the resulting application is highly influenced by Mastoparan SAR, and one must also consider the high cytotoxicity of many of the compounds.

Validation of MD Simulations: The validation of MD simulations is another significant issue. Although the GC-MD is performed with 5 replicas, it is not clearly validated whether the selected simulation time is sufficient for the task. This is critical since outliers are screened out. It is necessary to test if longer MD simulations are needed. For the final MD simulations, the timeframe is absolutely too short, and several replicas are required. In its current form, the simulation time is insufficient, as true equilibrium or even metastable states cannot be reached in a timeframe shorter than several tens of microseconds. Replicas are mandatory at this stage.

Manuscript Structure and Readability: The manuscript heavily emphasizes computational details, which is good, but it complicates following the manuscript. The overall scheme of the manuscript is unclear when one starts to read the results section, making it almost impossible to understand the validity of the results at this stage.

(Remarks on code availability)

Reviewer #2

(Remarks to the Author)

The authors proposed an algorithm focusing on activity cliffs within peptide libraries that can help prioritize the high-potential candidates. They proposed a computational model for screening peptides with anticancer peptides using reinforcement learning techniques and identified 3 potential anticancer peptides without toxicity through experimental validation procedure. Through report 15 out the top 100 peptides selected using this method showed potent activity against breast cancer cell, including drug-resistant triple-negative breast cancer through insilco and successive wet-lab experiments to validate the ACP potential of top 3 peptide identified using TARSA's method.

However, I have several comments which author should address to increase the scope and strength of manuscript.

1. In the result section, specially in motif discovery, it is mentioned TARSA was applied to PDBlarge to identify potent anticancer motifs, which were subsequently used to construct a library of fixed length 12-mer peptides. But how did the

author confirm that those discovered motifs have biological significances, how did you confirm that they are really helical ACPs motifs? Are there any statistical calculation to confirm the presence those motifs in training samples? And if these motifs are prevalent in helical peptides, what are the most preferential position of those helical motifs which can assist in design of anticancer peptides with higher potency?

2. Why did the author limit the exploration of bioactivity prediction assessment with random forest, XGBoost, and MultiLayer Perceptron? Is there any motivation in selecting only those models for prediction? If yes, can you clarify the reason?

3. In Figure 3, Though the discovered peptide has relatively low molecular weight which can ease the synthesis process, but why does the discovered peptides don't catch the regular distribution of anticancer peptides available in CancerPPD? Is this because of properties specific to helical ACPs? Please mention it clearly why discovered peptides have relatively shorter variance in case of charge density and hydrophobic moment? Are these a good indicator to explain helical ACPs properties? Please review and mention it clearly.

4. Authors mentioned about the active and inactive ACP peptides in manuscript specially Figure 2. But didn't mention about the threshold he applies for IC50 value to classify into experimentally active and inactive. Please check it and add details on how to create this dataset.

5. During Dataset Construction for training the TARSA model, the author selected 1.5M 12 Mers (PDB12mer) for dataset construction? Does it ensure that, this dataset include representative samples (natural variation in motifs presence) specially in Helical ACPs with helical motifs? How did the authors ensure that those helical motifs mostly lie within 12 Mers peptide candidates for training?

6. In Figure 6, relatively the charge density, charge, molecular weights are not symmetric to the Mastoparan training dataset. What is the major reason for these difference in mean? Is this because, the discovered peptides were designed from the comprehensive helical peptides which have likelihood of diverse physiochemical properties compared to Mastoparan? Please mention it in manuscript clearly.

(Remarks on code availability)

Reviewer #3

(Remarks to the Author)

The manuscript of Pandey et al. is a bioinformatics paper focussing on the application of machine learning (ML) to drug discovery. The authors suggest and demonstrate the implementation of an ML approach for (rational) sparse sampling of (ultra)large peptide libraries aiming to select a reduced number of potentially bioactive sequences for further validation. Computationally sound and, in principle, correct in the conclusions, the manuscript, in its current state, to my regret, is unsuitable for the broad audience of Nature Communications. It would be better received in a more specialized journal (bioinformatics, ML or peptide medicinal chemistry), but significant revision can be recommended even for an alternative submission.

The submitted manuscript appears as a compilation of two separate works (steaming from two manuscripts): one on sparse sampling/ML for drug discovery (a more substantial aspect) and another - on mastoparan optimization as an "anticancer peptide" (a weaker aspect). This origin explains the discrepancy in the nomenclature being used throughout the fused manuscript (i.e. datasets (peptide libraries) are named differently; the workflow ("PepSce"), the Oracle function also vary in description, etc.) and must be uniformized should authors revise the submission. The manuscript has to be restructured/rewritten; use of the terms should not occur before defining them (e.g. "activity cliffs"); the introduction should focus more on comparing with the published successful ML approaches for peptide drug discovery (e.g. <https://doi.org/10.1016/j.ijbiomac.2024.138880>) and computational ways of sparse sampling (e.g. like it is implemented in high-dimensional NMR); the (training/evaluating) datasets should be clearly defined in the main text (a table, perhaps); the four phases of the workflow should be indicated (in Fig.2.); Fig. 2 could not be discussed before Fig.1. In Fig.1. it is unclear what "membrane MD" means and why it is used. At the moment, MD (both coarse-grained and all-atom) appears unnecessary as it does not contain any comparison to "non-ACP" benchmarks. No change in the outcome/conclusions would emerge if you remove all the MD data. SD for blood cell lysis results should be reported, and the number of technical and biological replicates should be clearly stated when discussing results. The TARSA performance would be beneficial to compare against other known ML algorithms on the same datasets using conventional comparison parameters.

The next problem is what the authors globally aim to search for (and how they validate the results). Their aim is not the manuscript-claimed "anticancer peptides" but rather "12mer membranotropic cationic helical peptides with two (very different in nature and in validating experiments) selected cell lines selectivity. The apparent misconception of the search strategy manifests itself already in the practical output of the approach: starting from 1.6B peptides in the initial search pot (plus another 30M in the alternative screening branch), the authors ended up with only 3 (more or less) "anticancer peptides". Even among 105 (top!) candidates, only 15 (i.e. 14%) possessed recognizable targeted "anticancer" cytotoxicity in the HTS validation step as it was designed. Moreover, the ic50 values against target MDA-MB-231 cells for 15 best are within a not very impressive low micromolar (7-93µM) range. Further (this result is hidden in the appendix Fig. H2 but should be emphasized and discussed in the main text), the cytotoxicities of the 3 ultimate "anticancer peptides" were not genuinely cancer-selective (approx. same ic50 value against non-cancerous MCF10A for all three screening effort "winners"). Together with the incoherence of the lytic activities (PBMC and RBC data differ significantly, Fig.7), these entire results challenge the whole idea that a universal "anticancer" selectivity can be shown (and used for peptide classification) by

comparing activity against one adherent cell line vs activity in a hemolytic or leucolytic experiment, which is the core endpoint in the mastoparan-optimization and is implemented here for ACP/non-ACP classification. Therefore, something appears conceptually wrong with the selection criteria or TARSA training.

It is not clear also why for the validation from the 110K candidates of the PDB pot 95. In contrast, of the 2.8M candidates originating from the GEN pot, only 10 were selected for experimental validation. The origin (PDB/GEN) of peptides should have also been maintained throughout the validation steps. Could this be that none of the 10 (top-most!) GEN-originated sequences did work (the first "active" peptide has the number 11) and the whole PDBlarge screen study was initiated due to this failure?

Next, the authors should know that the (protein) PDB dataset is redundant and is biased towards structures of soluble (non-membrane) proteins and that short polypeptides are "conformationally plastic" in the sense that their conformation vastly depends on the environment. Hence, there are no environment-independent "intrinsically helical" sequences - even their benchmark mastoparan (as the authors showed previously) is a random coil in aqueous buffers and only folds helically in the presence of membrane mimics. In contrast, for peptides (protein fragments) from the PDB, a helical fold would be dictated by proteinous neighbourhood and (often) crystal packing conditions (dominating X-ray-derived structures). So, the consequent problem would be to adequately implement peptide descriptors into QSAR (e.g. hydrophobic moment would be different per unstructured and helically-folded peptide). In contrast to small molecules, for peptides, good descriptors should be the 4D or 3D ones. This problem has to be discussed in the pre-validation part of the manuscript. To the same problem - it is not clear why 12mers were selected as a target - as far as I know, mastoparan is a 13mer (by the way, which out of 40 known mastoparans was used should be defined)- and whether on the TARSA stages, the peptides for the training/evaluation possessed free N/C-termini. Mastoparan has a charged N-term but carries C-amidation (activity profile changes when the terminus is nonprotected, as the authors previously showed). Fragments of proteins in the PDB database would be terminally not charged (termini would, in most cases, participate in the peptide bonds. Several other less serious corrections may be suggested, but they may change/disappear upon revision.

(Remarks on code availability)

Version 1:

Reviewer comments:

Reviewer #1

(Remarks to the Author)

The authors have conducted the MD simulations as requested. Although there are still issues with the way the results are reported, particularly regarding the convergence of MD, the simulations are now clearly better and more robust. I would especially like to have the trajectories available via a public archive. At the same time, the bias issue due to limited structural data remains valid. This is a more general issue that cannot be corrected without a new study involving a large experimental part. As such, this is outside the scope of the normal review process, and I am satisfied with these responses.

(Remarks on code availability)

Reviewer #2

(Remarks to the Author)

The authors answered all comments and I don't have more comments.

(Remarks on code availability)

Reviewer #3

(Remarks to the Author)

The manuscript has been improved, but I am not convinced it is in a publishable form yet. Some of my concerns were addressed, and others were formally touched upon in apparently generative AI-supported answers. This manuscript will be better received in specific Drug Discovery or Bioinformatics communities in a more specialised journal - the view that revision failed to change.

The major issues with this manuscript remained after this revision.

Namely, their virtual screening study's stated primary object (training, screening, and validation) are "anticancer peptides" (ACP). However, how the authors introduce and define peptide sequences (features) they are searching for is, to my regret, incapable of achieving the desired activity. Amphipathic, cationic helical peptides targeting the plasma membrane are a marginal (and, indeed, very well-studied: hence, the conclusion) type of bioactive peptides if one looks at them from the perspective of the "anticancer" function. Successful (i.e. highly potent, highly selective, biostable and bioavailable in vivo) are the peptides that are conformationally non-helical and possess proteins, not lipid membranes, as the targets in the primary MoA. I suggest the authors to redefine the peptides they are searching for as "mastoparan-like" or "MDA-MB-231 cytotoxic" or alike - but avoid generalisation of searching for an (universal) ACP. Without this detailisation, the readership will be misled, and this is better to avoid from the scientific rigour perspective.

The next remaining problem is a poor "wet lab" validation methodology. As seen from the methods - the authors purchased both the SPOT and SPPS-produced peptides. I come from the peptide synthesis field and can assure the editor that doubling any of the libraries or (in a number of tested peptides) or extending the length of the peptides beyond 12mers (up to 20-25mers, for instance), economically would not be a significant burden as both technologies are pretty developed as of now.

(I would not really like to stress here the problems with concentration determination with NanoDrop for sequences w/o Trp, inconsistency of the cell assays in different media, different amounts of FBS, different cell numbers, differences in adherence, different readouts (LDH release, oxidative stress-sensitive metabolic resazurin transformation, Hb-release), different peptide-exposure times, etc. All these cavities should be well-known to the authors. All disallow any robust conclusion about selectivity if comparing RBC, PBMC, and MDA-MB-231, which are left without critical discussion). Further, I insist - in the current form of the manuscript, the in silico (CG and MD) investigations are useless - they bring no contribution to the workflow (or such impact is not clearly described), do not affect selection and do not influence the conclusions and, therefore, could be readily removed or shifted to SI with no damage to the manuscript narrative.

Next, at least from the text (bioinformatic part), it is unclear what is meant by %potency for the selected sequences. If IC50 to PBMC (<15%) and to MDA-MB-231 (>50%, Fig. 5A-C) was meant, then the calculated values must be numerically compared to experimental data in the validation effort.

Still, I would like to know the dataset identity (PDB12mer or GEN12mer origin) in the validation step - the request that the authors preferred not to address.

There are further minor inconsistencies in the methodology, terminology, figures, and workflow descriptions, but those may make sense to address after a significant revision should the authors implement my above concerns.

(Remarks on code availability)

Version 2:

Reviewer comments:

Reviewer #3

(Remarks to the Author)

I regret to state that the authors did not address my concerns to my satisfaction. Hence, I cannot recommend the current revision for publication. I reiterate my suggestion to resubmit to an ML, peptide science, or drug discovery journal, where it would be more appreciated by the audience and find the paradigm coherence.

Just reflecting on the response to my previous revision remarks:

1. The major, rather semantic, but fundamental problem remains, directly affecting the impact and the manuscript placement. Nature Communications is a highly ranked journal with a broad audience, broader than the author's field. However, the manuscript misleadingly states that the major physicochemical hallmarks of "an anticancer peptide" are features of "a membranolytic peptide" (by definition, these would be non-selective, lysing all types of membranes and incapable of discriminating between cancer/non-cancer cells). This is similar to having a cationic detergent, which, despite being active membranolytic, would have less of a pharmacological sense to develop as an anticancer/anti-tumour drug. Helical and membranolytic peptides are the least capable of advancing to clinical studies and approval (check the current pipelines and the 29 FDA approvals). This (non-helicity and other than membranolytic MoAs for the advanced ACP) should be stressed and critically discussed. I never doubted the existence of the term ACP, but I doubt that the authors' study/search is truly an ACP. Hence, I suggest calling within their manuscript the PepSce TARSA-screened target peptides and the final three sequences selected, not "potent ACPs"/"non-toxic towards healthy human cells", but rather "Mastoparan-like", "membranolytic helical", "MDA-MB-231/PBMC-discriminating helical peptides". This concern remains a major conceptual problem of this manuscript.

2. When pointing to the "wet lab" validation problems, I did not ask for justification of the NanoDrop use (that instrument would have problems even for the W-containing sequences in the SpotArray), but to critically address experimental problems and respectful tone down of the statements regarding validity of the experimental conclusions in the discussion section at least. (see reflection to p.5 below). Also, these experimental problems should prompt the authors to reconsider their nomenclature (see above) and avoid generalizations, like calling the "MDA-MB-231 mid uM cytotoxic peptides" as "ACPs" and "PBMC-low lytic peptides" as "non-toxic against healthy human cells".

If the Editor suggests and the authors are willing to implement this request, they should pay special attention to their "standard in the field" statement on the IC50 determination shown in Figs. 7 & 8, which claim "3 biological/4 technical replicates", but the results (Figs. 7 vs. 8) differ significantly (e.g., against MDA-MB-231). To judge the values, I would need to examine the raw data or have a definition of a biological/technical replicate as understood by the authors.

3. Regarding the CG-MD. For me, it sounds contradictory if in the manuscript the authors suggest only validation and mechanistic plausibility evaluation (as the purpose of MD), but in the response to my critique, they claim they would have revised ("had the MD results contradicted our hypothesis, we would have revisited and refined our discovery approach"). The same kind of validation/plausibility information can be obtained from analyzing the physicochemical properties and composition, rather than spending computational time on simulations in cholesterol-free symmetric bilayers, which yields output that is difficult to correlate with RBC and/or PBMC membrane-perturbing results. If the MD data is omitted from the main manuscript, it will not change either in the narrative or in the conclusions. I believe the in silico data warrant SI in the context of this manuscript or a separate publication if the authors wish to avoid wasting the results. Further to this point, Fig.1A shows CG-MD as part of the first TARSA application. Was it also applied to the 36M helices of the PDBlarge? Further in Fig. 1 B, "Membrane MD" appears to be part of TARSA, with an unclear function, as the authors claim not to use it for filtering (same for the "literature properties").

4. Even with this clarification of the "potency" meaning, from the manuscript, it should be clear at which concentrations of peptides and for which cells the parameter is applied every time.
5. I did not stress in the second round (maybe I should have) that some of my initial concerns/suggestions were not addressed/ignored, despite a very lengthy ChatGPT-assisted answer, mentioning only the identity of the library origin for experiments in the validation step. (I would still expect all of my concerns to be implemented in the further revisions, should the Editor conclude on it). However, now the authors provide (rather hide in the supplementary) the identity, but without any indication in the manuscript main text and a clear justification (or at least a comment), explaining why out of 110K (in the fig.1)/200K (in the text) they take "Top 95 of PDB12mer candidates" and from "2.8M of Gen12mer candidates" only "Top 10" for validation. For the sake of real pipeline validation, it would have been incredibly useful to keep the two libraries separated. E.g., in the validation step for the PDB12mers, their approach formally provides a ca. 12% hit rate, whereas the GEN12mer library can be claimed to have a 40% hit rate.
6. There are further corrections required - still different phases of the suggested approach are non-uniformly described/illustrated, and some minor inconsistencies (in terminology and word capitalisation) persist, etc.

(Remarks on code availability)

The original comments are provided in *italics*. Our responses in **blue plain type**. Inserts into text are underlined. Specific mentions of updated sections are **blue bolded**.

Reviewer 1

The manuscript describes the development of an ML-based virtual screening pipeline for anti-cancer helical peptides, aiming to achieve high performance for giga-scale libraries. While the topic is undoubtedly valuable, the current form of the manuscript makes it challenging to evaluate the true performance of the application.

1. *Training Set Bias: I am particularly concerned about the potential bias due to the small initial training set size (210 Mastoparan derivatives). This limitation means the resulting application is highly influenced by Mastoparan SAR, and one must also consider the high cytotoxicity of many of the compounds.*

Response: We sincerely appreciate the reviewer's concern regarding potential bias in the training set due to its relatively small size (210 Mastoparan derivatives) and the possibility that this may overly influence the structure-activity relationship (SAR) modeling. We acknowledge that an initial dataset of this size inherently limits generalizability and may lead to an overrepresentation of Mastoparan-like physicochemical properties in the predicted anticancer peptides (ACPs). However, this issue is a broader challenge in machine learning (ML)-based ACP discovery, as the lack of diverse, tissue-specific ACP datasets with experimentally verified activity under uniformized and reproducible conditions remains a fundamental bottleneck in translating ML models to real-world applications. This is further discussed below but in keeping with the reviewer's comments we have added the following sentences early in the discussion (**P32, L8-14**): "The training set used was biased towards a set of mastoparan derivatives and we acknowledge the inherent potential bias. However, this dataset was uniquely valuable due to its continuously-valued % cell inhibition measurements obtained under standardized conditions with particular cancer cells, in contrast to literature ACP datasets that generally use non standardized and/or binary classification results with varied cell lines. To further mitigate overfitting and improve generalization, we used an ensemble approach to incorporate external ACP datasets with greater sequence diversity."

Necessity for using Mastoparan Dataset despite its small size

Despite its limited size, the Mastoparan derivatives dataset was uniquely valuable for training our regression model due to its continuously-valued % cell inhibition measurements obtained under standardized conditions. In contrast, many publicly available ACP datasets primarily contain binary classification labels (ACP vs. non-ACP), making them less suitable for training regression models capable of predicting graded biological activity. This was a key reason for selecting these Mastoparan derivatives as the foundation for training TARSA. Furthermore, since

this dataset was derived from a controlled in-house study, it provided a consistent experimental framework, eliminating variability introduced by cross-laboratory differences in assay conditions - a common issue in most ACP datasets.

Concerns about overfitting to Mastoparan SAR

To mitigate concerns regarding potential overfitting to Mastoparan SAR, we incorporated external ACP datasets with greater sequence diversity while maintaining the rigor of our predictive framework. In particular, we integrated ACP sequences from a heterogeneous, publicly available dataset- CancerPPD, which contains experimentally validated ACPs from various sources. However, since these datasets often lack standardized continuous inhibition values, we considered ensemble approach consisting of (i) A regression model trained on the Mastoparan dataset (due to its continuous-valued % inhibition data), ensuring quantitative predictions of peptide potency, and (ii) A classification model trained on CancerPPD ACPs, providing additional validation of whether a given peptide is likely to exhibit anticancer activity. This ensemble approach ensured that peptides identified by TARSA were not simply predicted to be active based on Mastoparan-like features but also validated independently against a broader ACP sequence space. Only peptides for which both models agreed on ACP potential were advanced for experimental validation.

It is well established in the machine learning literature that ensemble models improve generalization performance, particularly for out-of-distribution (OOD) samples¹⁻³. Consistent with this, we observed that incorporating the CancerPPD-trained classification model as a filter improved the sequence variability of TARSA-predicted ACPs (Fig. 3C). Specifically, TARSA-identified peptides exhibited greater sequence diversity compared to Mastoparan derivatives, demonstrating that the model was not simply replicating Mastoparan SAR. Feature distribution analysis (Fig. 3D-F) revealed that the physicochemical properties of TARSA-generated peptides occupied an intermediate space between Mastoparan derivatives and ACPs from CancerPPD. This further confirms that our consensus filtering approach enabled the model to generalize beyond Mastoparan SAR, capturing a broader range of ACP-relevant structural motifs.

Concerns about toxicity

The reviewer also raises an important concern regarding the high cytotoxicity of peptides. It is worth mentioning that this is a feature of most anti-cancer drugs. We hypothesize that this effect was due to our training dataset of Mastoparan analogs, which often exhibit strong anticancer effects due to their membrane-disruptive nature, they also tend to be hemolytic and often non-selective. To address this in our modeling setup, we incorporated an additional filtering step that deprioritized peptides predicted to have excessive cytotoxicity using a toxicity-predictive model trained on hemolytic Mastoparan derivatives (in-house PBMC active dataset). While this filtering

step successfully reduced the frequency of highly cytotoxic peptides, some TARSA-discovered ACP candidates still exhibited cytotoxicity upon experimental validation, as the reviewer pointed out. This is an expected limitation in translational ML models, as no SAR model perfectly correlates with biological activity *in vitro*. The inherent tradeoff between potency and selectivity in ACP design remains a challenge, but our toxicity filtering approach represents an important step toward mitigating this issue. We have now discussed this in the updated manuscript in **section “Novel peptides selectively inhibited the viability of MDA-MB-231 breast cancer cells” (P27 L8 – P29 L14) and Figure 8.**

In summary, while our initial training dataset was limited to 210 Mastoparan derivatives, we implemented multiple safeguards—including ensemble learning, external dataset integration, diversity filtering, and cytotoxicity screening—to ensure that TARSA did not overfit to Mastoparan-specific SAR. The observed increase in sequence diversity (Figure 3C) and the feature distribution of TARSA-generated peptides (Figures 3D-3F) intermediate between Mastoparan and CancerPPD provide strong evidence that our approach enabled the model to generalize beyond Mastoparan-like peptides and identify novel ACP candidates with improved selectivity.

2. *Validation of MD Simulations: The validation of MD simulations is another significant issue. Although the GC-MD is performed with 5 replicas, it is not clearly validated whether the selected simulation time is sufficient for the task. This is critical since outliers are screened out. It is necessary to test if longer MD simulations are needed. For the final MD simulations, the timeframe is absolutely too short, and several replicas are required. In its current form, the simulation time is insufficient, as true equilibrium or even metastable states cannot be reached in a timeframe shorter than several tens of microseconds. Replicas are mandatory at this stage.*

Response: We thank the reviewer for the comment. Regarding coarse-grain simulations, we have now performed three independent 100-ns MD runs for each peptide, instead of a single shorter run, with the details highlighted in Section G1 of Supplementary Information (**SI pg 32-35**). Additional analyses (RMSD, contact profiles, and replicate comparisons) are included to confirm convergence (amended **Figure G2**). We have also added an additional control set from the DRAMP set of experimentally confirmed inactive peptides, and thoroughly compared the results of this set with the previous one (**amended Figure G2 and rewritten Section G1 of Appendix (SI pg 32-35)**).

Regarding all-atom simulations, we extended the simulations from 500 ns to 1 μ s total for each system, and ran three replicas per peptide, as suggested by the reviewer. We note that the choice of this timescale and number of replicas is consistent with recent studies employing all-atom molecular dynamics to study the interactions of machine learning-derived peptides with cell membranes⁴.

3. *Manuscript Structure and Readability: The manuscript heavily emphasizes computational details, which is good, but it complicates following the manuscript. The overall scheme of the manuscript is unclear when one starts to read the results section, making it almost impossible to understand the validity of the results at this stage.*

Response: We sincerely appreciate the reviewer's feedback regarding the manuscript's structure and readability. We fully acknowledge the importance of clear and structured presentation and hence we have reorganized our manuscript to improve the contextual flow. In particular, we have

- **Reorganized and rewritten the results section** by first providing a brief overview of our entire workflow before getting into the presentation of the experimental results. This helps the reader to understand the subsequent Results section better as they now have an understanding of why each subsection of results section is necessary and how it affects our overall ACP search pipeline.
- Removed the section on **ablation tests** and **moved it to the appendix I1** to reduce computational burden in the main text.
- **Rearranged the figures** to have a better logical flow and connection to our main text.

Reviewer 2

The authors proposed an algorithm focusing on activity cliffs within peptide libraries that can help prioritize the high-potential candidates. They proposed a computational model for screening peptides with anticancer peptides using reinforcement learning techniques and identified 3 potential anticancer peptides without toxicity through experimental validation procedure. Through report 15 out the top 100 peptides selected using this method showed potent activity against breast cancer cell, including drug-resistant triple-negative breast cancer through insilco and successive wet-lab experiments to validate the ACP potential of top 3 peptide identified using TARSA's method. However, I have several comments which author should address to increase the scope and strength of manuscript.

1. *In the result section, specially in motif discovery, it is mentioned TARSA was applied to PDBlarge to identify potent anticancer motifs, which were subsequently used to construct a library of fixed length 12-mer peptides.*

a. *But how did the author confirm that those discovered motifs have biological significances, how did you confirm that they are really helical ACPs motifs?*

Response: We sincerely appreciate the reviewer's insightful query regarding how we confirmed that the discovered motifs from TARSA possess biological significance and whether they truly represent helical ACP motifs. Below, we provide a detailed clarification of our study design and

validation strategies, which ensured that the identified motifs were both structurally and functionally relevant.

The reviewer correctly points out the importance of verifying whether the discovered motifs are truly helical. We would like to clarify that this property is inherently built into our dataset construction methodology. We specifically curated the PDB_{large} dataset by extracting protein segments that were annotated as helices using the Dictionary of Secondary Structure of Proteins (DSSP)^{5,6}. DSSP is a widely recognized computational tool that assigns secondary structures to protein sequences based on their crystallographic or NMR-determined atomic coordinates. Because DSSP was used as the primary filter, all sequences in PDB_{large} are structurally confirmed α -helix, 3-10 helix, or π helix by construction. TARSA was subsequently applied to PDB_{large} to identify a subset of sequences with a high probability of anticancer activity. Since the parent dataset itself comprises only experimentally validated helical structures, it follows that the identified motifs also retain helical conformations.

Thus, by design, the discovered motifs are structurally helical ACP candidates, making additional confirmation of helicity redundant.

Next, in order to establish the biological relevance of the discovered motifs, we employed the following validation strategies. Since all motif sequences were extracted from experimentally resolved structures in the PDB, we leveraged the metadata associated with their parent proteins to gain functional insights. Specifically, for the top 480 most potent TARSA-predicted motifs, we examined the biological roles of their source proteins:

- o **Apoptotic and Tumor-Suppressor Proteins:** Many of the parent proteins were annotated in PDB as involved in apoptosis, tumor suppression, or cell cycle regulation, providing strong evidence that their derived peptide fragments might also possess anticancer properties.
- o **Proteins from Antimicrobial and Anticancer-Rich Species:** Some motifs were derived from proteins belonging to species previously reported as rich sources of antimicrobial or anticancer peptides (e.g., amphibian secretions, marine invertebrates, and venomous organisms).
- o **Functional Keyword Analysis:** We performed a manual analysis on PDB keywords associated with the parent proteins of these 480 motifs, revealing a significant overrepresentation of terms related to membrane interaction, cytotoxicity, and immune modulation—common functional themes among known ACPs.

These analyses strongly suggest that the identified motifs are biologically relevant and not random sequences selected from helical proteins. This discussion is now presented in the "In Silico Validation of Discovered Anticancer Peptide Candidates" section of the main text and in

Appendix F4: "Resonance Between Discovered Motifs and Known ACPs: Insights from PDB Metadata."

While computational predictions provide strong supporting evidence, we acknowledge that ultimate confirmation of anticancer activity requires experimental validation. To this end, we selected a subset of predicted ACP motifs for *in vitro* testing, as detailed in the experimental validation section. The observed anticancer activity of TARSA-generated peptides further reinforces the biological relevance of the discovered motifs.

We sincerely appreciate the reviewer's constructive feedback, as it allowed us to provide further clarification on the structural and functional relevance of the discovered motifs.

b. Are there any statistical calculation to confirm the presence those motifs in training samples?

Response: We have thoroughly examined the discovered motifs from PDB_{large} by comparing their Levenshtein edit distance with respect to the 210 Mastoparan derivatives in our training dataset. This analysis revealed that the minimum edit distance between any discovered motif and its closest match in the training set was 5, indicating that none of the motifs appeared exactly in the training samples (added new **Appendix Figure F7**). This statistical calculation helps confirm that the motifs discovered in PDB_{large} are distinct from the peptides used in training.

c. And if these motifs are prevalent in helical peptides, what are the most preferential position of those helical motifs which can assist in design of anticancer peptides with higher potency?

Response: Regarding the prevalence of the motifs, the only potential motif with % cell inhibition greater than 45%, which was a minimum of 4-mers and found in at least three training examples, is the 'LKAL' motif. This motif appeared in 150 of the 210 Mastoparan (INLKALAALAKKIL) derivatives. However, having said that there is no evidence that a peptide as small as 4 amino acids can per se have anti-cancer activity.

As for positional preference, since the training dataset is derived from single amino acid substitutions in the Mastoparan sequence, the position of the 'LAKL' motif, whenever present, was consistently fixed starting at index 2. This positional consistency across the training peptides indicates a potential preferential position for its occurrence. When examining the relationship between the presence of this motif and the potency of the ACPs, we did not observe a significant enhancement in potency. The distribution of potency of the peptides containing the 'LAKL' motif was similar to the overall potency distribution of the entire training dataset, as shown in the new **Appendix Figure F8**. Thus, while the motif appears to be frequent in the training set, its presence alone did not confer higher potency.

We have added these statistical analyses to **Appendix F4 (SI pg 28-30)** under subsection **“Sequence and Positional Analysis of Discovered Motifs”**

2. *Why did the author limit the exploration of bioactivity prediction assessment with random forest, XGBoost, and MultiLayer Perceptron? Is there any motivation in selecting only those models for prediction? If yes, can you clarify the reason?*

Response: We thank the reviewer for this inquiry. The choice of appropriate machine learning method and model architecture for bioactivity prediction was a fundamental design choice we had to make. Given that our primary objective of building a system that can efficiently perform virtual screening of very large libraries, this choice was guided by the aim to have the optimal tradeoff between prediction speed and accuracy. Hence, we prioritized simpler models above overparametrized models such as those based on Transformers and large-language models were not considered because of this reason. We have now addressed this in **section “Descriptor Relevance in QSAR Modeling for Peptide Potency” (P11 L23- P24 L5)** .

3. *In Figure 3 (now Figure 5), though the discovered peptide has relatively low molecular weight which can ease the synthesis process, but*

a. *why does the discovered peptides don't catch the regular distribution of anticancer peptides available in CancerPPD? Is this because of properties specific to helical ACPs? Please mention it clearly why discovered peptides have relatively shorter variance in case of charge density and hydrophobic moment?*

Response: CancerPPD is a heterogeneous dataset comprising ACPs associated with diverse cancer types and secondary structures. While this broad diversity makes CancerPPD valuable for training classification models that distinguish general ACPs from non-ACPs—similar to the Consensus filtering approach described in Section “Consensus filtering and visual analysis of candidate ACPs” of our study—it is not necessarily optimal for regression-based models aimed at discovering ACPs specific to a particular cancer type, such as breast cancer.

To address this limitation, we leveraged our Mastoparan-derived inhouse dataset, which consists of systematically designed single-residue variants of Mastoparan, each with experimentally measured cytotoxicity against MDA-MB-231 breast cancer cells. This dataset provides a quantitative structure-activity relationship (QSAR) framework that allows our model to learn fine-grained physicochemical patterns directly relevant to breast cancer ACPs. However, the primary limitation of this dataset is its lower diversity, as all peptides share a common helical scaffold and differ from Mastoparan by only a single amino acid substitution.

As illustrated in Figure 5D-F, the key ACP properties identified through both literature review and feature-ranking analysis exhibit a distinct distribution in the top $k = 480$ peptide candidates predicted by our regression-based screening model. **Since the regression model was trained on the Mastoparan dataset, it is expected that the discovered peptides exhibit physicochemical properties more aligned with Mastoparan-derived sequences than with the broader CancerPPD dataset.** This is because CancerPPD includes ACPs targeting a wide range of cell types beyond breast cancer and incorporates peptides with diverse secondary structures, including non-helical ACPs. Also there have been few studies examining whether all of the amino acids in longer peptides are required for optimal activity.

Importantly, despite differences in exact distributions, the discovered peptides still fall within the broader range of ACP properties observed in CancerPPD (Figure 5D-F). This suggests that while our approach does not explicitly replicate the global property distribution of CancerPPD, it successfully identified peptides that retained biologically relevant physicochemical characteristics commonly associated with validated ACPs.

Lower Variance in Charge Density and Hydrophobic Moment

The relatively lower variance in charge density and hydrophobic moment observed in the discovered peptides is a direct consequence of the homogeneity of the Mastoparan-derived training dataset. Since Mastoparan is an amphipathic α -helical peptide, the charge and hydrophobic moment distributions within the dataset are inherently constrained. As a result, the regression model trained on this dataset prioritizes physicochemical features within this narrower range, leading to a more conservative discovery of ACPs with physicochemical properties that mirror those of Mastoparan variants rather than the broader CancerPPD dataset.

This highlights a fundamental exploration-exploitation tradeoff in ACP discovery. CancerPPD offers greater exploration potential due to its high diversity, but a model trained on such a dataset may generalize too broadly and struggle to identify breast cancer-specific ACPs. In contrast, the Mastoparan dataset enables greater exploitation, allowing the model to identify peptides with high efficacy against MDA-MB-231 cells, but with a more constrained physicochemical profile and generally shorter (and thus cheaper to make) sequences. This tradeoff underscores the scarcity of high-quality, diverse datasets for ACP discovery, emphasizing the need for future dataset expansion to improve the generalizability of computational peptide design models, possibly through iterative analyses. Therefore, we would contend that our TARSA scheme may be useful in future dataset expansion.

In response to the reviewer's question, we have briefly addressed this issue in section **"In silico validation of discovered anticancer peptide motifs"** (P21 L1-L6) and included a detailed

discussion in **Appendix F4** under the **section “Influence of Training Data on ACP Property Distributions” (SI Pg 27)**

b. *Are these a good indicator to explain helical ACPs properties? Please review and mention it clearly.*

Charge density and hydrophobic moment are well-established physicochemical properties that play a critical role in the bioactivity of α -helical ACPs, particularly in their membrane interactions and selective cytotoxicity toward cancer cells. Feature ranking of the physicochemical descriptors used for our regression model also showed these properties prominently. We have highlighted this in the manuscript **section “Descriptor Relevance in QSAR Modeling for Peptide Potency”** as follows:

“this model identified key physicochemical properties—such as sequence length, charge density, and hydrophobic moment (Figure 3D)—which are well-established determinants of peptide bioactivity, particularly in helical peptides”

4. *Authors mentioned about the active and inactive ACP peptides in manuscript specially Figure 2. But didn't mention about the threshold he applies for IC50 value to classify into experimentally active and inactive. Please check it and add details on how to create this dataset.*

Response: We appreciate the reviewer's insightful query regarding the classification threshold for active and inactive ACP peptides, particularly in relation to Figure 2. We would like to clarify that the dataset used for this classification was primarily derived from CancerPPD, a curated repository of experimentally validated ACPs spanning various cancer cell types, including breast cancer. The positive examples were directly obtained from CancerPPD.

For the negative dataset, we followed a well-established approach described in Vijayakumar et al. (2015) and constructed the dataset using peptides extracted from UniProt. The selection process for negative examples was as follows:

- Peptides with a length between 15 and 100 amino acids were selected to ensure similarity in sequence length distribution between actives and inactives.
- Peptides annotated with keywords indicative of anticancer activity, such as "apoptosis," "anti-cancer," "anti-tumor," and "programmed cell death," were removed to prevent contamination with potential ACPs.
- The remaining sequences were filtered using the CD-HIT server to eliminate redundancy, ensuring that no two peptides shared 100% sequence identity.
- This process resulted in a final dataset of 4,019 negative peptides and were made available by Vijayakumar et al.⁷ (2015),

We had originally described the data curation in Appendix D2 but this has now been moved to the main text for clarity. Specifically, **section “Consensus filtering and Visual Analysis of Candidate ACPs”** now mentions:

“During the consensus filtering stage, a binary ACP classification model, trained on a diverse set of publicly available ACPs from CancerPPD and non-ACPs curated by Vijayakumar et al. (2015), was employed.”

With regards to the experimental classification of actives from the peptide spot array, we employed a threshold of 60% inhibition as the cutoff to identify peptides for further testing. This is described in section “Novel peptides exhibited toxicity toward MDA-MB-231 breast cancer cells” of the main text as follows

“Among the 105 peptides tested, fifteen reduced cancer cell viability by >60% at this peptide concentration and thus were selected for further investigation”

For IC_{50} -based validation, we categorized a peptide as active if it exhibited a relatively low IC_{50} against MDA-MB-231 cells while maintaining a significantly higher IC_{50} vs. peripheral blood mononuclear cells (PBMCs) and red blood cells (RBCs). Specifically, we applied a threshold where the IC_{50} in non-cancerous cells had to be at least 5-fold higher than in MDA-MB-231 cells.

This approach was already described in the manuscript (section “Novel peptides selectively inhibited the viability of MDA-MB-231 breast cancer cells”), as seen in the sentence:

“Notably, peptides 11 (“LLQWLLKRLKAK”), 29 (“IILKLLDFILK”), and 57 (“TLLTAIVKFLK”) displayed IC_{50} values of $31.9 \pm 2.0 \mu M$, $14.3 \pm 1.5 \mu M$, and $15.9 \pm 2.0 \mu M$, respectively, for MDA-MB-231 cells, while their IC_{50} values for non-cancerous cells occurred at concentrations at least 5.5-fold higher.”

To further clarify this point for the reader, we have explicitly added the term “threshold” in the revised text to highlight the classification criteria based on IC_{50} ratios. In particular, we have **added following to section Novel peptides selectively inhibited the viability of MDA-MB-231 breast cancer cells**

“Peptides were classified as active if their IC_{50} values against MDA-MB-231 cells were at least 5-fold lower than their IC_{50} values against PBMCs or RBCs.”

5. *During Dataset Construction for training the TARSA model, the author selected 1.5M 12 Mers (PDB12mer) for dataset construction?*

a. *Does it ensure that, this dataset include representative samples (natural variation in motifs presence) specially in Helical ACPs with helical motifs?*

Response: Thank you for your insightful question regarding the representativeness of the PDB_{12mer} dataset, particularly in terms of the natural variation in motifs, especially for helical anticancer peptides (ACPs).

In order to respond to this query, we would like to clarify the construction of PDB_{12mer} dataset. The construction of the 1.5M 12-mer dataset (PDB_{12mer}) was a careful, systematic process designed to ensure the inclusion of diverse helical motifs, with a specific emphasis on those relevant to ACPs.

The foundation of this dataset was the comprehensive PDB_{large} dataset, which consists of all protein sequences in the Protein Data Bank (PDB) annotated as helices by DSSP^{5,6}. This dataset provided an expansive starting point of 36 million sequences, from which we began our analysis.

Next, we employed the TARSA screening algorithm to sift through the PDB_{large} dataset and identified 3.2 million sequences predicted to exhibit high biological activity according to our ACP regression model. This subset, which we refer to as the PDB_{top-motifs} dataset formed the basis for constructing the final PDB_{12mer} dataset. (PDB_{top-motifs} nomenclature is used only for the sake of explanation in this response).

Given that many of the sequences in PDB_{top-motifs} exceeded the desired 12-mer length, we implemented a sliding window approach to generate 12-mer subsequences from longer peptides i.e., we converted longer sequences into multiple 12-mer subsequences. The 12-mers in the PDB_{top-motifs} dataset were used as-is in the PDB_{12mer} dataset. The final PDB_{12mer} dataset, consisting of 1.5 million unique 12-mer peptides, therefore captures a wide variety of helical motifs whose features are broadly aligned with anticancer activity.

It is important to note that while PDB_{12mer} is not a direct representative sample of the entire PDB_{large} helices dataset, it is highly enriched in helical motifs with strong predicted anticancer activity. This enrichment is a result of the initial TARSA screening process based on predicted biological activity, ensuring that the dataset includes sequences that are more likely to exhibit high anticancer potency. Moreover, the PDB_{12mer} dataset remains representative of the PDB_{top-motifs} dataset by construction. PDB_{top-motifs} was specifically selected for its focus on motifs that are potentially biologically active. Thus, PDB_{12mer} mirrors the diversity of helical motifs present in PDB_{top-motifs}, including natural variations in sequence and structural context, but with an emphasis on potency.

In summary, while PDB_{12mer} is not an unbiased, random sample from the entire PDB_{large} dataset, it is a well-curated and enriched subset that captures a wide range of helical motifs relevant to ACPs, ensuring a high level of diversity in terms of both motif structure and biological relevance.

b. *How did the authors ensure that those helical motifs mostly lie within 12 Mers peptide candidates for training?*

Response: This follows from the above response. The helical motifs dataset $PDB_{top-motifs}$ is contained within PDB_{12mer} dataset by construction.

6. *In Figure 6, relatively the charge density, charge, molecular weights are not symmetric to the Mastoparan training dataset. What is the major reason for these difference in mean? Is this because, the discovered peptides were designed from the comprehensive helical peptides which have likelihood of diverse physicochemical properties compared to Mastoparan? Please mention it in manuscript clearly.*

Response: We appreciate the reviewer's insightful observation regarding the differences in charge density, charge, and molecular weight distributions between the discovered peptides and the Mastoparan training dataset in Figure 6. As the reviewer correctly pointed out, these differences primarily arise due to the fundamental disparities between the library of helical peptides (PDB_{large}) used for motif discovery and the Mastoparan derivatives dataset ($D_{Mastoparan}$) used for training the regression model, as well as the exploratory nature of our TARSA-driven peptide selection strategy.

The Mastoparan derivatives dataset used for training is inherently constrained in sequence space, since it consists of single amino acid substitutions of the Mastoparan peptide. This results in relatively homogeneous physicochemical properties due to the structural and functional constraints of the parent Mastoparan scaffold.

In contrast, our discovered peptides were derived from PDB_{large} , which comprises a comprehensive and diverse collection of naturally occurring helical peptides. These peptides exhibit substantial physicochemical diversity in terms of charge, hydrophobicity, molecular weight, and structural motifs, reflecting their broad evolutionary origins and functional roles in biological systems. Since PDB_{large} is not biased toward Mastoparan-like peptides, the distribution of key physicochemical properties in our discovered sequences naturally differs from the Mastoparan dataset.

Furthermore, our reinforcement learning (RL)-based TARSA framework is designed to optimize anticancer activity, rather than to enforce strict adherence to Mastoparan-like physicochemical properties. While the Mastoparan dataset served as an initial reference for model training, the RL-driven search dynamically explores peptide space, selecting sequences predicted to have high anticancer potency even if their charge, molecular weight, or charge density deviate from the Mastoparan scaffold. This flexibility is beneficial for discovering novel ACPs, as strict adherence

to Mastoparan-like properties may limit the discovery of novel sequences with superior therapeutic potential.

We have made following **addition to section “Consensus filtering and Visual Analysis of Candidate ACPs”**: “It was noteworthy that the physicochemical properties of these peptides deviated from those of the Mastoparan training dataset, which consisted of 210 derivatives limited to single amino acid substitutions of the Mastoparan scaffold. In contrast, the discovered peptides originated from PDB_{large} - a diverse dataset of naturally occurring helices with broader distributions of charge, molecular weight, and hydrophobicity, explaining the divergence of the physicochemical properties of the discovered peptides.” and have also **updated the caption of Figure 6**.

Reviewer 3

The manuscript of Pandey et al. is a bioinformatics paper focussing on the application of machine learning (ML) to drug discovery. The authors suggest and demonstrate the implementation of an ML approach for (rational) sparse sampling of (ultra)large peptide libraries aiming to select a reduced number of potentially bioactive sequences for further validation. Computationally sound and, in principle, correct in the conclusions, the manuscript, in its current state, to my regret, is unsuitable for the broad audience of Nature Communications. It would be better received in a more specialized journal (bioinformatics, ML or peptide medicinal chemistry), but significant revision can be recommended even for an alternative submission.

1. *The submitted manuscript appears as a compilation of two separate works (steaming from two manuscripts): one on sparse sampling/ML for drug discovery (a more substantial aspect) and another - on mastoparan optimization as an "anticancer peptide" (a weaker aspect). This origin explains the discrepancy in the nomenclature being used throughout the fused manuscript (i.e. datasets (peptide libraries) are named differently; the workflow ("PepSce"), the Oracle function also vary in description, etc.) and must be uniformized should authors revise the submission.*

Response: We sincerely appreciate the reviewer’s feedback and the opportunity to clarify the structure and intent of the manuscript. Our overarching objective was to develop a comprehensive, cost-effective pipeline for the identification of novel anticancer peptides (ACPs). This objective inherently involved two interdependent components: (1) efficient *in silico* screening tool for peptide discovery, and (2) accessible experimental validation strategy to identify promising initial hits. These components are not separate works but rather integral to

demonstrating the practical applicability of machine learning (ML)-driven peptide discovery leading to identification of promising, experimentally confirmed leads.

We recognize that this dual approach may have given the impression of two distinct research efforts. However, our work is fundamentally unified by its emphasis on the full discovery-to-validation pipeline. Importantly, while many ML-driven drug discovery studies stop at computational predictions, our study uniquely strengthens the field by prospectively validating the *in silico* findings. We believe that this combination of computational methodologies with experimental wet lab results enhances the impact and reliability of our approach, bridging a critical gap in ML-based drug discovery and we believe that this is a major strength of our work.

Additionally, we thank the reviewer for highlighting inconsistencies in nomenclature across the manuscript. We have carefully revised the text to ensure uniform terminology, improving clarity and coherence. In particular, we have now clarified in **section “Descriptor Relevance in QSAR Modeling for Peptide Potency”** that oracle refers to the ACP activity regression model throughout the manuscript. Specifically, we have mentioned that

“Hereafter, this optimized ACP activity predictor is referred to as oracle proxy f_{θ}^0 .”

We acknowledge that multiple datasets and libraries were used at different stages of this work. We have now **added Table 1** to provide summary of all datasets used. Furthermore in caption of **Figure 1**, we have clarified that PepSce is our overall peptide discovery workflow and TARSA algorithm is the reinforcement learning component of PepSce workflow responsible for fast screening of large libraries.

We appreciate this valuable suggestion, as it has helped us enhance the manuscript’s readability and scientific rigor.

2. *The manuscript has to be restructured/rewritten, use of the terms should not occur before defining them (e.g. "activity cliffs")*

Response: We appreciate the reviewer’s feedback regarding manuscript structure and terminology clarity. We have revised the manuscript to provide clear definitions of technical terms at their first introduction in the main text. For example, "activity cliffs" is explicitly defined when first mentioned in the introduction section (**Pg4, L14-17**).

3. *The introduction should focus more on comparing with the published successful ML approaches for peptide drug discovery (e.g. <https://doi.org/10.1016/j.ijbiomac.2024.138880>) and computational ways of sparse sampling (e.g. like it is implemented in high-dimensional NMR)*

Response: We thank the reviewer for insightful suggestion to enhance the introduction by drawing clearer comparisons with published successful ML approaches for ACP drug discovery and drawing parallel between our method and computational methods related to sparse sampling in NMR. In response we have revised the **introduction section** of the manuscript. In particular, we have said the following

“Machine learning (ML) and deep learning (DL) techniques relying on QSAR models have gained prominence due to their improved predictive power. In these works, a QSAR model is trained on molecular descriptors to predict peptide potency. MLACP, AntiCP 2.0, and ACPred-Fuse, rely on descriptors like amino acid composition (AAC) and dipeptide composition (DPC) combined with ML classifiers, including support vector machines (SVM), random forests (RF), and LightGBM. Deep learning models, such as ACP-DL, DeepACP, and ACP-check, employ architectures like Bidirectional- Long Short-Term Memory networks, Convolutional Neural Network, and Transformer networks to capture sequence-based patterns and physicochemical properties more effectively. More recent efforts, such as ACP-ESM, leverage large-scale protein language models like ESM⁸ to enhance predictive accuracy. Several recent reviews have focused on broad use of artificial intelligence approaches for ACP predictions (though without experimental validation). In contrast, recent work by Yue et. al represents a notable step towards integrating DL-methods with experimental validation by predicting ACP activity of 3.8 million sequences from UniProt and 100,000 sequences produced by deep generative models.”

In regards to sparse sampling in NMR, we have said following in the introduction

“This approach parallels sparse sampling techniques in high-dimensional NMR spectroscopy, which accelerate data acquisition by leveraging the inherent sparsity of NMR spectra. Instead of acquiring a full Nyquist-sampled dataset, sparse sampling selectively captures a subset of data points in the indirect dimensions (such as frequency), significantly reducing experiment time while preserving spectral resolution. Similarly, TARSA strategically samples a subset of peptides from a library—where most peptides are inactive—to maximize the discovery of active candidates while minimizing computational costs. Unlike NMR spectra, which represent continuous signals, TARSA operates on a discrete peptide space, adapting sparse sampling principles to a biological search framework.”

4. *The (training/evaluating) datasets should be clearly defined in the main text (a table, perhaps)*

Response: We thank the reviewer for this valuable suggestion. To improve clarity, we have moved Table A1 from the appendix to the main text, where it is now presented as **Table 1**. This table provides a comprehensive summary of all datasets, including their sources, the number of

sequences, and their role in training, validation, or evaluation. In addition, we have updated the Results section to explicitly reference this table. Specifically, **Section Results** now states: "Table 1 summarizes the different datasets used in this work; see Appendix J2 for library curation details."

5. *The four phases of the workflow should be indicated (in Fig.2.)*

Response: We thank the reviewer for the suggestion to clearly indicate the four phases of the workflow in **Figure 2-D**. We have updated the figure to explicitly label these phases.

6. *Fig. 2 could not be discussed before Fig.1*

Response: We thank the reviewer for pointing this out. We have reviewed the manuscript accordingly and **rearranged the figures** to keep the ordering consistent with how they are described in the main text.

7. *In Fig.1. it is unclear what "membrane MD" means and why it is used*

Response: To improve manuscript's clarity, we **revised Figure 1A**, replacing "membrane MD" term with "coarse-grain MD" term and enclosing it within a separate dotted box to indicate that this step is useful but auxiliary and does not actively influence TARSA's decision-making process.

Coarse-grained molecular dynamics (CG-MD) simulations were employed as an additional validation step to assess the membrane interaction potential of the discovered motifs. This approach is widely used in peptide research to model peptide-membrane interactions at a reduced computational cost while preserving key biophysical insights, such as penetration depth, lipid perturbation, and contact frequency with the membrane bilayer. The purpose of CG-MD in this study was to qualitatively confirm that the identified motifs exhibit membrane-disruptive properties consistent with known ACPs, reinforcing confidence in their biological relevance before experimental validation. CG-MD is discussed and defined in the section: "*In silico validation of discovered anticancer peptide motifs*".

8. *At the moment, MD (both coarse-grained and all-atom) appears unnecessary as it does not contain any comparison to "non-ACP" benchmarks. No change in the outcome/conclusions would emerge if you remove all the MD data.*

Response: We thank the reviewer for her/his comment. We have now extended our simulations and performed multiple, independent replicas, specifically a) three 100 ns coarse-grain replicas (instead of single runs) for each system, and b) three 1 μ s all-atom replicas (extended from the original 500 ns). These extended runs allow us to (a) capture any potential slow timescale

processes, (b) differentiate stable membrane insertion from transient contacts, and (c) gather statistically relevant data across multiple replicas.

We have also included an additional control set of experimentally confirmed inactive peptides from the DRAMP set to demonstrate that the insertion behavior is indeed specific to our designed ACPs (**Figure G2** and **Section G1 of Appendix (SI pg 32-35)**).

With these revisions, the MD results more robustly reinforce the membrane-active mechanism of the lead peptides, supporting the biological activity data, similarly to other recent works in the field⁴.

9. *SD for blood cell lysis results should be reported, and the number of technical and biological replicates should be clearly stated when discussing results.*

Response: We appreciate this comment from the reviewer. The SD values that were calculated are reported in Figure 7 and based on the curve fitting applied to the dose-response curve using our graphical analysis software (Graphpad Prism). For most of the MDA-MB-231 curves, the maximum and minimum are well defined and therefore the software can easily assess a variability from among the presented data points. For many of the RBC and PBMC dose response curves, the maximal toxicity is often not achieved at the highest peptide dose which makes it inherently difficult for the software to fit a curve to the data points, resulting in somewhat large or unknown variability of the calculated IC50. Furthermore, since the RBC and PBMC data was derived from cells isolated from human volunteers, there is an inherent natural variability in the peptide response due to differences in the composition of the cells between donors. This is contrast to the MDA-MB-231 data which was derived from treating cells propagated from the same source and for which response is more consistent. In general, the method worked well to identify those peptides with high activity against MDA-MB-231 cells but low cytotoxicity towards normal cells as the most therapeutically relevant peptide will be those that exert little to no toxic effects within the concentration range tested.

In response, we have added following sentence *Cytotoxicity and hemolysis assays for PBMCs and RBCs* subsection of Methods **section Wet-Laboratory Characterization**.

“All hemolysis and PBMC cytotoxicity assays were carried out using cells isolated from three separate donors and each biological replicate was performed in technical duplicate.”

10. *The TARSA performance would be beneficial to compare against other known ML algorithms on the same datasets using conventional comparison parameters.*

Response: We thank the reviewer for pointing this out. In response, we have added **Table 2** to the main text to show a direct comparison of TARSA’s favorable performance compared to other established ML algorithms.

11. *The next problem is what the authors globally aim to search for (and how they validate the results). Their aim is not the manuscript-claimed "anticancer peptides" but rather "12mer membranotropic cationic helical peptides with two (very different in nature and in validating experiments) selected cell lines selectivity.*

a. *The apparent misconception of the search strategy manifests itself already in the practical output of the approach: starting from 1.6B peptides in the initial search pot (plus another 30M in the alternative screening branch), the authors ended up with only 3 (more or less) "anticancer peptides". Even among 105 (top!) candidates, only 15 (i.e. 14%) possessed recognizable targeted "anticancer" cytotoxicity in the HTS validation step as it was designed.*

b. *Moreover, the ic_{50} values against target MDA-MB-231 cells for 15 best are within a not very impressive low micromolar (7-93 μ M) range.*

Response: We appreciate the reviewer's thoughtful feedback and the opportunity to clarify the objectives, methodology, and key findings of our study. Our primary aim was indeed the discovery of novel ACPs, but we acknowledge that the selection process inherently favored membranotropic cationic helical peptides, given that this structural and physicochemical profile is well-documented in the literature as a hallmark of many known ACPs including the Mastoparan on which our model training is based upon. We accept that this limitation is implicit in our design approach, but ultimately we were still aiming for anticancer peptides using a novel, hitherto-unpublished strategy.

Furthermore, it is essential to emphasize that the identified ACPs represent the first round of candidates, and subsequent optimization through iterative analysis, rational design and chemical modifications will be warranted in the future. We have now mentioned this as a prospective direction for **future work in the Discussion section (P33 L23)** of the updated manuscript. The key contribution of our work is the development of an end-to-end ACP discovery pipeline, which not only generates experimentally validated hits but also produces a high-quality dataset for data-driven ACP research. The field has yet to fully leverage deep learning due to the scarcity of reliable data, and our study addresses this gap by providing a structured computational-experimental framework.

Addressing the Cell-line Selection Criteria:

The reviewer raises an important point regarding selectivity. In the **section "Novel peptides selectively inhibited the viability of MDA-MB-231 breast cancer cells" (P26 L2-5)** of the revised draft we specifically emphasize that selected ACPs were tested against three distinct cell lines—MDA-MB-231 (triple-negative breast cancer), RBCs (red blood cells) and PBMCs (peripheral blood mononuclear cells)—to evaluate cancer cell specificity while minimizing toxicity to healthy cells. Most studies assess toxicity using certain cell lines or metabolically-inactive red blood cells, which could be argued is a potential issue. Our selection was guided by prior work from Hilchie et al.⁹,

who investigated Mastoparan's selective cytotoxicity toward MDA-MB-231 relative to PBMCs. Our findings align with their observations: membranotropic ACPs primarily differentiate cancerous from non-cancerous cells based on membrane composition differences, such as increased anionic lipid content and altered membrane fluidity in cancer cells. Similarly, a comparable effort to ours in designing ACPs using machine learning, Grisoni et al.¹⁰ also compared the selectivity of their ACPs by validating them against MCF-7 breast cancer cells and healthy human erythrocytes. The two cell types thus serve as a reasonable model system for initial validation, and additional cancer cell lines could be incorporated in future studies to further generalize the findings.

Addressing the Search Strategy and the Number of ACP Hits:

1. Reduction from 1.6 billion Peptides to 3 experimentally validated ACPs:

Our approach was designed to identify functional ACPs through a rigorous *in silico* and experimental screening process. The initial peptide search space of 1.6 billion candidates was systematically reduced through multiple filtering steps, prioritizing peptides based on predicted anticancer potential, physicochemical properties, and structural compatibility with known ACP motifs. Such a pre-filtering based on physicochemical and structural properties from the existing literature is not uncommon in bioactive peptide screening studies. A recent impactful work on the screening of hexapeptides for antimicrobial activity, published by Huang et. al¹¹ in Nature BioMedical Engineering, went from a hexapeptide search space of 64 million sequences to a candidate pool of 3.93 million sequences, based on pre-filtering guided by the known structural characteristic that AMPs are typically small amphipathic peptides with a net positive charge at physiological pH. Similarly, our approach prioritized helical peptides, a hallmark of many known ACPs, to systematically narrow the search space from 1.6 billion to 36 million helical sequences in the PDB. The ultimate three experimentally-validated ACPs, can be considered a sample of the highly active peptide space, and are a direct outcome of this stringent selection process. While the absolute number may seem small relative to the initial search space, it is important to recognize that high attrition rates are expected in drug discovery pipelines, particularly for membrane-disruptive ACPs, which require precise balance between activity and selectivity to avoid off-target cytotoxicity. Again, staying consistent with recent impactful works in deep learning-driven peptide discovery, reporting the discovery of the top-k final peptides where k is a single digit number is quite common. Huang et. al¹¹ reported their top-3 AMP candidates, HydrAMP reported 9, Das et al. reported 2, and Zhang et. al¹² reported 3. In the case of ACPs, deep learning driven works complete with experimental validation are harder to find, which uniquely places our work. We thank the reviewer for pointing out the work of Yue et. al¹³, which was performed concurrent with our work. They too point out the lack of computational studies where algorithms are integrated with laboratory experiments. They also report the IC50 for their 10 most successful candidates in Fig.8, and except for a single very cytotoxic peptide (with equal

activity against HUVEC cells), most peptides show higher activity against a single cell line K562 colon cancer cells and activity against 4T1 breast cancer cells is generally lower than the activities we show. Another recent work by Zakharova et al.¹⁴ reports 4 selective ACPs, none of which are more active than ours.

2. Experimental Hit Rate of 14% (15/105 Candidates):

In peptide-based drug discovery, successful hit rates from large screening pools are typically low. In similar high-throughput AMP and ACP discovery studies, experimental hit rates frequently fall below 0.01%, and often only a small fraction of these hits exhibit potency within a therapeutically relevant range. This is especially true when the amount of source data is modest as described in our study. The fact that 14% (15 out of 105) of our top-ranked candidates displayed anticancer activity in HTS validation is therefore a non-trivial success rate, supporting the robustness of our computational pipeline. Here, it is also worth stating that our criteria for defining a successful hit were stringent. We only classified peptides as ACP candidates if they exhibited >60% inhibition of viability in spot-array validation. In contrast, Yue et al. and Ma et al.¹⁵ used a much lower threshold (>20% inhibition), which, if applied to our study, would have increased our reported hit rate to 22 out of 105 candidates.

Additionally, Ma et al. evaluated peptides across 12 different cancer cell lines and classified a peptide as “active” if it showed activity against at least one cell line. This broader classification unsurprisingly led to a higher reported hit rate. However, when focusing specifically on MDA-MB-231, the highly aggressive and treatment-resistant triple-negative breast cancer (TNBC) cell line used in our study, only 3 out of 40 peptides (7.5%) exhibited activity. Notably, these peptides were among the weakest performers in their dataset and were also cytotoxic to control human cell lines, indicating a lack of selective anticancer activity. These findings emphasize the extreme challenge of targeting MDA-MB-231 cells, making our decision to use this cell line for primary screening particularly stringent and meaningful.

Addressing the IC₅₀ Range concern:

The reviewer suggests that the IC₅₀ values of 7–93 μM for MDA-MB-231 cells are not particularly impressive. While we acknowledge that lower IC₅₀ values are generally desirable, it is important to place these results in the context of recent ACP literature, where similar potency values have been observed (Table below). It must also be noted that the IC₅₀ of our top 15 candidates are between 7.8μM and 23.9μM with two outliers reported at 41.3μM and 93.1μM.

Table. IC₅₀ range of most promising candidates reported in recent studies.

Cancer Cell Line(s)	IC ₅₀ range (μM)	# peptides	Reference	Comments	Year
MDA-MB231	20 - 24	1	Hilchie et. al ⁹	After 24 h incubation	2017
B16F10, BxPC-3, K562, PANC-1, HeLa, HCT-116, MIA-PaCa-2, 4T1, HepG2, U251, PC12, A549	3.91 - >50	10	Yue et. al ¹³	After 36 h incubation. Activity highest in K562 colon cancer cells. (one generally cytotoxic peptide sPep2 excluded)	2025
MDA-MB231	7.39	1	Law et. al ¹⁶	After 48h incubation. After 24h IC ₅₀ was 64 μM	2023
MCF-7	35.34	1	Velayutham et al. ¹⁷	IC ₅₀ value of 35 μM, at 24 h	2023
HepG2	18.75	1	Velayutham et al. ¹⁸	After 24 hr, IC ₅₀ is 21 μM and after 48 hr, 18.75 μM	2022
HeLa, MCF7, MDA-MB-231	7.8-19	4	Zakharova et al. ¹⁴	After 72 h incubation, & MCF-10A 12-20 μM	2022
MDA-MB-231, BT-20, BT-474, SKBR3	3.5 - >100	6	Oliveira et al. ¹⁹	After 24 h incubation	2021
MDA-MB-231, MCF-7, TamR3	7.8 - 25	15	Ours	After 24h incubation. 7.8 μM to 23.9 μM (2 notable outliers 41.3 & 93.1 μM).	2025

The above table illustrates that our IC₅₀ values (7.8–23.9 μM for most candidates, with outliers at 41.3 μM and 93.1 μM) are well within the expected range for ACPs, despite our use of a very challenging cancer line¹⁵. Furthermore, peptide-based drugs often exhibit distinct pharmacokinetics, stability, and mechanisms of action compared to small molecules, meaning that IC₅₀ alone does not fully determine their therapeutic potential. Many approved peptide therapeutics have IC₅₀ values in the low-to-mid micromolar range but achieve strong *in vivo* efficacy due to targeted delivery, protease resistance, and/or improved bioavailability.

We acknowledge that further optimization is required to enhance potency. Future work may explore strategies such as iterative modeling (including successful peptides in further rounds of ML), sequence modifications (e.g., D-amino acid substitutions for protease resistance) and cyclization to enhance stability and target affinity, but this is beyond the scope of the present work.

In response to reviewer's concern, we have now added the above discussion in **Appendix H4 (SI pg 40-42)** and **Table H1**.

12. *Further (this result is hidden in the appendix Fig. H2 but should be emphasized and discussed in the main text), the cytotoxicities of the 3 ultimate "anticancer peptides" were not genuinely cancer-selective (approx. same ic50 value against non-cancerous MCF10A for all three screening effort "winners"). Together with the incoherence of the lytic activities (PBMC and RBC data differ significantly, Fig.7)*

Response: We appreciate the reviewer's keen observations regarding the selectivity of the identified ACPs and the differences in lytic activity across PBMCs and RBCs. In response to this concern, we have now moved the Supplemental Fig H2 to main text as **Figure 8** and added a discussion of these results in **section "Novel peptides selectively inhibited the viability of MDA-MB-231 breast cancer cells" (P27 L8 – P29 L14)**.

Regarding selectivity, we acknowledge that the identified ACPs did not exhibit strong cancer specificity against MCF10A cells. However, it is important to clarify that our model was trained using MDA-MB-231 and PBMC data, and its primary objective was to identify peptides with cytotoxic activity against this challenging cancer cell line while minimizing toxicity to immune cells, as has been previously done in case of Mastoparan⁹. Beyond MDA-MB-231 and PBMC cells, the evaluation of these peptides against other cancerous cells (e.g. MCF7 and TamR3) and non-cancerous cells (MCF10A and RBCs) was conducted as an exploratory analysis to assess whether the top-ranked candidates could extend beyond the training distribution. While none of the peptides demonstrated an ideal therapeutic window in this expanded evaluation, their discovery provides valuable insights into sequence-function relationships that can inform future efforts to optimize selectivity.

We agree with the reviewer that this represents an important caveat of our current approach. But it is also important to note that MDA-MB-231 (triple-negative breast cancer), MCF7 (hormone receptor-positive breast cancer), and TamR3 (tamoxifen-resistant breast cancer) represent very challenging lines due to their distinct resistance mechanisms and aggressive phenotypes. The fact that our identified ACP candidates exhibited cytotoxic effects against these difficult-to-treat cancer cell lines is encouraging and suggests their potential as lead compounds

for further optimization. As such, future work should focus on fine-tuning these peptides through rational design or AI-driven strategies to improve their selectivity, and therapeutic index, while reducing off-target toxicity. Importantly, our findings demonstrate that our pipeline can generate diverse ACP sequences that differ from those in the training set, thus providing a foundation for iterative improvements in peptide design.

13. *these entire results challenge the whole idea that a universal "anticancer" selectivity can be shown (and used for peptide classification) by comparing activity against one adherent cell line vs activity in a hemolytic or leucolytic experiment, which is the core endpoint in the mastoparan-optimization and is implemented here for ACP/non-ACP classification.*

Response: We appreciate the reviewer's insightful comment and would like to clarify that our study does not claim that a universal anticancer peptide (ACP) selectivity paradigm exists. In fact, we share the reviewer's skepticism and actively challenge the feasibility of universally selective ACPs, since ACP efficacy is largely dictated by cell-type-specific biological factors. Our approach does not attempt to generalize selectivity across all cancer types but instead focuses on a well-defined and challenging breast cancer model while incorporating additional datasets to improve predictive accuracy.

Addressing the concern of universal ACP selectivity

Anticancer selectivity in peptides is highly context-dependent, influenced by multiple factors including membrane composition, intracellular targets, proteolytic stability, and cellular uptake mechanisms. Cancer cells exhibit significant heterogeneity in lipid profiles, receptor expression, and metabolic activity, making broad-spectrum ACP classification challenging. For example, MDA-MB-231 TNBC cells present a unique surface charge distribution due to higher phosphatidylserine surface exposure, making them more susceptible to cationic ACPs, while some other cancer cell types, such as glioblastomas or leukemias, could have different membrane fluidity, lipid organization, or resistance mechanisms, influencing ACP potency and selectivity. While hemolytic (erythrocytes) and leucolytic (PBMC) assays are widely used to assess peptide selectivity^{9,14,19,20}, they are not comprehensive predictors of ACP activity across all cancer types because normal blood cells have distinct membrane properties compared to malignant cells.

Given these complexities, we agree with the reviewer that selectivity testing in a single adherent cancer cell line is not exhaustive. To strengthen our conclusions, we conducted additional selectivity validation experiments in MDA-MB-231 (TNBC, high metastatic potential), MCF7 (ER+, low metastatic potential breast cancer), TamR3 (tamoxifen-resistant breast cancer model), MCF10A (non-tumorigenic mammary epithelial cells, used as a control for toxicity assessment).

These additional studies allowed us to further explore the specificity of ACPs beyond MDA-MB-231 and assess their therapeutic relevance in a broader breast cancer context. Our findings suggest that ACP activity across MDA-MB-231 vs. hemolytic or leukolytic assays provides a useful framework for initial ACP candidate selection, but additional refinements in peptide design and selectivity optimization are necessary, which we intend to address in a future study.

Clarification on ACP/ non-ACP classification

The CancerPPD dataset has been widely used for ACP classification; however, it presents significant heterogeneity in terms of cell line variability, assay conditions, and peptide concentrations, which can introduce systematic biases in model training. While many studies have relied on CancerPPD-based ACP/non-ACP classification models, our objective was to examine and develop a more targeted model by focusing on Mastoparan analogues with well-defined experimental measurements of % cell inhibition in MDA-MB-231 cells (our primary model system). It must be noted that the CancerPPD classification model has no bearing on hemolytic activity against PBMCs for cytotoxicity evaluation.

This $D_{\text{Mastoparan}}$ dataset allowed us to construct a regression model specifically optimized for ACP potency prediction in MDA-MB-231 cells, providing a more reliable cell-line-specific prediction framework than a broad classification model alone. While we prioritized Mastoparan-based regression modeling for cell-line-specific potency prediction, we did not entirely exclude CancerPPD. ACPs across different cancer types share certain conserved physicochemical and structural features, such as: amphipathicity and net charge, which influence membrane disruption, secondary structure preferences, such as α -helicity, which contribute to ACP potency, and hydrophobicity, which affects peptide penetration into lipid bilayers.

To leverage these shared properties, we trained an ACP classification model using CancerPPD to capture broad-spectrum ACP characteristics. This classification model was then combined with the MDA-MB-231 regression model in an ensemble approach, which allowed us to learn generalized ACP features from CancerPPD and improve overall predictive accuracy by integrating both generalized and targeted information.

In response to reviewer's concerns, we have now added the above discussion on selectivity considerations in **Appendix H5 (SI pg. 43-44)** of the revised supplemental information.

14. *Therefore, something appears conceptually wrong with the selection criteria or TARSA training.*

Response: We appreciate the reviewer's concern and would like to clarify that the selection criteria for TARSA screening were conceptually grounded in well-established principles of ACP structure-activity relationships. The preference for helical structures in ACPs is extensively

documented in the literature, and our screening approach was explicitly designed to align with these findings. Hence, we screen peptides with putative helical structure as provided by DSSP, consistent with earlier studies²¹. For example, Lerksuthirat et al.²⁰, showed that out of their top-25 experimentally validated peptides, 19 were predicted to have α -helical conformation (ACP-preferred structure). This reasoning is now presented in **Results section**. In particular, we have added the following at **P5 L23- P6 L4**

“Helical peptides are the most extensively studied structural class of ACPs^{22–26}, and shorter peptides are preferred due to cost-effective synthesis. Therefore, the presented approach prioritized the discovery of short helical ACP candidates by screening peptides with putative helical structure as provided by DSSP, consistent with earlier works”

15. *It is not clear also why for the validation from the 110K candidates of the PDB pot 95. In contrast, of the 2.8M candidates originating from the GEN pot, only 10 were selected for experimental validation. The origin (PDB/GEN) of peptides should have also been maintained throughout the validation steps. Could this be that none of the 10 (top-most!) GEN-originated sequences did work (the first "active" peptide has the number 11) and the whole PDBlarge screen study was initiated due to this failure?*

Response: We appreciate the reviewer’s concern regarding the discrepancy in the number of peptides selected for experimental validation from the PDB-derived (PDB pot) versus generative model-derived (GEN pot) candidates. We would like to clarify that the primary factor influencing this decision was the cost and feasibility of experimental validation.

Experimental validation of ACPs is costly and resource-intensive, which explains why many generative modeling studies lack experimental validation despite promising computational results. A recent study by Szymczak et. al⁴ in Nature Communications found that five out of 13 AMP-generative models did not validate their predictions, and none provided full code and data for reproducibility. For ACPs, experimental validation is even rarer. Unlike small molecules, which benefit from molecular docking as a screening proxy, ACPs, especially membranolytic ones, lack reliable *in silico* predictors of activity. To mitigate this challenge, virtual screening was prioritized to increase confidence before wet-lab validation. PDB-derived peptides (PDB pot) offered biological context, enhancing their likelihood of success, whereas purely generative sequences (GEN pot) lacked such grounding, making large-scale experimental validation financially unfeasible. This selection strategy does not imply generative models failed but reflects a pragmatic allocation of resources. Interestingly, in the comprehensive study by Yue et al. that the reviewer also suggested, the authors examined 3.8 million sequences from UniProt, compared to only 100,000 sequences from a generative model.

We acknowledge that this rationale was not explicitly discussed and appreciate the reviewer raising this concern. To address this, we have added a detailed discussion on this in **Appendix H1 (SI pg. 36-37)**.

16. *Next, the authors should know that the (protein) PDB dataset is redundant and is biased towards structures of soluble (non-membrane) proteins and that short polypeptides are "conformationally plastic" in the sense that their conformation vastly depends on the environment. Hence, there are no environment-independent "intrinsically helical" sequences - even their benchmark mastoparan (as the authors showed previously) is a random coil in aqueous buffers and only folds helically in the presence of membrane mimics. In contrast, for peptides (protein fragments) from the PDB, a helical fold would be dictated by proteinous neighbourhood and (often) crystal packing conditions (dominating X-ray-derived structures). So, the consequent problem would be to adequately implement peptide descriptors into QSAR (e.g. hydrophobic moment would be different per unstructured and helically-folded peptide).*

Response: We appreciate the reviewer's insights and acknowledge the structural biases in the PDB and the conformational plasticity of short peptides. The assumption of "helicity" was a necessary simplification made to drive our discovery pipeline, focusing on the peptides' active conformation upon membrane interaction rather than their aqueous state. This is a reasonable assumption, as many ACPs are known to adopt amphipathic structures when they interact with biological membranes. Therefore, if their helical conformation within a globular protein were amphipathic, then this would likely be mimicked in a resulting membrane bound ACP as well. For e.g., the Temporin family of AMPs and ACPs, have a tendency to adopt an amphipathic α -helical conformation in a membranemimetic solvent, such as 50% trifluoroethanol (TFE)/water, whereas they exist in a random coil conformation in water²⁷. Hence, PDB-derived sequences were prioritized for their evolutionary relevance, not for a fixed helical conformation.

To address the reviewer's concern, we have retrospectively analyzed 20 random experimentally validated PDB peptides, finding that 15/20 had >60% predicted helicity with >70% confidence using an expert jury of four secondary structure prediction tools. For the remaining five, Alphafold3 and JPred confirmed helical tendencies with high confidence. We have now added **Appendix A2 (SI pg.2)** to the updated manuscript to discuss this concern in detail and provide secondary structure prediction for these 20 peptides as **supplemental materials (helicity.doc)**.

17. *In contrast to small molecules, for peptides, good descriptors should be the 4D or 3D ones. This problem has to be discussed in the pre-validation part of the manuscript.*

Response: We appreciate the reviewer's insightful comment on the significance of 3D and 4D molecular descriptors in peptide characterization and have used similar tactics ourselves in the

past. We agree that these descriptors offer valuable insights into peptide conformation, flexibility, solvent interactions, and binding mechanisms, but the computational cost to compute them for a very large peptide libraries outweigh their benefits, making them non-optimal for large virtual screening campaigns. Infact, we had experimented with 3D inductive descriptors in section “Descriptor Relevance in QSAR Modeling for Peptide Potency”. While they gave respectable predictive accuracy for %-cell inhibition regression task ($\rho = 0.65$), it was estimated that it would take nearly 68 years to compute these descriptors for a 36 million peptide library.

In response, we have introduced this problem in **section “Descriptor Relevance in QSAR Modeling for Peptide Potency” (P14 L1-7)** and added a new **Appendix A4 (SI pg 3-4)** discussing it in detail.

18. *To the same problem - it is not clear why 12mers were selected as a target - as far as I know, mastoparan is a 13mer (by the way, which out of 40 known mastoparans was used should be defined)- and whether on the TARSA stages, the peptides for the training/evaluation possessed free N/C-termini. Mastoparan has a charged N-term but carries C-amidation (activity profile changes when the terminus is nonprotected, as the authors previously showed). Fragments of proteins in the PDB database would be terminally not charged (termini would, in most cases, participate in the peptide bonds*

Response: We appreciate the reviewer’s comments and clarify that 12-mers were selected for standardization in high-throughput synthesis, ensuring uniform peptide length within the spot-array method. This choice also facilitated exploration beyond the 14-mer Mastoparan analogs while preserving key amphipathic helical features. Additionally, 12-mers strike a balance between structural relevance, synthetic accessibility, and cost-effectiveness, making them ideal for initial discovery. Notably, the 12-mer bovine bactenecin is the smallest known cationic AMP in nature, and is sufficient to maintain at least 2 alpha helical turns in membrane like environments²⁸. We have added **Appendix A3 (SI pg. 3)** to discuss “Rationale for Selecting 12-Mer Peptides” in detail.

Terminal Modifications and Training Data

The reviewer raises an excellent point regarding the influence of terminal modifications on activity. We agree that many bioactive peptides, including Mastoparan and related ACPs/AMPs, feature C-terminal amidation or N-terminal acetylation, which affect stability, charge distribution, and interactions with biological membranes.

Our QSAR model was trained on Mastoparan-NH₂ (INLKALAALAKKIL-NH₂), which has a free N-terminus and an amidated C-terminus. In our experience, N-terminal amidation is crucial for

biological activity. Thus, all synthesized peptides retained this amidation for consistency with predictions. However, TARSA-screened peptide libraries considered only amino acid sequences, without terminal charges although there is also not a free C-terminus since many peptides are incorporated into larger protein structures. We have clarified these details in the manuscript in methods section “Wet-Laboratory Characterization”(P40 L6-10) and Appendix J2 (SI pg 48).

Justification for PDB-Derived Peptides

We acknowledge the reviewer’s concern that peptides derived from PDB fragments generally lack free termini due to their incorporation into larger protein structures. However, the primary motivation for including PDB-derived sequences in our screening library was not to preserve terminal charge states, but rather to ensure that our dataset was enriched with structurally validated α -helical scaffolds known to be stable in aqueous or physicochemically relevant helices that share key properties with natural ACPs, such as amphipathicity and cationic charge distribution.

While the termini in PDB fragments are often constrained by peptide bonds, our deep learning model primarily learns from intrinsic sequence features such as amino acid composition, helicity, charge, and hydrophobic moment, rather than explicitly modeling free termini. This aligns with prior studies in peptide design, where secondary structure formation and membrane interaction properties often play a more dominant role than terminal modifications in defining activity.

References

1. Abe, T., Kelly Buchanan, E., Pleiss, G., Zemel, R. & Cunningham, J. P. Deep Ensembles Work, But Are They Necessary?
2. Dietterich, T. G. Ensemble Methods in Machine Learning. *Lect. Notes Comput. Sci. (including Subser. Lect. Notes Artif. Intell. Lect. Notes Bioinformatics)* **1857 LNCS**, 1–15 (2000).
3. Hao, Y., Lin, Y., Zou, D. & Zhang, T. On the Benefits of Over-parameterization for Out-of-Distribution Generalization.
4. Szymczak, P. *et al.* Discovering highly potent antimicrobial peptides with deep generative model HydrAMP. *Nat. Commun.* **2023 141 14**, 1–23 (2023).
5. Touw, W. G. *et al.* A series of PDB-related databanks for everyday needs. *Nucleic Acids Res.* **43**, D364–D368 (2015).
6. Kabsch, W. & Sander, C. Dictionary of protein secondary structure: pattern recognition of hydrogen-bonded and geometrical features. *Biopolymers* **22**, 2577–2637 (1983).

7. Saravanan, V. & Lakshmi, P. T. V. ACP: A web server for prediction and design of anti-cancer peptides. *Int. J. Pept. Res. Ther.* **21**, 99–106 (2015).
8. Rives, A. *et al.* Biological structure and function emerge from scaling unsupervised learning to 250 million protein sequences. *Proc. Natl. Acad. Sci. U. S. A.* **118**, e2016239118 (2021).
9. Hilchie, A. L. *et al.* Mastoparan is a membranolytic anti-cancer peptide that works synergistically with gemcitabine in a mouse model of mammary carcinoma. *Biochim. Biophys. Acta (BBA)-Biomembranes* **1858**, 3195–3204 (2016).
10. Grisoni, F. *et al.* Designing anticancer peptides by constructive machine learning. *ChemMedChem* **13**, 1300–1302 (2018).
11. Huang, J. *et al.* Identification of potent antimicrobial peptides via a machine-learning pipeline that mines the entire space of peptide sequences. *Nat. Biomed. Eng.* **2023** *7*, 797–810 (2023).
12. Zhang, J. *et al.* Large-scale screening of antifungal peptides based on quantitative structure–activity Relationship. *ACS Med. Chem. Lett.* **13**, 99–104 (2021).
13. Yue, J. *et al.* Discovery of anticancer peptides from natural and generated sequences using deep learning. *Int. J. Biol. Macromol.* **290**, 138880 (2025).
14. Zakharova, E., Orsi, M., Capecchi, A. & Reymond, J. L. Machine Learning Guided Discovery of Non-Hemolytic Membrane Disruptive Anticancer Peptides. *ChemMedChem* **17**, e202200291 (2022).
15. Ma, Y. *et al.* Efficient Mining of Anticancer Peptides from Gut Metagenome. *Adv. Sci.* **10**, 2300107 (2023).
16. Law, D. *et al.* In silico identification and in vitro assessment of a potential anti-breast cancer activity of antimicrobial peptide retrieved from the ATMP1 *Anabas testudineus* fish peptide. *PeerJ* **11**, e15651 (2023).
17. Velayutham, M. *et al.* Aquatic Peptide: The Potential Anti-Cancer and Anti-Microbial Activity of GE18 Derived from Pathogenic Fungus *Aphanomyces invadans*. *Molecules* **28**, 6746 (2023).
18. Velayutham, M. *et al.* Anti-Cancer and Anti-Inflammatory Activities of a Short Molecule, PS14 Derived from the Virulent Cellulose Binding Domain of *Aphanomyces invadans*, on Human Laryngeal Epithelial Cells and an In Vivo Zebrafish Embryo Model. *Molecules* **27**, 7333 (2022).
19. Oliveira, F. D. *et al.* The antimetastatic breast cancer activity of the viral protein-derived peptide vCPP2319 as revealed by cellular biomechanics. *FEBS J.* **289**, 1603–1624 (2022).
20. Lerksuthirat, T. *et al.* ALA-A2 Is a Novel Anticancer Peptide Inspired by Alpha-

- Lactalbumin: A Discovery from a Computational Peptide Library, In Silico Anticancer Peptide Screening and In Vitro Experimental Validation. *Glob. Challenges* **7**, 2200213 (2023).
21. Sofi, M. A. & Wani, M. A. IRNN-SS. *Int. J. Bioinform. Res. Appl.* **20**, 608–626 (2024).
 22. Uggerhøj, L. E. *et al.* Rational Design of Alpha-Helical Antimicrobial Peptides: Do's and Don'ts. *ChemBioChem* **16**, 242–253 (2015).
 23. Huang, Y. *et al.* Role of helicity of α -helical antimicrobial peptides to improve specificity. *Protein Cell* **5**, 631–642 (2014).
 24. Pan, F. *et al.* Anticancer effect of rationally designed α -helical amphiphilic peptides. *Colloids Surfaces B Biointerfaces* **220**, 112841 (2022).
 25. Huang, Y., Feng, Q., Yan, Q., Hao, X. & Chen, Y. Alpha-Helical Cationic Anticancer Peptides: A Promising Candidate for Novel Anticancer Drugs. *Mini-Reviews Med. Chem.* **15**, 73–81 (2015).
 26. Hadianamrei, R., Tomeh, M. A., Brown, S., Wang, J. & Zhao, X. Rationally designed short cationic α -helical peptides with selective anticancer activity. *J. Colloid Interface Sci.* **607**, 488–501 (2022).
 27. Wade, D. *et al.* Antibacterial activities of temporin A analogs. *FEBS Lett.* **479**, 6–9 (2000).
 28. Wieczorek, M. *et al.* Structural Studies of a Peptide with Immune Modulating and Direct Antimicrobial Activity. *Chem. Biol.* **17**, 970–980 (2010).

The original comments are provided in *italics*. Our responses in **blue plain type**. Inserts into text are underlined. Specific mentions of updated sections are **blue bolded**.

Reviewer 3

1. *This manuscript will be better received in specific Drug Discovery or Bioinformatics communities in a more specialised journal - the view that revision failed to change. The major issues with this manuscript remained after this revision.*

Namely, their virtual screening study's stated primary object (training, screening, and validation) are "anticancer peptides" (ACP). However, how the authors introduce and define peptide sequences (features) they are searching for is, to my regret, incapable of achieving the desired activity. Amphipathic, cationic helical peptides targeting the plasma membrane are a marginal (and, indeed, very well-studied: hence, the conclusion) type of bioactive peptides if one looks at them from the perspective of the "anticancer" function. Successful (i.e. highly potent, highly selective, biostable and bioavailable in vivo) are the peptides that are conformationally non-helical and possess proteins, not lipid membranes, as the targets in the primary MoA. I suggest the authors to redefine the peptides they are searching for as "mastoparan-like" or "MDA-MB-231 cytotoxic" or alike - but avoid generalisation of searching for an (universal) ACP. Without this detailisation, the readership will be misled, and this is better to avoid from the scientific rigour perspective.

Response: We strongly disagree with the assertion that our study is misplaced in scope or impact, or that our use of the term *anticancer peptides (ACPs)* is misleading or scientifically inappropriate. The term ACP is broadly adopted in the literature to describe peptides that exhibit any measurable inhibitory activity against any cancer cell line, regardless of their precise mechanism of action or structural conformation. As indicated before, several peer-reviewed studies report the peptides as “anticancer peptides” based solely on their ability to kill cancer cells, even if these peptides have limited selectivity – non broad-spectrum toxicity towards a wide range of cancer types i.e. some peptides with activity against even a single cancer cell line would still be named as an anticancer peptide. All training peptide sequences used in our study were sourced from well-established, peer-reviewed ACP databases (e.g., CancerPPD, ACP Mastoparan analogues), each of which classifies peptides as ACPs based on empirical evidence of anticancer activity. There is, to our knowledge, no formal subclassification system in place—such as “MDA-MB-231-specific peptides” or “mastoparan-like ACPs”—that has supplanted this standard terminology. We respectfully note that there is no known “MDA-MB-231 peptide toxicity database,” nor has any nomenclatural convention been adopted across the community to support such restrictive classifications.

Regarding the broader claim that membrane-active peptides represent only a marginal and less relevant subclass of ACPs, we would contend that the substantial scientific literature on this subject and the continued research interest from numerous research groups around the world

would refute this statement. Rational ACP design studies, such as by Pan et al.¹, focus on amphipathic, helical peptides targeting the plasma membrane. As highlighted in our previous response, most other reported ACPs in literature are indeed helical peptides. Huang et al.² present an extensive review on *Alpha-helical cationic anticancer peptides as a promising candidate for novel anticancer drugs*.

Additionally, regarding the comment on protein targeted mechanism of action, we would like to point the reviewer to the following comment by Gaspar et al.³

“the mechanism and selectivity criteria by which ACPs kill cancerous cells is still a controversial theme although some major conclusions can be outlined.”

One such conclusion is that many ACPs have a membranolytic mechanism of action leading to necrosis. Several such peptides are listed in the tables below (see Table from Zhang et al.⁴). While protein-targeting peptides represent an important direction of research, membranolytic ACPs have also demonstrated selective activity and therapeutic relevance, as supported by extensive studies. In fact, to the best of our knowledge, even until recently, the mechanism of action of many selective ACPs remained an open research question as pointed out by Pan et al. in section 3.1 of their work¹. They comment that the *“mechanism by which these parameters (amphiphilicity, charge density, hydrophobicity) affect the biological activities of the AMPs/ACPs is still subject to further clarification.”*

Table 1. Peer-reviewed studies with membranolytic mechanism of action for ACPs.

ACP	Cancer Type	Mechanism
LL-37	Human oral squamous carcinoma cells	Toroidal pore mechanism
α -Defensins	Human myeloid leukemia cell line (U937)	Cytolytic activity
β -Defensin-3	HeLa, Jurkat, and U937 cancer cell lines	Binding to cell membrane to cause cytolysis
Bovine Lactoferricin	Drug-resistant and drug-sensitive cancer cells	Cytolysis and immunogenicity
Gomesin	Murine and human cancer cell lines, including melanoma and leukemia	Carpet model for membrane destruction
Cecropin B1	NSCLC cell line	Tumor growth inhibition via pore formation and apoptosis
Magainin 2	Human lung cancer cells (A549)	Pore formation on cell membranes
Brevinin	Lung cancer (H460), melanoma cells, glioblastoma (U251MG), and colon cancer (HCT116) cell lines	Penetration into the lipid bilayer causing cell death
Phylloseptin-PHa	Breast cancer cells (MCF-7) and breast epithelial cells (MCF10A)	Penetration into the lipid bilayer causing cell death
Dermaseptins	Prostate cancer cells (PC-3)	Pore formation on the lipid bilayer

Chrysohsin-1, -2, and -3	Human fibrosarcoma (HT-1080), histiocytic lymphoma (U937), and cervical carcinoma (HeLa) cell lines	Disruption of the plasma membrane
---	-----------------------------------

AMP/ACP Name	Tumor Target	Mechanism
BMAP-28	HTC	MP/Ca influx
CA-MA-2	STC	MP
Cecropin A	HTC	MP
D-K6L9	STC	MP
Gomesin	STC	MP
KLA	STC	MP
LL37	Ovarian CA	MP
LTX-315	HTC/STC	MP/ICD
Melittin	STC	MP
MG2B	MCF-7	MP
Pardaxin	STC	MP

Table 2: Representative anticancer peptides reported to act primarily through a membrane-permeabilizing (MP) mechanism of action. HTC: hematological tumor cells; STC: solid tumor cells. Adapted from Deslouches et al.⁵

We further note that our study provides preliminary insights into the membrane interaction behavior of selected peptides through all-atom MD simulations. While a full mechanistic elucidation is beyond the scope of the present work, we believe that our findings contribute meaningfully to the growing body of evidence supporting membrane-targeting ACPs.

It is also worth reiterating that our primary contribution in this work is the development of a novel scalable reinforcement learning algorithm that enables fast screening of large peptide libraries, which application we show for a demonstrative problem of ACP discovery. To this effect, the effectiveness and generalizability of the algorithm itself are agnostic to the particular biological mechanism involved. If the reviewer can make available datasets for non-helical, non-membranolytic peptides, we would be happy to apply our framework to such targets to demonstrate its versatility.

- The next remaining problem is a poor "wet lab" validation methodology. As seen from the methods - the authors purchased both the SPOT and SPPS-produced peptides. I come from the peptide synthesis field and can assure the editor that doubling any of the libraries or (in a number of tested peptides) or extending the length of the peptides beyond 12mers (up to 20-25mers, for instance), economically would not be a significant burden as both technologies are pretty developed as of now. (I would not really like to stress here the problems with concentration determination with NanoDrop for sequences w/o Trp, inconsistency of the cell assays in different media,*

different amounts of FBS, different cell numbers, differences in adherence, different readouts (LDH release, oxidative stress-sensitive metabolic resazurin transformation, Hb-release), different peptide-exposure times, etc. All these cavities should be well-known to the authors. All disallow any robust conclusion about selectivity if comparing RBC, PBMC, and MDA-MB-231, which are left without critical discussion).

Response: We agree that peptide synthesis technologies both SPOT and SPPS have matured significantly, and we appreciate the reviewer's expertise in this domain. We would like to emphasize to the reviewer and the editor that the objective of the wet lab assessment in this study was to demonstrate the utility of our proposed computational pipeline, rather than to perform a broad analysis of anticancer peptide sequences. The SPOT array enabled a low-cost method to identify "hits" from the initial peptide screen and this activity was confirmed with 95% pure peptide samples, which is standard practice in the field. It is important to note that we did not set out to solve the anticancer peptide debate, nor did we make any such bold claim in our manuscript.

As an academic group with limited financial resources, we made careful choices to balance cost, feasibility, and scientific value. Our SPOT array included 111 peptides from which we selected 15 representative peptides for SPPS and repurchased 3 top hits for further testing. While we could have in principle made more peptides, we would argue that what we did represents substantial validation of our computational methods and strategies.

Regarding the limitations of peptide arrays, we have adopted strategies to account for peptides that lack a tryptophan by using the weighted average yield of peptide across the entire peptide array. This only occurred for about 8% of the spots, so it did not affect the results overall.

In summary, we clarify that the decision to screen the numbers of peptides we did was not based on scientific reluctance but on pragmatic constraints. We emphasize that our study aims to demonstrate the *feasibility of a scalable in silico screening strategy*, and that the biological assays are intended as an initial validation—*not* as a comprehensive pharmacological or toxicological characterization. We view this work as a good first step, and future studies should certainly include larger peptide libraries, longer sequences, and more sophisticated in vitro and in vivo models. However, such studies are more efficiently conducted in an iterative fashion, continually learning from improved peptides such as the ones presented in our study.

- 3. Further, I insist - in the current form of the manuscript, the in silico (CG and MD) investigations are useless - they bring no contribution to the workflow (or such impact is not clearly described), do not affect selection and do not influence the conclusions and, therefore, could be readily removed or shifted to SI with no damage to the manuscript narrative.*

Response: As outlined in our previous response about MD to the reviewer, the MD simulations were not designed to influence peptide selection or act as an additional filtering step. Rather, they were incorporated into the study for a different but important reason i.e. to serve as an independent validation of the mechanistic plausibility of our computational discovery pipeline prior to experimental testing. Specifically, we have stated the following on **P21 L17-19**:

“Coarse-grained molecular dynamics (CG-MD) simulations were performed as an independent validation of mechanistic plausibility of the peptides selected by our computational pipeline, prior to the experimental testing.”

We hypothesized that membranolytic anticancer peptides identified through our method would exhibit higher membrane contact ratios and greater penetration depths into a simulated plasma membrane, compared to non-cytotoxic peptides, when assessed via coarse-grained MD simulations.

The observed results supported this hypothesis, thus providing critical confirmation that our discovery pipeline was selecting peptides with the expected membrane-disruptive behaviors associated with anticancer activity. This hypothesis and corroborating results are spelled out on **P21 L19- P22 L04**. Specifically, we state:

“Drawing on findings by Das et al., which demonstrated that membranolytic antimicrobial peptides exhibit greater membrane contact and deeper insertion into lipid bilayers than non-cytotoxic counterparts, this hypothesis was extended to the membranolytic ACPs discovered by TARSA. The CG-MD simulations confirmed that several the discovered motifs engaged in effective interactions with lipid bilayer membranes, consistent with a membranolytic mode of action characteristic of potent ACPs. Notably, peptides exhibiting high membrane contact densities and greater penetration depths displayed interaction patterns similar to those of experimentally validated breast cancer inhibitors (Figure G1-A).”

This information is far from “useless” and demonstrates the potential of our pipeline to be applied to large-scale ACP discovery.

A more detailed discussion on these results is provided in **Supplemental section G: P32-37**.

Importantly, had the MD results contradicted our hypothesis, we would have revisited and refined our discovery approach before proceeding to the resource-intensive stages of high-throughput in silico screening and experimental validation. Thus, the MD analysis, even though it did not alter the final selection, acted as a necessary quality control step, ensuring the scientific rigor of our computational strategy. Given this, we believe that the MD investigations add value to the manuscript by demonstrating the biophysical plausibility of the computationally predicted peptides, and contribute to the overall credibility of the discovery workflow.

We have already highlighted the utility of MD simulations in the overall computational workflow. In particular, on **P22 L4-7**, we have stated the following:

“This alignment between CG-MD results and experimental observations provided critical support for the predictive accuracy of TARSA, reinforcing its utility for discovering viable ACP candidates prior to its application for costly high throughput virtual screening and experimental validation.”

Furthermore, in addition to the coarse-grained simulations, we also conducted all-atom MD simulations on our top three discovered and experimentally validated ACPs. These atomistic simulations provided valuable insights into a probable mechanism of action of these peptides. As indicated above, since the precise membrane-disruptive mechanisms of ACPs remain an open research question, such simulations contribute important mechanistic understanding that is otherwise difficult to obtain experimentally. It is well established in the field that MD simulations

are routinely employed to investigate mechanistic hypotheses that are challenging to address through experimental techniques alone⁶⁻⁸.

Finally, we wish to highlight that **another reviewer specifically emphasized the importance of the MD component of this work** and encouraged us to significantly expand these analyses to strengthen the manuscript. We have since incorporated additional simulations and detailed analyses as recommended.

In light of these points, we assert that the MD investigations are not only scientifically justified but are integral to the rigor and credibility of our study. Shifting these analyses to the Supplementary Information would diminish an important mechanistic validation step that underscores the biological relevance of the computational pipeline.

To emphasize the impact of MD investigations, we have updated the section on ***“In silico assessment of discovered anticancer peptide motifs”*** to include a more detailed discussion on the impact of CG-MD on the overall workflow. In particular, **P21 L17 – P22 L07** say the following

“Coarse-grained molecular dynamics (CG-MD) simulations were performed as an independent validation of mechanistic plausibility of the peptides selected by our computational pipeline, prior to experimental testing. Drawing on findings by Das et al., which demonstrated that membranolytic antimicrobial peptides exhibit greater membrane contact and deeper insertion into lipid bilayers than non-cytotoxic counterparts, this hypothesis was extended to the membranolytic ACPs discovered by TARSA. The CG-MD simulations confirmed that several the discovered motifs engaged in effective interactions with lipid bilayer membranes, consistent with a membranolytic mode of action characteristic of potent ACPs. Notably, peptides exhibiting high membrane contact densities and greater penetration depths displayed interaction patterns similar to those of experimentally validated breast cancer inhibitors (Figure G1-A). This alignment between CG-MD results and experimental observations provided critical support for the predictive accuracy of TARSA, reinforcing its utility for discovering viable ACP candidates prior to its application for costly high throughput virtual screening and experimental validation.”

4. Next, at least from the text (bioinformatic part), it is unclear what is meant by %potency for the selected sequences. If IC50 to PBMC (<15%) and to MDA-MB-231 (>50%, Fig. 5A-C) was meant, then the calculated values must be numerically compared to experimental data in the validation effort.

Response: We thank the reviewer for highlighting this important point regarding the clarity of the term "% potency" In our study, "% potency" referred specifically to the predicted percentage of cell inhibition, as output by our regression model. To clarify, the regression model was trained on an in-house dataset comprising 210 Mastoparan derivatives, where the experimental endpoints were the measured percentage of cell inhibition against MDA-MB-231 breast cancer cells and PBMCs, respectively. These values were obtained from SPOT-synthesized peptide arrays followed by experimental assessment of cytotoxicity. When selecting the peptides for experimental validation, we chose those peptides that had the predicted % cell inhibition values > 40% towards MDA-MB-231 while simultaneously having a predicted % cell inhibition value < 15% towards PBMC.

In response to the reviewer's suggestion, we have revised the manuscript to more explicitly define this terminology. We have replaced the term "potency" with "predicted % cell inhibition" where appropriate.

5. Still, I would like to know the dataset identity (PDB12mer or GEN12mer origin) in the validation step - the request that the authors preferred not to address.

Response: We apologize for this oversight and have now included a supplementary file, peptide_origin.xlsx, which specifies the origin of each of the 105 experimentally tested peptides, indicating whether they were derived from the PDB12mer or GEN12mer libraries.

REFERENCES

1. Pan, F. *et al.* Anticancer effect of rationally designed α -helical amphiphilic peptides. *Colloids Surf B Biointerfaces* **220**, 112841 (2022).
2. Huang, Y., Feng, Q., Yan, Q., Hao, X. & Chen, Y. Alpha-Helical Cationic Anticancer Peptides: A Promising Candidate for Novel Anticancer Drugs. *Mini-Reviews in Medicinal Chemistry* **15**, 73–81 (2015).
3. Gaspar, D., Salomé Veiga, A. & Castanho, M. A. R. B. From antimicrobial to anticancer peptides. A review. *Front Microbiol* **4**, (2013).
4. Zhang, Y., Wang, C., Zhang, W. & Li, X. Bioactive peptides for anticancer therapies. *Biomaterials Translational* **4**, 5 (2023).
5. Deslouches, B., Di, Y. P., Deslouches, B. & Peter Di, Y. Antimicrobial peptides with selective antitumor mechanisms: prospect for anticancer applications. *Oncotarget* **8**, 46635–46651 (2017).
6. Ruiz Munevar, M. J. *et al.* Cation Chloride Cotransporter NKCC1 Operates through a Rocking-Bundle Mechanism. *J Am Chem Soc* **146**, 552–566 (2024).
7. Kumar, A., Mishra, B., Konar, A. D., Mylonakis, E. & Basu, A. Molecular Dynamics Simulations Help Determine the Molecular Mechanisms of Lasioglossin-III and Its Variant Peptides' Membrane Interfacial Interactions. *Journal of Physical Chemistry B* **128**, 6049–6058 (2024).
8. Herce, H. D. & Garcia, A. E. Molecular dynamics simulations suggest a mechanism for translocation of the HIV-1 TAT peptide across lipid membranes. *Proc Natl Acad Sci U S A* **104**, 20805–20810 (2007).

The original comments are provided in *italics*. Our responses in **blue plain type**. Inserts into text are underlined.

Reviewer 3

- I regret to state that the authors did not address my concerns to my satisfaction. Hence, I cannot recommend the current revision for publication. I reiterate my suggestion to resubmit to an ML, peptide science, or drug discovery journal, where it would be more appreciated by the audience and find the paradigm coherence.*

Just reflecting on the response to my previous revision remarks:

The major, rather semantic, but fundamental problem remains, directly affecting the impact and the manuscript placement. Nature Communications is a highly ranked journal with a broad audience, broader than the author's field. However, the manuscript misleadingly states that the major physicochemical hallmarks of "an anticancer peptide" are features of "a membranolytic peptide" (by definition, these would be non-selective, lysing all types of membranes and incapable of discriminating between cancer/non-cancer cells). This is similar to having a cationic detergent, which, despite being active membranolytic, would have less of a pharmacological sense to develop as an anticancer/anti-tumour drug. Helical and membranolytic peptides are the least capable of advancing to clinical studies and approval (check the current pipelines and the 29 FDA approvals). This (non-helicity and other than membranolytic MoAs for the advanced ACP) should be stressed and critically discussed. I never doubted the existence of the term ACP, but I doubt that the authors' study/search is truly an ACP. Hence, I suggest calling within their manuscript the PepSce TARSA-screened target peptides and the final three sequences selected, not "potent ACPs"/"non-toxic towards healthy human cells", but rather "Mastoparan-like", "membranolytic helical" ", MDA-MB-231/PBMC-discriminating helical peptides". This concern remains a major conceptual problem of this manuscript.

Response: We thank the reviewer for this important critique. We recognize that this issue is central to how our work is framed for a broad audience of *Nature Communications*. We have had good experiences in publishing papers in *Nature Communications* and only submit our most distinctive work to this journal. Indeed, we consider that especially the computational method novelty and excellent practical outputs, merit strong consideration for this journal.

The vast majority of reviews on anti-cancer peptides include membranolytic peptides in this class, although we agree with the reviewer that we perhaps used somewhat ambiguous wording in describing this. We agree that many of the peptides we identified and validated likely act primarily through membranolytic mechanisms. This is consistent with the fact that our computational model was trained on mastoparan-derived sequences, which are known to be membranolytic. Furthermore, this work is aligned with numerous antimicrobial peptides (AMPs) models that we have previously published on, and where we were able to quantitatively capture SAR of membranolytic peptides [<https://pubs.acs.org/doi/abs/10.1021/jm8015365>; <https://pubs.acs.org/doi/abs/10.1021/cb800240j>; <https://onlinelibrary.wiley.com/doi/abs/10.1111/j.1747-0285.2010.01044.x>], which enabled us to develop extremely potent antimicrobial agents, some of which are in advanced clinical trials.

As the reviewer notes, in regard to anti-cancer peptides this particular mechanism differs fundamentally from those ACPs being considered for commercialization, which often display reduced helicity and alternative non-membranolytic modes of action. We acknowledge that membranolytic helices may be less likely to progress clinically due to their lack of selectivity across membrane types.

In response, we have carefully revised the manuscript to avoid overgeneralization. Specifically, throughout the manuscript, as requested by the reviewer, we have replaced references to “potent ACPs” and “non-toxic peptides” with more precise descriptors, as the reviewer suggested, namely “*mastoparan-like membranolytic helical peptides*” and “*MDA-MB-231/PBMC-discriminating helical peptides.*” These changes are tracked in the updated submission. However, some of the most notable changes to this effect are enumerated below

- Updated manuscript title to “*A Scalable Reinforcement Learning Approach for Screening Large Peptide Libraries for Bioactive Peptide Discovery*” to give clear indication of the scope and primary contribution of our work.”
- Replaced the term “novel ACPs” in the abstract with “novel membranolytic peptides”
- We now conclude the introduction P5 L18-21 by modifying our findings from “... cost-effective platform for identifying synthesizable ACPs with validated anti-cancer activity.” to “...cost-effective platform for identifying synthesizable peptides with validated membranolytic activity.”
- We now clarify to the reader that finding Mastoparan-like peptides is the demonstrative example we choose in this study. We do not claim to have discovered ACPs, but rather mastoparan-like peptides like the reviewer suggested. It’s noteworthy that mastoparan itself is an ACP. As such, we clarify on P6 L2-4 “As a demonstrative example, we sought ACP Mastoparan-like helical peptides with membranolytic activity.”
- P8 L13-14 instead of “with the goal of prioritizing the identification of ACPs” now reads “with the goal of prioritizing the identification of membranolytic candidates.”
- Modified Figure 2D (workflow) to now say bioactive peptides instead of ACPs.
- P11 L14-16 is updated from “... the effectiveness of various peptide representations for modeling ACPs” to “... the effectiveness of various peptide representations for modeling membranolytic Mastoparan-like helical peptides”
- P20 L13-15, changed “the discovered peptides exhibit characteristics consistent with known ACPs” to “the discovered peptides exhibit characteristics consistent with known membranolytic mastoparan-like peptides.”
- In Discussion P31 L11-14, we replaced “... cost-effective virtual screening of ultra-large peptide libraries ACPs.” to “... cost-effective virtual screening of ultra-large peptide libraries for helical, membranolytic mastoparan-like peptides with demonstrated activity towards MDA-MB231.”
- The conclusions (P34 L19-21) now say “TARSA approach enabled efficient exploration of a 36 million peptide library and identified regions of the chemical space enriched in Mastoparan-like membranolytic peptides.”

Further, in line with the reviewer’s recommendation, we now explicitly discuss in the Discussion that advanced ACPs in clinical pipelines are predominantly non-helical and non-membranolytic,

and we present this as a limitation of our current work. We also stress that while membranolytic helical peptides can still provide useful models for probing peptide–membrane interactions and validating computational discovery pipelines, further methodological advances will be needed to capture the broader diversity of ACP mechanisms relevant to clinical translation.

Specifically, we have added the following in the discussion (P34 L4-11)

It is also noteworthy that the most advanced ACPs in clinical development are predominantly non-helical and non-membranolytic, often acting through mechanisms distinct from direct membrane disruption. In contrast, the TARSA-identified peptides presented here represent mastoparan-like, membranolytic helical scaffolds that achieve selective cytotoxicity against MDA-MB-231 cells relative to PBMCs. While this provides a clear proof-of-concept for TARSA in recovering biologically active peptides from large-scale screening, subsequent studies should also leverage TARSA to explore libraries enriched in non-helical and non-membranolytic peptides, thereby aligning more closely with the structural and mechanistic hallmarks of clinically advanced ACPs.

Finally, and importantly, we would like to reiterate that the central contribution of our study is the development of the TARSA computational framework for efficient large-scale in silico screening of bioactive peptides. While in this work we applied it to mastoparan-like membranolytic helices, the framework itself is generalizable and could be used to explore other classes of bioactive peptides, including ACPs with non-membranolytic mechanisms. We therefore believe that the significance of our contribution lies not only in the specific peptide examples presented but also in the methodological advance of the computational model.

2. When pointing to the "wet lab" validation problems, I did not ask for justification of the NanoDrop use (that instrument would have problems even for the W-containing sequences in the SpotArray), but to critically address experimental problems and respectful tone down of the statements regarding validity of the experimental conclusions in the discussion section at least. (see reflection to p.5 below). Also, these experimental problems should prompt the authors to reconsider their nomenclature (see above) and avoid generalizations, like calling the "MDA-MB-231 mid uM cytotoxic peptides" as "ACPs" and "PBMC-low lytic peptides" as "non-toxic against healthy human cells". If the Editor suggests and the authors are willing to implement this request, they should pay special attention to their "standard in the field" statement on the IC₅₀ determination shown in Figs. 7 & 8, which claim "3 biological/4 technical replicates", but the results (Figs. 7 vs. 8) differ significantly (e.g., against MDA-MB-231). To judge the values, I would need to examine the raw data or have a definition of a biological/technical replicate as understood by the authors.

Response: We thank the reviewer and acknowledge the need to qualify our conclusions. We agree that some experimental limitations could affect the strength of our findings and so in the revised Discussion, we have moderated our statements and adjusted the terminology used throughout the manuscript. In particular, rather than broadly referring to peptides as “ACPs” or “non-toxic against healthy human cells,” we now describe them more precisely as inhibitory towards MDA-MB-231 cells and MDA-MB-231/PBMC discriminating (see our response to #4 for examples) as recommended by the reviewer. In accordance, we have also updated the section title “Novel

peptides selectively inhibited the viability of MDA-MB-231 breast cancer cells.” to “*Novel peptides selectively discriminated MDA-MB-231 cells from healthy human blood cells*”

We have also addressed the lack of clarity regarding the replicates; in our study, biological replicates refer to independent experiments performed on different days with freshly prepared cell cultures, while technical replicates refer to repeated measurements within the same experiment (i.e., parallel wells in the same plate). We have added this definition to the Methods section, as suggested (P42 L2-4).

We also thank the reviewer for pointing our oversight in number of biological and technical replicates. We have now corrected it in figure captions. For Figure 7, the data represent 2 biological replicates and 4 technical replicates. For Figure 8, the data represent 3 biological replicates and 3 technical replicates in MDA-MB-231 cells. In addition, we have provided the raw data for review.

3. Regarding the CG-MD. For me, it sounds contradictory if in the manuscript the authors suggest only validation and mechanistic plausibility evaluation (as the purpose of MD), but in the response to my critique, they claim they would have revised (“had the MD results contradicted our hypothesis, we would have revisited and refined our discovery approach”). The same kind of validation/plausibility information can be obtained from analyzing the physicochemical properties and composition, rather than spending computational time on simulations in cholesterol-free symmetric bilayers, which yields output that is difficult to correlate with RBC and/or PBMC membrane-perturbing results. If the MD data is omitted from the main manuscript, it will not change either in the narrative or in the conclusions. I believe the *in silico* data warrant SI in the context of this manuscript or a separate publication if the authors wish to avoid wasting the results. Further to this point, Fig.1A shows CG-MD as part of the first TARSA application. Was it also applied to the 36M helices of the PDBlarge? Further in Fig. 1 B, “Membrane MD” appears to be part of TARSA, with an unclear function, as the authors claim not to use it for filtering (same for the “literature properties”).

Response: We thank the reviewer for this thoughtful critique and for highlighting the need to clarify the role of CG-MD in our study. We agree that CG-MD simulations were not intended as a filtering or selection step in the TARSA pipeline, but rather as a post-hoc validation tool to assess mechanistic plausibility of membrane perturbation for selected peptides. Our earlier response may have unintentionally overstated their influence on discovery decisions. To clarify: had CG-MD results indicated grossly inconsistent behavior with our working hypothesis, we would have interpreted this as a sign to refine our mechanistic assumptions but not to retroactively alter the selection of peptides already prioritized.

The reviewer is correct that certain aspects of peptide–membrane interactions can be inferred from physicochemical properties and composition. In agreement with reviewer’s suggestion, we have moved the CG-MD section to SI entirely and pointed the reader to appendix F4 in the SI where these results are discussed (P22 L1-2).

Regarding Fig. 1A and 1B, we acknowledge that our schematic may give the impression that membrane MD was systematically applied at the large-library screening stage. To clarify: CG-MD was **not applied** to all the 36M helices of PDBlarge. Instead, it was performed only on top-480 out of 3.2M helical motifs discovered by TARSA from PDBlarge. As suggested by the reviewer,

we have updated the figure 1 to remove the CG-MD and Literature Properties assessment as they do not directly affect the results.

4. Even with this clarification of the "potency" meaning, from the manuscript, it should be clear at which concentrations of peptides and for which cells the parameter is applied every time.

Response: We have omitted the term potency when used ambiguously in *in silico* context. Throughout the main text, we have replaced it with explicit mention of the corresponding cell-line and % cell-inhibition.

While all changes are tracked in the updated manuscript, the most notable changes are the following,

- Table 1, updated description to "Cell inhibition at 12 μ M of Mastoparan single amino acid substitution derivatives on TNBC cell" and "Cell inhibition at 12 μ M of Mastoparan derivatives on healthy human PBMC cells" respectively.
- In Fig 2(c) caption, replaced "potency prediction" with "*percentage cell-inhibition of PBMC and MDA-MB231 cells at 12.5 μ M*"
- On P14, L17-18, replaced "high-potency peptides" with "peptides predicted to have high MDA-MB231 cell inhibition"
- P15, L12-13: Replaced "average predicted potency of 45.2%" with "average predicted cell inhibition of 45.2% towards MDA-MB231 cells."
- *Updated Fig.4 Caption as follows "(C) Retrieval of MDA-MB-231 inhibitory peptides from the PDB_{12mer} dataset using the TARSA policy. (D-F) Exploration and discovery of peptides with high predicted MDA-MB-231 inhibition in the GEN_{12mer} dataset, with TARSA avoiding low-potency regions."*
- P17 L12-13: Replaced "TARSA not only achieved a faster screening time but also identified peptides with a higher average predicted potency" with "... higher average predicted MDA-MB-231 cell inhibition"
- P18 L12-13 Replaced "TARSA's ability to prioritize the discovery of ACP" with "TARSA's ability to prioritize the discovery of highly membranolytic candidates"

We have also specified that all predictive models were trained at 12.5 μ M concentration. Specifically, for MDA-MB-231 cells, on P11 L14-16, we say

As such, in this study, the effectiveness of various peptide representations for modeling membranolytic Mastoparan-like helical peptides, and ML algorithms to predict the cell inhibition of MDA-MB231 breast cancer cells at 12.5 μ M were investigated.

Similarly, for PBMC, on P22 L14-15, we say

cytotoxicity toward healthy white blood cells (PBMCs) at 12.5 μ M was predicted for the filtered candidates ($p = 0.75 \pm 0.11$)

5. I did not stress in the second round (maybe I should have) that some of my initial concerns/suggestions were not addressed/ignored, despite a very lengthy ChatGPT-assisted answer, mentioning only the identity of the library origin for experiments in the validation step. (I would still expect all of my concerns to be implemented in the further revisions, should the

Editor conclude on it). However, now the authors provide (rather hide in the supplementary) the identity, but without any indication in the manuscript main text and a clear justification (or at least a comment), explaining why out of 110K (in the fig.1)/200K (in the text) they take "Top 95 of PDB12mer candidates" and from "2.8M of Gen12mer candidates" only "Top 10" for validation. For the sake of real pipeline validation, it would have been incredibly useful to keep the two libraries separated. E.g., in the validation step for the PDB12mers, their approach formally provides a ca. 12% hit rate, whereas the GEN12mer library can be claimed to have a 40% hit rate.

Response: We sincerely thank the reviewer for pointing out this oversight and for reiterating the importance of clarity in describing our validation process. Our answer was created by us, not ChatGPT, and all of the references we cited in our response actually exist and were read by us.

The peptide libraries used for validation were drawn from two distinct sources: (i) *PDB12mer* (structurally derived sequences) and (ii) *GEN12mer* (de novo generated sequences). From these, we tested the *Top-95 PDB12mer candidates* and the *Top-10 GEN12mer candidates*. The discrepancy in numbers reflected our initial prioritization strategy i.e. we focused more heavily on PDB-derived sequences for experimental validation because they provided a natural structural reference set. In contrast, GEN12mers were included primarily to assess whether our pipeline could extend beyond known structural motifs, which is why a smaller subset was tested. We have now included this justification on P23 L2-5. We say,

“As this study prioritized validation of PDB-derived peptide libraries through virtual screening, with generative models included as a secondary exploratory module, 95 peptides were chosen from the PDB_{12mer} library and 10 peptides from the Gen_{12mer} library.”

We further agree that presenting the validation results with explicit separation of PDB12mer and GEN12mer candidates would have been more informative. Indeed, as the reviewer points out, the observed hit rates differ between the two libraries in our SPOT array validation. Of the 15 that reduced cancer cell viability by >60%, 11/95 were derived from PDB12mer and 4/10 from GEN12mer. We have included this information along with the origin library of corresponding peptides in Figure 7 caption. We say,

“Eleven out of 95 considered Peptides (ID<100) originated from PDB_{12mer} while the remaining four out of 10 considered peptides originated from GEN_{12mer}”

6. There are further corrections required - still different phases of the suggested approach are non-uniformly described/illustrated, and some minor inconsistencies (in terminology and word capitalization) persist, etc.

Response: We thank the reviewer for pointing out lingering inconsistencies in the manuscript. We have now carefully re-read the entire submission to ensure uniformity. All suggestions raised by the reviewer in this and previous rounds have also been incorporated to the best of our knowledge. If there are any additional specific issues the reviewer would like to highlight, we would be happy to address them promptly.